

**Examination of effects of aerosols on a pyroCb and their dependence on fire**
**intensity and aerosol perturbation using a cloud-system resolving model**
Seoung Soo Lee[1], George Kablick III[2,3], Zhanqing Li[2]
[1]Research Foundation, San Jose State University, San Jose, California, USA
[2]Earth System Science Interdisciplinary Center, University of Maryland, College Park,
Maryland, USA
[3]US Naval Research Laboratory, Washington, DC, USA





## Abstract

This study investigates how a pyrocumulonimbus (pyroCb) event influences water vapor concentrations and cirrus cloud properties near the tropopause, specifically focusing on how fire-produced aerosols affect this role via a modeling framework. Results from a case study show that when observed fire intensity is high, there is an insignificant impact of fire-produced aerosols on the convective development of the pyroCb and associated changes in water vapor and the amount of cirrus cloud near the tropopause. However, as fire intensity weakens, the effects of aerosols on microphysical variables and processes such as droplet size and autoconversion increase. Modeling results shown herein indicate that aerosol-induced invigoration of convection is significant for pyroCb with weak-intensity fires and associated weak surface heat fluxes. Thus, there is a greater aerosol effect on the transportation of water vapor to the upper troposphere and the production of cirrus cloud with weak-intensity fires, whereas these effects are muted with strong-intensity fires.



## 1. Introduction

A recent study by Kablick et al. (2018) has shown that pyrocumulonimbus (pyroCbs) can
transport significant amounts of water vapor to the upper troposphere and the lower
stratosphere (UTLS) and thus may have a role in seasonal UTLS water vapor budgets. Any
change in water vapor in the UTLS has an exceptionally strong influence on the global
radiation budget and thus Earth's climate (Solomon et al., 2010). PyroCbs involve and
control cirrus clouds around their tops that reach the UTLS. Changes in cirrus clouds in the
UTLS are known to have a strong influence on the global radiation budget (Solomon et al.,
2010). The examination of mechanisms through which pyroCbs affect water vapor and
cirrus clouds in the UTLS can thus be a way of better understanding climate changes.
By definition, pyroCbs initiate over a fire, and the large surface energy release affects
their dynamic, thermodynamic and microphysical development. The dynamics of these
events has been shown to be mostly controlled by fire-induced latent and sensible heat
fluxes at and near the surface. However, questions remain about what role the large
concentration of cloud condensation nuclei (CCN) contained in smoke has on the vertical
development and microphysical properties. Studies (e.g., Rosenfeld et al., 2008; Storer et
al., 2010; Tao et al., 2012) have shown that aerosols affect cumulonimbus clouds, and this
raises a possibility that fire-generated aerosols affect pyroCb development. As an example
of aerosol impacts on cumulonimbus clouds, these studies have demonstrated that increases
in aerosol loading can make the size of droplets (i.e., cloud-liquid particles) smaller.
Individual aerosol particles act as seeds for the formation of droplets and thus increasing
aerosol loading or increasing aerosol concentrations lead to more droplets formed. More
droplets mean more competition among them for available water vapor needed for their
condensational growth, and this more competition makes individual droplets smaller.
Aerosol-induced smaller sizes of droplets reduce the efficiency of the growth of cloud-
liquid particles to raindrops via autoconversion that is a collection process among cloud-
liquid particles for them to grow to be raindrops, given that the efficiency is proportional
to the sizes. This reduced efficiency leads to less cloud liquid converted to rain. More
cloud liquid is thus available for transport to places above the freezing level by updrafts.



This eventually induces more freezing of cloud liquid, which enhances parcel buoyancy,
and this enhancement invigorates updrafts and associated convection.

Compared to the research done on the role played by fire-generated heat fluxes in

the development of pyroCbs and their effects on water vapor and cirrus clouds in the UTLS,
the research on that role by fire-generated aerosols has been scarce. Motivated by this lack
of understanding, this paper focuses on the role by those aerosols in the development of a
pyroCb and its effects on water vapor and cirrus clouds in the UTLS. To examine that role,
this study extends the previous modeling work that was described in Kablick et al. (2018).
That modeling work compared effects of fire-generated heat fluxes on the development of
a pyroCb and its impacts on the UTLS water vapor and cirrus clouds to those of fire-
generated aerosols. In that comparison, those effects of fire-generated aerosols were shown
to be negligible as compared to those effects of heat fluxes. However, aerosol effects on
cloud development vary with cloud basic properties such as basic updrafts that are
determined by environmental conditions (e.g., Khain et al., 2008; Lee et al., 2008; Tao et
al., 2012). Basic updrafts are determined by environmental instability as represented by
convective available potential energy (CAPE). Lee et al. (2008) have shown that different
clouds with different basic updrafts, which are due to different CAPE, show different
sensitivity of cloud microphysical and thermodynamic development to aerosol
concentration. Hence, it is hypothesized that aerosol effects on the pyroCb development
and its impacts on the UTLS water vapor and cirrus clouds can vary depending on the
intensity of the pyroCb basic updrafts.
Based on this hypothesis, to examine the potential variation of aerosol effects on the
pyroCb development and its impacts on the UTLS water vapor and cirrus clouds with the
varying basic updrafts of pyroCbs, numerical simulations are performed. These simulations
are for a case of a pyroCb which is identical to that in Kablick et al. (2018), and performed
by using a cloud-system resolving model (CSRM) which is able to resolve cloud-scale
dynamic, thermodynamic and microphysical processes. By resolving these processes that
play a critical role in the development of clouds and their interactions with aerosols, we are
able to obtain confident information on aerosol effects on the pyroCb development and its
impacts the UTLS water vapor and cirrus clouds, and on associated dynamic,
thermodynamic and microphysical mechanisms. The basic modeling methodology in this





study is similar to that used by Kablick et al. (2018). However, this study uses a more
sophisticated microphysical scheme, i.e., a bin scheme, rather than the two-moment bulk
scheme used by Kablick et al. (2018). Note that Kablick et al. (2018) examined aerosol
effects on the convective development of a specific pyroCb case study, simulating
microphysical conditions, detrained water vapor mixing ratios, and cirrus cloud properties
only considering a basic updraft framework. The present study expands upon that work by
performing sensitivity simulations in which basic updrafts in the pyroCb are allowed to
vary, enabling us to ascertain the dependence of those aerosol effects on basic updrafts.
Note that CAPE, which determines basic updrafts in convective clouds, are strongly
dependent on surface latent and sensible heat fluxes (e.g., Houze, 1993), and in the case of
pyroCb these fluxes are controlled by fire intensity. Hence, these sensitivity simulations in
turn enable us to study the dependence of those aerosol effects on fire intensity. Here, we
see that the pyroCb basic updrafts are controlled by fire intensity and thus the pyroCb basic
updrafts are referred to as fire-driven updrafts, henceforth.
Aerosol effects on clouds are initiated by an increase in aerosol concentration, which
can be caused by an increase in aerosol emission at and near the surface, and dependent on
how much aerosol concentration increases, or on the magnitude of an increase in aerosol
concentration, i.e., aerosol perturbation (e.g., Rosenfeld et al., 2008; Koren et al., 2012).
This dependence has not been examined in Kablick et al. (2018) and this study examines
this dependence by performing additional sensitivity simulations where the magnitude of
aerosol perturbation varies.

**2. CSRM**

We use the Advanced Research Weather Research and Forecasting (ARW) model, a
nonhydrostatic compressible model, as the CSRM. Prognostic microphysical variables are
transported with a fifth-order monotonic advection scheme (Wang et al., 2009). Shortwave
and longwave radiation parameterizations have been included in all simulations by
adopting the Rapid Radiation Transfer Model (RRTM; Mlawer et al., 1997; Fouquart and
Bonnel, 1980).



To represent the microphysical processes, the CSRM adopts a bin scheme based on
the Hebrew University Cloud Model described by Khain et al. (2009). The bin scheme
solves a system of kinetic equations for the size distribution functions of water drops, ice
crystals (plate, columnar and branch types), snow aggregates, graupel and hail, as well as
cloud condensation nuclei (CCN) and ice nuclei (IN). Each size distribution is represented
by 33 mass doubling bins, i.e., the mass of a particle $m_k$ in the $k$th bin is determined as $m_k$
$= 2m_{k-1}$.
The cloud-droplet nucleation parameterization, which is based on Köhler theory, is
used to represent cloud-droplet nucleation. Arbitrary aerosol mixing states and arbitrary
aerosol size distributions can be fed to this parameterization. To represent heterogeneous
ice-crystal nucleation, the parameterizations by Lohmann and Diehl (2006) and Möhler et
al. (2006) are used. In these parameterizations, contact, immersion, condensation-freezing,
and deposition nucleation paths are all considered by taking into account the size
distribution of IN. Homogeneous aerosol (or haze particle) and droplet freezing, based on
the size distribution of droplets, is also considered following the theory developed by Koop
et al. (2000).

**3. Case description and simulations**


**3.1 Control run**


The control run for an observed pyroCb case is performed over a forested site in the
Canadian Northwest Territories (60.03° N, 115.45° W). Kablick et al. (2018) give details
about the site and the pyroCb case. The control run is identical to the Full Simulation in
Kablick et al. (2018) except for the different microphysical schemes between them;
remember that this study uses a bin scheme, while Kablick et al. (2018) used a bulk scheme.
The control run is performed for one day from 12:00 GMT on August 5[th] to 12:00 GMT
on August 6[th] in 2014 and captures the initial, mature, and decaying stages of the pyroCb.
This simulation is performed in a three dimensional domain with horizontal and vertical
lengths of 300 km and 20 km, respectively. For the simulation, the horizontal resolution is



500 m and the vertical resolution is 200 m to resolve cloud dynamic, thermodynamic and
microphysical processes.
Figure 1 shows a satellite image of the observed pyroCb when it is about to advance
into its mature stage. In Figure 1, the red circle marks the fire spot whose spatial length is
~ 40 km. To emulate this in the simulation, at the center of the simulation domain, a fire
spot with a diameter of 40 km is placed as shown in Figure 2. In the fire spot, the surface
latent and sensible heat fluxes are set at 1800 and 15000 W m$^{-2}$, respectively. In areas
outside of the fire spot in the domain, the surface latent and sensible heat fluxes are set at
310 and 150 W m$^{-2}$, respectively. These surface heat-flux values follow the previous
studies which are Trentmann et al. (2006) and Luderer et al. (2006) and adopt boreal forest
emissions. Following Kablick et al. (2018), the surface heat-flux values are prescribed
with no temporal variation and no consideration of interactions between heat fluxes and
the atmosphere in the control run. Hence, the setup for the surface heat fluxes is idealized
and this enables a better isolation of aerosol effects themselves on the pyroCb development
and its impacts on the UTLS water vapor and cirrus clouds for the given surface heat fluxes
by excluding effects of interactions between the surface heat fluxes and atmosphere on
those development and impacts.
For the selected pyroCb case, aerosol properties that can be represented by aerosol
chemical composition, size distribution and concentration are unknown. Hence, in the fire
spot for the first time step, the concentration of aerosols acting as CCN is prescribed to be
15000 cm$^{-3}$ in the planetary boundary layer (PBL), and decreases exponentially with height
above the PBL top. Outside of the fire spot for the first time step, the concentration of
aerosols acting as CCN is prescribed to be 150 cm$^{-3}$ in the PBL and also decreases
exponentially with height above this layer. For the control run, the other aerosol properties
are assumed to follow typical values determined in previous studies. For example, Reid et
al. (2005) have shown that aerosol mass produced by forest fires is generally composed of
~50-70% of organic-carbon (OC) compounds, ~5-10% of black-carbon (BC) material, and
~20-45% of inorganic species. Based on those results, the approximate median value of
each chemical component percentage range is used in the control run. Aerosol particles
are assumed to be composed of 60% OC, 8% BC, and 32% inorganic species. In the control
run, OC is assumed to be water soluble and composed of (by mass) 18 % levoglucosan



(C6H10O5, density = 1600 kg m$^{-3}$, van't Hoff factor = 1), 41 % succinic acid (C4H6O4,
density = 1572 kg m$^{-3}$, van't Hoff factor = 3), and 41 % fulvic acid (C33H32O19, density
= 1500 kg m$^{-3}$, van't Hoff factor = 5) based on typically observed chemical composition of
OC compounds over fire sites (Reid et al., 2005). In the control run, the inorganic species
is assumed to be ammonium sulfate, a representative inorganic species associated with fires
(Reid et al., 2005). This chemical composition taken for aerosol particles is assumed to be
spatiotemporally unvarying in the control run. According to Reid et al. (2005),
Knobelspiessel et al. (2011), and Lee et al. (2014), it is reasonable to assume that the initial
aerosol size distribution follows the unimodal lognormal distribution in fire sites. Hence,
the control run adopts the unimodal lognormal distribution as an initial aerosol size
distribution. Those studies have indicated that in general, median aerosol diameter and
standard deviation of the distribution range from ~0.01 to ~0.03 μm and from ~2.0 to ~2.2,
respectively. By taking the approximate median value of each of these ranges, median
aerosol diameter and standard deviation of the adopted unimodal distribution are assumed
to be 0.02 μm and 2.1, respectively, for the control run. The unimodal distribution, which
is adopted by the simulation, in the PBL over the fire spot is shown in Figure 3 with log-
scales for x- and y-axes. For the control run, aerosol properties of IN and CCN are assumed
to be identical except that at the first time step, the IN concentration is 100 times lower
than the CCN concentration. This is based on a general difference in concentration between
CCN and IN (Pruppacher and Klett, 1978).
In Figure 1, the observed cirrus cloud at the top of the pyroCb is located to the
northeast of the fire spot due to the northeastward winds at the altitude of the cirrus cloud.
The cloud first formed around the fire spot. However, winds advected it northeastward.
The extent of the observed cirrus cloud is ~100 km. Figure 2 shows the field of cloud-ice
mass density at the top of the simulated pyroCb and at a time that corresponds to the
satellite image in Figure 1. This field in Figure 2 represents the simulated cirrus cloud in
the control run.  As observed, the simulated cirrus is located to the northeast of the fire spot
and the extent of the simulated cirrus cloud is ~ 100 km. Hence, we see that there is good
agreement in the morphology of the cirrus cloud between the observation and the
simulation.





238   Figure 4 shows the vertical distribution of the cloud reflectivity field in dBZ, which

239  is observed by the Cloudsat and averaged over the Cloudsat path, and its simulated

240  counterpart in the control run. The details of the reflectivity field are given in Kablick et al.

241  (2018). There is good agreement between observed and simulated cloud reflectivity fields.

242  The agreement in the observed and simulated cirrus cloud and reflectivity fields

243  demonstrates that the pyroCb-case simulation is reasonable.

244

245  **3.2 Low-aerosol run**

246

247  To see the role played by fire-generated aerosols in the development of the pyroCb and its

248  effects on water vapor and cirrus clouds in the UTLS, we repeat the control run by reducing

249  aerosol concentration in the fire spot from 15000 $cm^{-3}$ to the background aerosol

250  concentration (i.e., 150 $cm^{-3}$). This reduction removes fire-generated aerosols in the fire

251  spot. The only difference is in aerosol concentration in the fire spot and there are no other

252  differences in the simulation setup which is described in Section 3.1 between the control

253  run and this repeated run. Hence, comparisons between the control run and this repeated

254  run, which is referred to as the low-aerosol run, will identify the role played by fire-

255  generated aerosols in the pyroCb development and its impacts on the UTLS water vapor

256  and cirrus clouds. Here, the low-aerosol run is identical to the Low Aerosol Simulation in

257  Kablick et al. (2018) except for the different microphysical schemes between them.

258

259  **3.3 Additional runs**

260

261  We examine the above-mentioned potential variation of effects of fire-generated aerosols

262  on the pyroCb development and its impacts on the UTLS water vapor and cirrus clouds

263  with varying fire intensity and associated fire-driven updrafts. For the examination, we

264  repeat the control run by varying fire intensity. Remember that surface latent and sensible

265  heat fluxes on which fire-driven updrafts in convective clouds are strongly dependent are

266  controlled by fire intensity. Hence, fire intensity can be represented by fire-induced surface

267  latent and sensible heat fluxes and the variation of fire intensity in the repeated control run

268  is accomplished by the variation of these fire-induced surface heat fluxes. As a first step





for the examination, the control run is repeated by reducing fire-induced surface latent and
sensible heat fluxes by a factor of 2. Then, the control run is repeated again by reducing
these fluxes by a factor of 4. The first repeated run represents a case with medium fire
intensity, while the second repeated run represents a case with weak fire intensity. Relative
to these repeated runs, the control run represents a case with strong fire intensity.
Henceforth, the first repeated run is referred to as "the medium run" and the second
repeated run is referred to as "the weak run". Then, to see effects of fire-generated aerosols
on the pyroCb development and its impacts on the UTLS water vapor and cirrus clouds for
each of those cases with different fire intensity, the medium run and the weak run are
repeated with the identical initial aerosol concentration to that in the low-aerosol run. The
repeated medium run and weak run are referred to as "the medium-low run" and "the weak-
low run", respectively. The control run, the medium run, and the weak run are the polluted-
scenario runs, while the low-aerosol run, the medium-low run, and the weak-low run are
the clean-scenario runs. Comparisons between the medium run and the medium-low run
and those between the weak run and the weak-low run isolate those effects of fire-generated
aerosols for the case of medium fire intensity and the case of weak fire intensity,
respectively. Comparisons between the control run and the low-aerosol run identify those
aerosol effects for the case of strong fire intensity.
Effects of fire-generated aerosols on the pyroCb development and its impacts on the
UTLS water vapor and cirrus clouds can also be dependent on how large fire-induced
increases in aerosol concentrations are or the magnitude of fire-induced aerosol
perturbation in a fire spot. Motivated by this, the previously described simulations are
repeated by varying the magnitude of aerosol perturbation in the fire spot. To test the
sensitivity of results to the magnitude of fire-induced aerosol perturbation, for each fire
intensity, we repeat the polluted-scenario run by increasing and reducing the magnitude by
a factor of 2 in the fire spot but not outside of the fire spot. These simulations with the
increased magnitude have an aerosol concentration of $30000\ cm^{-3}$ at the first time step over
the fire spot in the PBL and are referred to as the control-30000 run, the medium-30000
run, and the weak-30000 run for strong, medium, and weak fire intensity, respectively.
These simulations with the reduced magnitude have an aerosol concentration of      7500
$cm^{-3}$ at the first time step over the fire spot in the PBL and are referred to as the control-



7500 run, the medium-7500 run, and the weak-7500 run for strong, medium, and weak fire
intensity, respectively. Motivated by the analysis described in Section 4.2, we additionally
repeat the medium run and the weak run with aerosol concentrations of   2000 and 1000
cm$^{-3}$ at the first time step over the fire spot in the PBL, respectively. The repeated medium
(weak) run is referred to as the medium-2000 (the weak-1000) run. Table 1 summarizes
the simulations.

### 4.  Results


Results from the control run and the low-aerosol run, which are equivalent to the Full
Simulation and the Low Aerosol Simulation in Kablick et al. (2018), respectively, are
described here. Kablick et al. (2018) mainly focused on comparisons themselves between
aerosol effects and heat-flux effects on pyroCb development and its impacts on the UTLS
water vapor and cirrus clouds. In this study, we expand upon the results of Kablick et al.
(2018) by providing additional details of the simulation results by focusing on the impacts
of pyroCb development on the UTLS water vapor and cirrus clouds, and aerosol effects on
pyroCb development.

Figure 5 shows the vertical distributions of the averaged updraft mass fluxes over

cloudy areas, i.e., areas where the sum of liquid-water content (LWC) and ice-water content
(IWC) is non-zero, and over the simulation period between 17:00 GMT on August 5$^{th}$ and
12:00 GMT on August 6$^{th}$ in the control run and the low-aerosol run for the case of strong
fire intensity. 17:00 GMT on August 5$^{th}$ is a time around which the pyroCb starts to from
and 12:00 GMT on August 6$^{th}$ is the end of the simulation period.  In this study, drops with
radii smaller (greater) than 20 µm are classified as droplets (raindrops). For the calculation
of LWC (IWC), we only considered droplets (ice crystals). Stated differently, droplet mass
but not rain mass is used to obtain LWC and the mass of ice crystals but not the mass of
snow aggregates, graupel and hail is used to obtain IWC.

The updraft mass flux is one of the most representative variables that are indicative

of the cloud dynamic intensity and the magnitude of convective invigoration. As seen in
Figure 5, the control run and the low-aerosol run for strong fire intensity have similar
updraft mass fluxes.  Table 2 gives the averaged values of updraft mass fluxes over cloudy


areas at all altitudes and over the simulation period between 17:00 GMT on August 5th and
12:00 GMT on August 6th for simulations. In Figure 5 and Table 2, updraft mass fluxes in
the control run are only ~3% greater than those in the low-aerosol run. Given the
hundredfold difference in aerosol loading over the fire spot between the runs, this 3%
difference in updraft fluxes is negligibly small.
Water vapor around and above the tropopause or the UTLS plays an important role in
the global radiation budget, thus garnering much attention from the climate-change
community. Motivated by this, we examine the role played by the pyroCb in the UTLS
water vapor.  The vertical distributions of averaged water-vapor mass density around and
above the tropopause (~ 13 km) are shown in Figure 6. The water-vapor mass density
shown by colored lines in Figure 6 is averaged over cloudy grid columns that have the non-
zero sum of liquid-water path (LWP) and ice-water path (IWP) and over the simulation
period between 17:00 GMT on August 5th and 12:00 GMT on August 6th. For the
calculation of LWP (IWP), we only considered droplets (ice crystals) as for the calculation
of LWC (IWC). The black line in Figure 6 shows the average of water-vapor mass density
over non-cloudy grid columns that have the zero sum of LWP and IWP and over the
simulation period between 17:00 GMT on August 5th and 12:00 GMT on August 6th in the
control run. This represents the background water-vapor mass density. Table 2 shows the
averaged values of water-vapor mass density over altitudes between 13 and 16 km for
simulations.
As seen in Figure 6, 16 km is an altitude to which the non-zero water-vapor mass
density over cloudy columns extends. The comparison between water-vapor mass density
over the cloudy columns and that over non-cloudy columns in the control run as shown in
Figure 6 and Table 2 demonstrates that there is a substantial increase in the amount of water
vapor in the UTLS due to the pyroCb. There is about five times greater water-vapor mass
over the cloudy columns that represent the pyroCb area than in the background outside the
pyroCb area in the control run.
Updrafts in the pyroCb transport water vapor to the UTLS, which leads to the
substantial increase in the amount of water vapor in the UTLS over the pyroCb area. For
the simulation period between 17:00 GMT on August 5th and 12:00 GMT on August 6th,
the averaged water-vapor mass fluxes at the tropopause over cloudy and non-cloudy grid



columns are $8.30 \times 10^{-6}$ and $0.57 \times 10^{-6}$ kg m$^{-2}$ s$^{-1}$, respectively. Due to the presence of the
pyroCb and associated updrafts in cloudy grid columns, there are substantial increases in
water vapor fluxes at the tropopause over those cloudy grid columns as compared to those
fluxes in the background over non-cloudy grid columns. This leads to larger water vapor
mass in the UTLS over the pyroCb than in the background outside the pyroCb or the
pyroCb area in the control run. It is also shown that the vertical extent of water vapor is
extended further up to ~ 16 km by the pyroCb as compared to the extent of ~14 km in the
background (Figure 6).

Similar to the situation with updraft mass fluxes, there is only a small (~2%) increase

in the averaged water vapor mass in the UTLS, as shown in Figure 6 and Table 2, in the
control run as compared to that in the low-aerosol run for strong fire intensity. This small
variation in updraft mass fluxes between the control run and the low-aerosol run also results
in a small variation in the transportation of water vapor to the UTLS and the averaged
water-vapor fluxes at the tropopause between these two simulations. These averaged
fluxes are over cloudy columns for the simulation period between 17:00 GMT on August
5$^{th}$ and 12:00 GMT on August 6$^{th}$. The averaged water-vapor fluxes vary from $8.30 \times 10^{-6}$
kg m$^{-2}$ s$^{-1}$ in the control run to $8.21 \times 10^{-6}$ kg m$^{-2}$ s$^{-1}$ in the low-aerosol run.

In addition to water vapor in the UTLS, cloud ice or ice crystals around the tropopause

play an important role in the global radiation budget. These ice crystals comprise cirrus
clouds. To identify the role played by the pyroCb in cirrus clouds, Figure 7 shows the
averaged cloud-ice mass density over cloudy areas for the simulation period of 17:00 GMT
on August 5$^{th}$ to 12:00 GMT on August 6$^{th}$ in the simulations. The altitude of homogeneous
freezing is at ~ 9 km , so cirrus clouds which are composed of ice crystals only are between
~ 9 km and ~13 km. Between ~ 9 km and ~ 13 km, the averaged cloud-ice mass density in
the control run is shown in Figure 7, and ranges from ~ 0.028 to ~ 0.037 g m$^{-3}$. In non-
cloudy areas and in the clear sky background outside the pyroCb, cloud-ice mass density
equals zero. Hence in the control run, the pyroCb increases cloud-ice mass density between
~0.028 to ~0.037 g m$^{-3}$ over the background.

Updrafts in the pyroCb produce supersaturation, which leads to the generation of cloud-

ice mass and associated cirrus clouds via deposition, the primary source of cloud-ice mass.
Similar to the situation with updraft mass fluxes, comparisons between the control run and





the low-aerosol run for strong fire intensity show that there is only a small increase (~4%)
in cloud-ice mass in the control run particularly between 9 km and 13 km, as shown in
Figure 7 and Table 2, as compared to that in the low-aerosol run. Due to the negligible
variation of updraft mass fluxes, there are negligible variations of supersaturation and
deposition between the simulations as seen in Figure 8, and thus a negligible variation of
cloud-ice mass between the control run and the low-aerosol run. Figure 8 shows the vertical
distributions of the averaged deposition rate over cloudy areas for the simulation period of
17:00 GMT on August 5[th] to 12:00 GMT on August 6[th].

In summary, the pyroCb and associated updrafts cause a substantial enhancement of

the transportation of water vapor to the UTLS. They also produce cirrus clouds. The role,
which is played by fire-generated aerosols and their effects on the pyroCb and its updrafts,
in the enhancement of the transportation of water vapor to the UTLS and in the production
of cirrus cloud is not significant for strong fire intensity.

**4.1 Dependence of aerosol effects on fire intensity**

The weak sensitivity of updrafts, water vapor, and cirrus clouds to aerosol loading in the
pyroCb may be related to fire intensity. When fire-generated surface heat fluxes and fire
intensity are increased, it is likely that in-cloud latent heat is also increased because a major
source of in-cloud latent heating is surface heat flux. Therefore, the aerosol-induced
perturbations of latent heating may be relatively small compared with large in-cloud latent
heat contributed by surface fluxes with very intense burning. In other words, aerosol-
induced increases in parcel buoyancy and updrafts are relatively small compared with the
large buoyancy and strong fire-driven updrafts produced by strong fire intensity and the
associated large in-cloud latent heat.

Considering that a major source of in-cloud latent heat is surface heat fluxes, when the

fire-generated surface heat fluxes and the fire intensity are reduced, in-cloud latent heat is
also likely to be smaller. Here, we are interested in how the magnitude of an aerosol-
induced perturbation of latent heating for a pyroCb with weak fire intensity is compared to
that with strong fire intensity. This is just based on a possibility that with background in-



cloud latent heat varying with fire intensity, the relative magnitude of aerosol-induced
perturbation of latent heat to surface flux-dominated latent heat may vary.

**4.1.1 Updrafts and the UTLS water vapor and cirrus cloud**

Figure 9 shows the averaged updraft mass fluxes over cloudy areas at each altitude in the
runs with different fire intensity for the simulation period of 17:00 GMT on August 5th to
12:00 GMT on August 6th.  Here, the averaged updraft mass fluxes in the low-aerosol run,
the medium-low run and the weak-low run represent fire-driven updrafts for strong,
medium and weak fire intensity, respectively. Due to different fire intensity and associated
CAPE, fire-driven updrafts vary between these runs. The variation of these fluxes between
the low-aerosol run and the control run for strong fire intensity, between the medium-low
run and the medium run for medium fire intensity, and between the weak-low run and the
weak run for weak fire intensity, respectively, is induced by fire-generated aerosols. As
seen in Figure 9 and Table 2, all of the cases of weak, medium and strong fire intensity
show aerosol-induced increases in updraft mass fluxes. Of interest is that the percentage
increase in updrafts in the case of weak fire from those in the weak-low run to those in the
weak run is the greatest, while the percentage increase in the case of strong fire from
updrafts in the low-aerosol run to those in the control run is the smallest. The percentage
increase for the case of medium fire from updrafts in the medium-low run to those in the
medium run is intermediate (Figure 9 and Table 2). Here, the percentage difference,
including both the percentage increase and decrease, is the relative difference in the value
of variables between the control run and the low-aerosol run for strong fire intensity,
between the medium run and the medium-low run for medium fire intensity or between the
weak run and the weak-low run for weak fire intensity. This percentage difference for
strong fire intensity is obtained as follows in this study:

$\frac{The\ control\ run\ minus\ the\ low-aerosol\ run}{The\ low-aerosol\ run} \times 100\,(\%)$       (1)

The percentage difference for medium fire intensity is obtained by replacing the control
run with the medium run and replacing the low-aerosol run with the medium-low run in



Equation (1). The percentage difference for weak fire intensity is obtained by replacing the
control run with the weak run and replacing the low-aerosol run with the weak-low run in
Equation (1). Associated with those greatest increases in updraft mass fluxes, the
percentage increases in water vapor and cloud-ice mass in the UTLS, which are calculated
by Equation (1), are the greatest in the case of weak fire as seen in Figures 10 and 11 and
Table 2. Figures 10 and 11 show the averaged water-vapor and cloud-ice mass density at
each altitude, respectively, over cloudy areas for the simulation period of 17:00 GMT on
August 5th to 12:00 GMT on August 6th in the runs with different fire intensity. Associated
with those smallest increases in updraft mass fluxes, the percentage increases in water
vapor and cloud-ice mass in the UTLS are the smallest in the case of strong fire as seen in
Figures 10 and 11 and Table 2. Associated with the medium increase in updraft mass fluxes,
the percentage increases in water vapor and cloud-ice mass in the UTLS are intermediary
in the case of medium fire as compared to the cases of strong and weak fire (Figures 10
and 11 and Table 2).

**4.1.2 Volume mean radius of droplets ($R_v$)**

**a.  Cloud droplet number concentration (CDNC) and LWC**

The simulation period is divided into four sub-periods for this next analysis: period 1
between 17:00 and 19:00 GMT on August 5th, period 2 between 19:00 and 21:00 GMT on
August 5th, period 3 between 21:00 GMT and 23:00 GMT on August 5th, and period 4
between 23:00 GMT on August 5th and 12:00 GMT on August 6th. The initial formation of
the pyroCb corresponds with the beginning of period 1, and along with periods 2 and 3
correspond to the initial stages of cloud development, while period 4 corresponds to the
mature and the decaying stages. As seen in Figure 12, CDNC, which is averaged over
cloudy areas at all altitudes and over period 1, decreases as the fire intensity and updrafts
decrease. However, the control run, the medium run, and the weak run have the much
higher averaged CDNC than the low-aerosol run, the medium-low run, and the weak-low
run, respectively. Remember that the control run, the medium run and the weak run have
higher aerosol concentrations than the low-aerosol run, the medium-low run and the weak-





low run, respectively, over the fire spot (Table 1). Increasing CDNC enhances competition
among droplets for a given amount of water, which is available for the condensational
growth of droplets, in a cloud. Enhanced competition eventually curbs the condensational
growth and reduces droplet size, which is represented by $R_v$ in this study. This explains
why $R_v$, which is averaged over cloudy areas at all altitudes and over period 1, is smaller
in the control run than in the low-aerosol run for strong fire intensity, in the medium run
than in the medium-low run for medium fire intensity, and in the weak run than in the
weak-low run for weak fire intensity, respectively,  as seen in Figure 12.  Of interest is that
as fire intensity weakens, although the averaged CDNC reduces, which tends to lower the
competition among droplets, the averaged $R_v$ decreases not only among the control run, the
medium run and the weak run with higher aerosol concentrations over the fire spot but also
among the low-aerosol run, the medium-low run and the weak-low run with lower aerosol
concentrations over the fire spot as shown in Figure 12. This is because $R_v$ is proportional
to $(\frac{LWC}{CDNC})^{\frac{1}{3}}$. Here, LWC represents the given amount of water which is available for the
condensational growth of droplets. This proportionality means that for a given CDNC, a
decrease in LWC also causes $R_v$ to decrease, i. e., a decrease in the available amount of
water for the condensational growth with no changes in CDNC induces a decrease in $R_v$.
As shown in Figure 12, LWC, which is averaged over cloudy areas at all altitudes and over
period 1, also decreases with weakening fire intensity and updrafts not only among the
control run, the medium run and the weak run but also among the low-aerosol run, the
medium-low run and the weak-low run.  Effects of LWC on $R_v$, which weakens as fire
intensity weakens, outweigh those of CDNC and this leads to the decrease in the averaged
$R_v$ with weakening fire intensity as seen in Figure 12.

Note that the averaged LWC in the control run is similar to that in the low-aerosol run

for strong fire intensity, while the averaged LWC in the medium run is similar to that in
the medium-low run for medium fire intensity. The averaged LWC in the weak run is also
similar to that in the weak-low run for weak fire intensity. The averaged LWC reduces with
weakening fire intensity from that in the control run to that in the weak run through that in
the medium run during period 1 (Figure 12). This reduction is similar to the reduction in
the averaged LWC from that in the low-aerosol run to that in the weak-low run through
that in the medium-low run during period 1 (Figure 12). Considering this, the fact that the





averaged CDNC is much higher in the control run than in the low-aerosol run leads to a
situation where $(\frac{LWC}{CDNC})$, which is the base of $(\frac{LWC}{CDNC})^{\frac{1}{3}}$, and thus $R_v$ are much smaller in the
control run than in the low-aerosol run for strong fire intensity; the fact that the averaged
CDNC is much higher in the medium run than in the medium-low run leads to a situation
where $(\frac{LWC}{CDNC})$ and $R_v$ are much smaller in the medium run than in the medium-low run for
medium fire intensity; the fact that the averaged CDNC is much higher in the weak run
than in the weak-low run leads to a situation where $(\frac{LWC}{CDNC})$ and $R_v$ are much smaller in the
weak run than in the weak-low run for weak fire intensity.
Using the averaged LWC and the averaged CDNC that are shown in Figure 12,
$\left(\frac{LWC}{CDNC}\right)$ is calculated. $(\frac{LWC}{CDNC})$ reduces from $4.27\times10^{-14}$ kg in the control run for strong fire
intensity to $8.08\times10^{-15}$ kg in the weak run for weak fire intensity through $3.09\times10^{-14}$ kg in
the medium run for medium fire intensity. $(\frac{LWC}{CDNC})$ reduces from $1.1\times10^{-12}$ kg in the low-
aerosol run for strong fire intensity to $8.1\times10^{-13}$ kg in the weak-low run for weak fire
intensity through $1.00\times10^{-12}$ kg in the medium-low run for medium fire intensity. Here, the
absolute variation of $(\frac{LWC}{CDNC})$ with varying fire intensity is about one order of magnitude
smaller among the control run, the medium run and the weak run than among the low-
aerosol run, the medium-low run and the weak-low run during period 1. Despite this, the
absolute and percentage variations of $(\frac{LWC}{CDNC})^{\frac{1}{3}}$ and thus the averaged $R_v$ is greater among
the control run, the medium run and the weak run than among the low-aerosol run, the
medium-low run and the weak-low run during period 1. This is due to the characteristics
of a function of the form " $x^{\frac{1}{3}}$ " whose reduction with reducing x is greater when x is
positive and smaller for an identical decrement in x. $(\frac{LWC}{CDNC})^{\frac{1}{3}}$ varies by $1.50\times10^{-5}$ kg from
$3.50\times10^{-5}$ kg in the control run for strong fire intensity to $2.00\times10^{-5}$ kg in the weak run for
weak fire intensity, while it varies by $9.80\times10^{-6}$ kg from $1.03\times10^{-4}$ kg in the low-aerosol
run for strong fire intensity to $9.32\times10^{-5}$ kg in the weak-low run for weak fire intensity.
Associated with this, the averaged $R_v$, as seen in Figure 12, shows a 47 % reduction from
3.20 μm in the control run for strong fire intensity to 1.70 μm in the weak run for weak


intensity, and  the averaged $R_v$ shows a 10 % reduction from 7.75  μm in the low-aerosol
run for strong intensity to 6.98 μm in the weak-low run for weak intensity during period 1.

Figure 13 plots the function  $y = x^{\frac{1}{3}}$ to demonstrate the behavior of $R_v$ (y) with respect

to $\frac{LWC}{CDNC}$ (x). In Figure 13 the arbitrary values $x_{L1}$ and $x_{L2}$ are large, and $x_{S1}$ and $x_{S2}$ are small.
In Figure 13, a situation is assumed where the variation of x-value from $x_{L1}$ to $x_{L2}$ is greater
than that from $x_{S1}$ and $x_{S2}$.  The variation between $x_{S1}$ and $x_{S2}$ emulates the variation of
$(\frac{LWC}{CDNC})$  among the control run, the medium run and the weak run with higher aerosol
concentrations over the fire spot, while the variation between $x_{L1}$ and $x_{L2}$ emulates the
variation of  $(\frac{LWC}{CDNC})$  among the low-aerosol run, the medium-low run and the weak-low
run with lower aerosol concentrations over the fire spot. Here, x and y correspond to $(\frac{LWC}{CDNC})$
and $(\frac{LWC}{CDNC})^{\frac{1}{3}}$, respectively. Despite the smaller variation between $x_{S1}$ and $x_{S2}$ than that
between $x_{L1}$ and $x_{L2}$, the corresponding variation of y is greater between $x_{S1}$ and $x_{S2}$ than
between $x_{L1}$ and $x_{L2}$. This graphically explains the larger variation of $(\frac{LWC}{CDNC})^{\frac{1}{3}}$ and thus the
averaged $R_v$ in Figure 12 despite the smaller variation of $(\frac{LWC}{CDNC})$ among the control run, the
medium run and the weak run than among the low-aerosol run, the medium-low run and
the weak-low run.

**b.  Equilibrium supersaturation**


During period 1 between 17:00 and 19:00 GMT on August $5^{th}$, the lower value of the
equilibrium supersaturation in rising air parcels in the control run than in the low-aerosol
run for strong fire intensity, in the medium run than in the medium-low run for medium
fire intensity, and in the weak run than in the weak-low run for weak fire intensity,
respectively, aids  the greater reduction in $R_v$ among the control run, the medium run and
the weak run than among the low-aerosol run, the medium-low run and the weak-low run
with weakening fire intensity as seen in Figure 12. Remember that the control run, the
medium run and the weak run have higher aerosol concentrations than the low-aerosol run,
the medium-low run and the weak-low run, respectively, over the fire spot (Table 1). Lee



et al. (2009) and Lee and Penner (2010) have also shown the lower equilibrium
supersaturation in simulations with higher aerosol concentrations than simulations with
lower aerosol concentrations. The rate of parcel supersaturation change is expressed by the
following equation as shown in Khain et al. (2000):

$\frac{dS}{dt} = A_1 W - A_2 \frac{dq_L}{dt}$ (2)

In Equation (2), $S$ is parcel supersaturation, while $W$ and $q_L$ are the updraft speed and the
mixing ratio of cloud-liquid particles (or droplets) in a rising air parcel, respectively. The
coefficients $A_1$ and $A_2$ are functions of thermodynamic parameters such as temperature,
latent heat of vaporization, air viscosity and pressure.  The first term in Equation (2)
represents the generation of supersaturation by adiabatic air cooling, while the second term
represents the depletion of supersaturation by the consumption of water vapor via the
condensational growth of unactivated aerosol particles. At first, supersaturation in a rising
air parcel increases with time as the parcel rises from around the surface. However,
eventually,  as the parcel goes up further, the increasing trend of supersaturation stops and
its decreasing trend starts in the rising air parcel, which is when $\frac{dS}{dt}$ becomes zero and
supersaturation reaches its maximum value or becomes equilibrium supersaturation
(Rogers and Yau, 1991). Rogers and Yau (1991) have shown that a higher aerosol
concentration and associated higher consumption of water vapor by aerosol particles before
supersaturation reaches its equilibrium value induce lower equilibrium supersaturation for
a given set of updraft speed, aerosol composition, a form of aerosol size distribution and
thermodynamic condition in a rising air parcel by using an idealized conceptual model.
This is consistent with Lee et al. (2009) and Lee and Penner (2010) who used a large-eddy
simulation model for a real case. Rogers and Yau (1991) have also shown that a lower
updraft speed and associated less adiabatic cooling induce lower equilibrium
supersaturation for a given set of aerosol concentration, aerosol composition and a form of
aerosol size distribution. According to well-known Köhler theory, smaller aerosol particles
have greater critical supersaturation for a given aerosol chemical composition and aerosol
particles with critical supersaturation, which is lower than the equilibrium supersaturation,



are activated (Rogers and Yau, 1991). Hence, as equilibrium supersaturation lowers in a
rising air parcel, the minimum size of activated aerosol particles increases for a given
aerosol composition as shown in Rogers and Yau (1991). Among aerosol particles that are
activated in the rising air parcel, aerosol particles with the minimum size have the largest
critical supersaturation, hence, all the particles with larger sizes than the minimum size are
activated.

During period 1, as fire intensity weakens and updraft speed decreases for the identical

initial aerosol concentration, the identical aerosol composition, and the identical assumed
form of initial aerosol size distribution, parcel equilibrium supersaturation lowers not only
among the low-aerosol run for strong fire intensity, the medium-low run for medium fire
intensity and the weak-low run for weak fire intensity but also among the control run for
strong fire intensity, the medium run for medium fire intensity and the weak run for weak
fire intensity. Also, during period 1, percentage differences in updraft speed, which is
averaged over areas with positive updraft speed and period 1, between the control run and
the low-aerosol run for strong fire intensity, between the medium run and the medium-low
run for medium fire intensity, and between the weak run and the weak-low run for weak
fire intensity, respectively, are less than 2% and thus negligibly small. These percentage
differences are calculated by Equation (1). During period 1, differences, which is also
calculated by Equation (1), in thermodynamic parameters (i.e., temperature, latent heat of
vaporization, air viscosity and pressure), which are also averaged over areas with positive
updraft speed and period 1, between the control run and the low-aerosol run, between the
medium run and the medium-low run, and between the weak run and the weak-low run,
respectively, are less than 5% and thus considered negligibly small. However, there are
two orders of magnitude higher aerosol concentration over the fire spot in the control run
than in the low-aerosol run for strong fire intensity, in the medium run than in the medium-
low run for medium fire intensity, and in the weak run than the weak-low run for weak fire
intensity, respectively, at the first time step. This is with the identical aerosol composition
and the identical assumed form of initial aerosol size distribution between the control run
and the low-aerosol run, between the medium run and the medium-low run, and between
the weak run and the weak-low run, respectively. Hence, mostly due to differences in
aerosol concentration between the control run and the low-aerosol run, between the


medium run and the medium-low run, and between the weak run and the weak-low run,
respectively, the averaged equilibrium supersaturation and the averaged associated
minimum size of activated aerosol particles over areas with positive updraft speed and
period 1, are lower and higher, respectively, in the control run than in the low-aerosol run
for strong fire intensity, in the medium run than in the medium-low run for medium fire
intensity, and in the weak run than the weak-low run for weak fire intensity. Associated
with this, as diagrammatically depicted in Figure 14, the increase in the averaged minimum
size with weakening fire intensity and associated decreasing updraft speed and equilibrium
supersaturation occurs in the size range that is closer to the right tail of the assumed
unimodal aerosol size distribution among the control run, the medium run and the weak
run  than among the low-aerosol run, the medium-low run and the weak-low run. Figure
14 is the same as Figure 3 but with linear scales for x- and y-axes. In Figure 14, $D_{cs}$ and
$D_{cw}$ represent the averaged minimum size in the low-aerosol run for strong fire intensity
and in the weak-low run for weak fire intensity, respectively, while $D_{ps}$ and $D_{pw}$ represent
the averaged minimum size in the control run for strong fire intensity and in the weak run
for weak fire intensity, respectively. The averaged equilibrium supersaturation reduces
from 0.21% in the control run for strong fire intensity to 0.10% in the weak run for weak
fire intensity. Associated with this, the averaged minimum size in diameter increases from
0.09 μm in the control run for strong fire intensity to 0.12 μm in the weak run for weak fire
intensity over period 1. The averaged equilibrium supersaturation reduces from 0.55% in
the low-aerosol run for strong fire intensity to 0.31% in the weak-low run for weak fire
intensity. Associated with this, the averaged minimum size increases from 0.04 μm in the
low-aerosol run for strong fire intensity to 0.07 μm in the weak-low run for weak fire
intensity over period 1.

The aerosol distribution as depicted in Figure 14 represents the averaged form of the

distribution over the simulation domain and period. This indicates that on average, the
overall initial form of aerosol size distribution is well maintained over the domain and
period, although aerosol concentration in each size bin of the distribution evolves with time
and space. The above-described size range, which is associated with the increase in the
averaged minimum size with weakening fire intensity and decreasing updraft speed, is
between the averaged minimum size with strong fire intensity and that with weak fire





intensity as seen in Figure 14. The concentration of aerosol particles with a size, which is
closer to the right tail, is lower than that with another size, which is less close to the right
tail, as long as these sizes are on the right-hand side of the distribution peak as seen in
Figures 3 and 14; since most of aerosol activation occurs for aerosol sizes on the right-hand
side of the peak, here we are only concerned with the size ranges on the right-hand side.
Stated differently, a larger portion of total aerosol concentration is over a size range that is
farther from the right tail than over the other range which is closer to the right tail, in case
the size increment over the two ranges is similar as can be seen in Figure 14. Note that
associated with similar updraft speeds between the control run and the low-aerosol run for
strong fire intensity, between the medium run and the medium-low run for medium fire
intensity, and between the weak run and the weak-low run for weak fire intensity,
respectively, during period 1, the reduction in updraft speed with weakening fire intensity
among the low-aerosol run, the medium-low run and the weak-low run is also similar to
that among the control run, the medium run and the weak run during period 1. This
contributes to a situation where the increment in the averaged minimum size, which is 0.03
micron, among the low-aerosol run, the medium-low run and the weak-low run is similar
to that among the control run, the medium run and the weak run during period 1 as
diagrammatically depicted in Figure 14. The increment is the averaged minimum size with
weak fire intensity minus that with strong fire intensity.
All aerosol particles with size greater than the minimum-activation size contribute to
the overall CDNC. Accordingly, as seen in Figure 14, the increase in the averaged
minimum size as fire intensity weakens reduces the number of aerosol particles that can be
activated and droplets. This reduction in the number of activated aerosol particles is equal
to the number of aerosol particles in the size range between the averaged minimum size
with strong and weak fire intensity. Figure 14 demonstrates that this increase in the
minimum-activation size with weakening fire intensity occurs closer to the tail among the
control run, the medium run and the weak run than among the low-aerosol run, the medium-
low run and the weak-low run. Recall that a large portion of the total aerosol concentration
is in the size range which is closer to the right tail of the assumed unimodal aerosol size
distribution than that which is less close to the right tail as long as changes in the minimum
size in these two size ranges are similar and these ranges are on the right-hand side of the



aerosol distribution. So, a similar increase in the averaged minimum-activation size for a
weakened fire results in a smaller percentage reduction in the total activated aerosol
concentration among the control run, the medium run and the weak run than among the
low-aerosol run, the medium-low run and the weak-low run during period 1. As seen in
Figure 12, CDNC, which is averaged over cloudy areas and period 1, varies by 8% from
850 cm$^{-3}$ in the control run with strong fire intensity to 780 cm$^{-3}$ in the weak run with weak
fire intensity. The averaged CDNC varies by 76% from 33 cm$^{-3}$ in the low-aerosol run with
strong fire intensity to 8 cm$^{-3}$ in the weak-low run with weak fire intensity. This contributes
to greater reduction in $(\frac{LWC}{CDNC})^{\frac{1}{3}}$ and thus R$_v$ as fire intensity weakens among the control run,
the medium run and the weak run than among the low-aerosol run, the medium-low run
and the weak-low run during period 1. This is for a similar LWC between the control run
and the low-aerosol run for strong fire intensity, between the medium run and the medium-
low run for medium fire intensity, and between the weak run and the weak-low run for
weak fire intensity, respectively.
The contribution is easily understood if we assume a situation where CDNC and
LWC in the control run, the low-aerosol run, the medium run, the medium-low run, the
weak run and the weak-low run are identical to those in Figure 12 except for the fact that
CDNC reduces by 82% from 850 cm$^{-3}$ in the control run with strong fire intensity to 150
cm$^{-3}$ in the weak run with weak fire intensity. In this situation, the CDNC percentage
reduction is greater between the control run and the weak run than between the low-aerosol
run and the weak-low run with weakening fire intensity. This causes a greater reduction in
$(\frac{LWC}{CDNC})^{\frac{1}{3}}$ (and thus R$_v$) between the low-aerosol run and the weak-low run than between the
control run and the weak run with weakening fire intensity. $(\frac{LWC}{CDNC})^{\frac{1}{3}}$ varies by 9.80×10$^{-6}$ kg
from 1.03×10$^{-4}$ kg in the low-aerosol run for strong fire intensity to 9.32×10$^{-5}$ kg in the
weak-low run for weak fire intensity, and it varies by 3.00×10$^{-7}$ kg from 3.50×10$^{-5}$ kg in
the control run for strong fire intensity to 3.47×10$^{-5}$ kg in the weak run for weak fire
intensity. Hence, the smaller percentage variation of CDNC plays a role in the greater
reduction in $(\frac{LWC}{CDNC})^{\frac{1}{3}}$ and thus R$_v$ with weakening fire intensity among the control run, the
medium run and the weak run than among the low-aerosol run, the medium-low run and



the weak-low run during period 1. Remember that the control run, the medium run and the
weak run constitute the polluted-scenario runs with higher aerosol concentrations over the
fire spot, and the low-aerosol run, the medium-low run and the weak-low run constitute the
clean-scenario runs with lower aerosol concentrations over the fire spot. This larger
reduction in $R_v$ for the polluted-scenario case compared with the clean-scenario case—both
with similar LWC and LWC reduction—is also reported in Reid et al. (1999). However,
Reid et al. (1999) did not explain the cause of the greater $R_v$ reduction for the polluted case.
The polluted case has 100 times higher aerosol concentration than the clean case in Reid
et al. (1999), which is similar to the polluted-scenario runs as compared to the clean-
scenario runs in this study.

**4.1.3 Autoconversion, freezing, deposition and condensation**

According to previous studies (e.g., Khairoutdinov and Kogan, 2000; Liu and Daum, 2004;
Lee and Baik, 2017), autoconversion is strongly dependent on the size of cloud droplets
and is proportional to the size of cloud droplets. This is explained by the fact that the
efficiency of collection among droplets is proportional to droplet size (Pruppacher and
Klett, 1978; Rogers and Yau, 1991). Larger droplets collect or coalesce with other droplets
more efficiently. Autoconversion is a collection process among droplets to form raindrops,
which means autoconversion rate and cloud droplet size is proportional. Due to the larger
$R_v$ during period 1, the subsequent autoconversion rates, which are averaged over cloudy
areas and over period 2, are higher in the low-aerosol run than in the control run for strong
fire intensity, in the medium-low run than in the medium run for medium fire intensity, and
in the weak-low run than in the weak run for weak fire intensity, respectively (Figure 15a).
Due to the larger absolute and percentage reduction in $R_v$, as described in Section 4.1.2,
there is a larger absolute and percentage reduction in autoconversion rate among the control
run, the medium run and the weak run than among the low-aerosol run, the medium-low
run and the weak-low run with weakening fire intensity during period 2 (Figure 15a). The
averaged autoconversion rates over period 2 reduce from $3.61\times10^{-6}$ g m$^{-3}$ s$^{-1}$ in the control
run with strong fire intensity to $0.93\times10^{-6}$ g m$^{-3}$ s$^{-1}$ in the weak run with weak fire intensity
through $2.01\times10^{-6}$ g m$^{-3}$ s$^{-1}$ in the medium run with medium fire intensity by 74%. Those



averaged autoconversion rates reduce from $4.52 \times 10^{-6}$ g m$^{-3}$ s$^{-1}$ in the low-aerosol run with
strong fire intensity to $3.94 \times 10^{-6}$ g m$^{-3}$ s$^{-1}$ in the weak-low run with weak fire intensity
through $4.43 \times 10^{-6}$ g m$^{-3}$ s$^{-1}$ in the medium-low run with medium fire intensity by 14%.
Associated with this, differences in the averaged autoconversion rates between the weak
run and the weak-low run for weak fire intensity are greater than those between the control
run and the low-aerosol run for strong fire intensity over period 2; differences in the
averaged autoconversion rates between the medium run and the medium-low run for
medium fire intensity are greater than those between the control run and the low-aerosol
run for strong fire intensity, and smaller than those between the weak run and the weak-
low run for weak fire intensity during period 2 (Figure 15a).

Due to smaller autoconversion rates, there is more cloud liquid available for freezing

in the control run than in the low-aerosol run for strong fire intensity, in the medium run
than in the medium-low run for medium fire intensity, and in the weak run than in the
weak-low run for weak fire intensity, respectively, particularly during period 2. Hence, the
rate of cloud-liquid freezing, which is averaged over cloudy areas and period 2, is greater
in the control run than in the low-aerosol run, in the medium run than in the medium-low
run, and in the weak run than in the weak-low run, respectively (Figure 15a). Remember
that the control run for strong fire intensity, the medium run for medium fire intensity and
the weak run for weak fire intensity constitute the polluted-scenario runs with higher
aerosol concentrations over the fire spot, and the low-aerosol run for strong fire intensity,
the medium-low run for medium fire intensity and the weak-low run for weak fire intensity
constitute the clean-scenario runs with lower aerosol concentrations over the fire spot.
Differences in autoconversion rates between the polluted-scenario run and the clean-
scenario run, which increase with weakening fire intensity, induce those differences in the
amount of cloud liquid available for freezing to get greater with weakening fire intensity
(Figure 15a). Thus, differences in the averaged rate of cloud-liquid freezing between the
polluted-scenario run and the clean-scenario run over period 2 gets greater with weakening
fire intensity (Figure 15a). Due to this, differences in freezing-related latent heat between
the runs increase with weakening fire intensity. When fire intensity is strong, the difference
in freezing-related latent heat, which is averaged over cloudy areas and period 2, between
the polluted-scenario run, which is the control run, and the clean-scenario run, which is the





low-aerosol run, is $1.60 \times 10^{-4}$ J m$^{-3}$ s$^{-1}$. However, with medium fire intensity, that difference
between the polluted-scenario run, which is the medium run, and the clean-scenario run,
which is the medium-low run, is $6.98 \times 10^{-4}$ J m$^{-3}$ s$^{-1}$, while with weak fire intensity, that
difference between the polluted-scenario run, which is the weak run, and the clean-scenario
run, which is the weak-low run, is $7.94 \times 10^{-4}$ J m$^{-3}$ s$^{-1}$. This corresponds to the variation of
the percentage differences, which are calculated by Equation (1), in the averaged freezing-
related latent heat between the polluted-scenario run and the clean-scenario run from 9%
with strong fire intensity to 83% with weak fire intensity through 51% with medium fire
intensity over the period 2.
As shown in Lee et al. (2017), enhanced freezing-related latent heat strengthens
updrafts in places where freezing occurs and this, in turn, enhances deposition and
deposition-related latent heat. Hence, although deposition, which is averaged over cloudy
areas and period 2, is slightly lower, due to those strengthened updrafts, the averaged
deposition and deposition-related latent heat are greater in the control run than in the low-
aerosol run for strong fire intensity, in the medium run than in the medium-low run for
medium fire intensity, and in the weak run than in the weak-low run for weak fire intensity,
respectively, during period 3 (Figures 15a and 15b). As seen in Figure 16, differences in
the averaged freezing rate (and thus the averaged freezing-related latent heating) over
cloudy areas between the control run and the low-aerosol run for strong fire intensity,
between the medium run and the medium-low run for medium fire intensity, and between
the weak run and the weak-low run for weak fire intensity, respectively, do not change
much up to ~20:30 GMT after they start to appear around 18:30 GMT. However, after
~20:30 GMT, these differences start to increase as time goes by for each fire intensity. This
is because as convection intensifies, the transportation of cloud liquid to places above the
freezing level starts to be effective around 20:30 GMT. The greater freezing and thus
freezing-related latent heat in the control run than in the low-aerosol run, in the medium
run than in the medium-low run, and in the weak run than in the weak-low run, respectively,
which start to be significant around 20:30 GMT as compared to those before 20:30 GMT,
invigorates updrafts, which are represented by the averaged updraft mass fluxes over cloud
areas. This subsequently causes updrafts to be stronger in the control run than in the low-
aerosol run for strong fire intensity, in the medium run than in the medium-low run for





medium fire intensity, and in the weak run than in the weak-low run for weak fire intensity,
respectively, from ~21:00 GMT on (Figure 16). Then, the stronger updrafts induce
deposition, which is averaged over cloudy areas, to be greater in the control run than in the
low-aerosol run for strong fire intensity, in the medium run than in the medium-low run for
medium fire intensity, and in the weak run than in the weak-low run for weak fire intensity,
respectively. This is around 10-20 minutes after the stronger updrafts in the control run
than in the low-aerosol run, in the medium run than in the medium-low run, and in the
weak run than in the weak-low run, respectively, start to occur (Figure 16).  Note that
deposition-related latent heat is about one order of magnitude greater than freezing-related
latent heat for a unit of mass of hydrometeors involved in phase-transition processes. This
contributes to much greater differences in deposition-related latent heat during period 3
than those in freezing-related latent heat between the control run and the low-aerosol run
for strong fire intensity, between the medium run and the medium-low run for medium fire
intensity, and between the weak run and the weak-low run for weak fire intensity,
respectively, during periods 2 or 3 (Figures 15a and 15b). To satisfy mass conservation,
the enhanced updrafts above the freezing level, due to enhanced freezing and deposition,
induce more updraft mass fluxes below the freezing level in the control run than in the low-
aerosol run for strong fire intensity, in the medium run than in the medium-low run for
medium fire intensity, and in the weak run than in the weak-low run for weak fire intensity,
respectively. This leads to more convergence around and below cloud base, which is air
flow from environment to cloud, in the control run than in the low-aerosol run for strong
fire intensity, in the medium run than in the medium-low run for medium fire intensity, and
in the weak run than in the weak-low run for weak fire intensity, respectively. The more
mass fluxes and the more convergence below the freezing level, in turn, enhance
condensation. Hence, condensation, which is averaged over cloud areas, starts to be greater
when time reaches ~22:30 GMT in the control run than in the low-aerosol run for strong
fire intensity, in the medium run than in the medium-low run for medium fire intensity, and
in the weak run than in the weak-low run for weak fire intensity, respectively (Figure 16).
This induces the averaged condensation and condensation-related latent heat to be greater
in the control run than in the low-aerosol run for strong fire intensity, in the medium run
than in the medium-low run for medium fire intensity, and in the weak run than in the





weak-low run for weak fire intensity, respectively, during period 4 (Figure 15c). Enhanced
condensation in turn enhances updrafts, establishing a feedback between freezing,
deposition, condensation, and updrafts and thus, enhancing freezing, deposition,
condensation, and updrafts further. This enhancement due to feedback eventually
determines the overall differences in the pyroCb properties and their impacts on the UTLS
water vapor and cloud ice between the control run and the low-aerosol run for strong fire
intensity, between the medium run and the medium-low run for medium fire intensity, and
between the weak run and the weak-low run for weak fire intensity, respectively.

Differences in freezing-related latent heat between the polluted-scenario run and the

clean-scenario run increase with weakening fire intensity, particularly during period 2.
Recall that the control run for strong fire intensity, the medium run for medium fire
intensity and the weak run for weak fire intensity are the polluted-scenario runs, while the
low-aerosol run for strong fire intensity, the medium-low run for medium fire intensity and
the weak-low run for weak fire intensity are the clean-scenario runs. Thus, percentage
differences in freezing-affected updrafts and subsequently in deposition-related latent heat,
which is averaged over cloudy areas and period 3, between the polluted-scenario run and
the clean-scenario run also increase with weakening fire intensity (Figures 15a, 15b and
16). Those differences, as calculated by Equation (1), in deposition-related latent heat are
16%, 181%, and 417 % for strong, medium, and weak fire intensity, respectively, as seen
in Figures 15b and 16. Since percentage increases in deposition-related latent heat in the
polluted-scenario run get greater with weakening fire intensity, the subsequent percentage
increases in updrafts in the polluted-scenario run as compared to updrafts in the clean-
scenario run get greater with weakening fire intensity, particularly during period 3 (Figure
16). During period 4, due to these greater increases in updrafts in the polluted-scenario run
with weaker fire intensity, the percentage increases in condensation in the polluted-
scenario run as compared to condensation in the clean-scenario run get greater with
weakening fire intensity (Figures 15c and 16). Then, the increases in condensation, in turn,
further enhance the increases in updrafts in the polluted-scenario run for each fire intensity.
This enhancement is greater with weaker fire intensity due to the greater increases in
condensation with weaker fire intensity. This leads to the greater overall effects of the
pyroCb on the UTLS water vapor and ice with weaker fire intensity.




**4.2 Dependence of aerosol effects on the magnitude of aerosol perturbation**


Table 3 shows that for each of the strong-, medium-, and weak-fire cases, there are
increases in the UTLS water-vapor mass and in the UTLS amount of cirrus clouds in the
run with the fire-induced aerosol perturbations of 30000 or 7500 cm$^{-3}$. These increases are
relative to the mass and the amount in the low-aerosol run for the strong-fire case, in the
medium-low run for the medium-fire case, and in the weak-low run for the weak-fire case,
respectively, with no fire-induced aerosol perturbation. Note that for each of the three types
of fire-induced aerosol perturbations of 30000, 15000 and 7500 cm$^{-3}$, aerosol-perturbation-
induced percentage increases in the UTLS water-vapor mass and the amount of UTLS
cirrus clouds get greater as fire intensity weakens (Tables 2 and 3). The qualitative nature
of results regarding the dependence of the percentage increases in the UTLS water-vapor
mass and the amount of UTLS cirrus clouds on fire intensity thus does not depend on the
magnitude of the fire-induced aerosol perturbation.
Until now, we considered the situation where the fire-induced aerosol perturbation
does not vary with fire intensity. Note that so far, we have taken interest in the sensitivity
to fire intensity of an aerosol perturbation on pyroCb development, UTLS water vapor,
and cirrus clouds. Hence, to examine and isolate the sensitivity, we have shown
comparisons among sensitivity simulations by varying only the fire intensity while
maintaining a constant aerosol perturbation. While working well for the isolation aspect,
this strategy does not reflect reality well. It may be that weaker fire intensity produces a
smaller aerosol concentration. This possibility is not that unrealistic, since stronger fire
likely involves more material burnt and more aerosols from it.
With this situation in mind, we make comparisons among three pairs of simulations:
the low-aerosol run and the control-30000 run for strong fire vs. the medium-low run and
the medium run for medium fire vs. the weak-low run and the weak-7500 run for weak fire.
Hence, among these three pairs, the magnitude of fire-induced aerosol perturbation reduces
with weakening fire, emulating the possibility that weaker fire intensity involves a less
amount of aerosols. For strong fire, the perturbation-related aerosol concentration is 30000
cm$^{-3}$, for medium fire, it is 15000 cm$^{-3}$, and for weak fire, it is 7500 cm$^{-3}$. As shown in





Tables 2 and 3, comparisons among these three pairs show that relative importance of
aerosol effects on the pyroCb development and its impacts on UTLS water vapor and cirrus
clouds increases for weaker fires, and it does not matter if the aerosol perturbation reduces
or stays constant with weakening fire intensity. In these comparisons, it is also possible
that when fire-induced aerosol perturbation is very low for medium or weak fire intensity,
the latent heat perturbation by aerosol perturbation can be very low. This very low latent
heat is not large enough to increase the relative importance of those aerosol effects with
weakening fire intensity. Based on this, the medium run and the weak run are repeated
again. The medium run is repeated with lower fire-induced aerosol perturbations than the
perturbation of 15000 cm$^{-3}$, while the weak run is repeated with lower fire-induced aerosol
perturbations than the perturbation of 7500 cm$^{-3}$. Recall that when the repeated medium
run has the aerosol perturbation of 2000 cm$^{-3}$, the repeated medium run is referred to as the
medium-2000 run; when the repeated weak run has the aerosol perturbation of 1000 cm$^{-3}$,
the repeated weak run is referred to as the weak-1000 run. The percentage increases in the
UTLS water vapor and the cirrus-cloud amount from the medium-low run to the medium-
2000 run or from the weak-low run to the weak-1000 run are smaller than those increases,
for the case of strong fire, from the low-aerosol run to the control-30000 run. This indicates
that when fire-induced aerosol perturbation reduces too much with weakening fire intensity,
the relative importance of aerosol effects on pyroCb development and its impacts on the
UTLS water vapor and cirrus clouds no longer increases with the weakening fire intensity.

## 5.   Summary and conclusion

This study investigates an observed case of a pyroCb using a modeling framework. In
particular, this study focuses on effects of fire-produced aerosols on pyroCb development
and its impacts on the UTLS water vapor and cirrus clouds. Results show that pyroCb
updrafts transport water vapor to the tropopause and above efficiently. This leads to a much
greater amount of water vapor around and above the tropopause (i.e., the UTLS) over the
pyroCb as compared to that in the background outside the pyroCb. The pyroCb also
generates a deck of cirrus cloud around the tropopause. It is found that the role played by
fire-produced aerosols or the fire-induced aerosol perturbation in the water-vapor





transportation to UTLS and the production of cirrus cloud in the pyroCb gets more
significant as fire intensity weakens.
As fire intensity weakens, due to the reduction in LWC, $R_v$ decreases despite the
reduction in CDNC that tends to increase $R_v$. During the initial stage, there is a similar
LWC between the polluted-scenario run (i.e., the control run for strong fire intensity, the
medium run for medium fire intensity and the weak run for weak fire intensity with the
fire-induced aerosol perturbation) and the clean-scenario run (i.e., the low-aerosol run for
strong fire intensity, the medium-low run for medium fire intensity and the weak-low run
for weak fire intensity with no fire-induced aerosol perturbation) for each fire intensity.
The reduction in LWC with weakening fire intensity among the polluted-scenario runs (i.e.,
the control run, the medium run and the weak run) is also similar to that among the clean-
scenario runs (i.e., the low-aerosol run, the medium-low run and the weak-low run). During
the initial stage, there are much greater CDNC in the polluted-scenario run than in the
clean-scenario run for each fire intensity, and the smaller CDNC reduction among the
polluted-scenario runs than among the clean-scenario runs with weakening fire intensity.
This situation during the initial stage induces $R_v$ to reduce much more among the polluted-
scenario runs than among the clean-scenario runs with weakening fire intensity. This
reduces autoconversion more among the polluted-scenario runs than among the clean-
scenario runs with weakening fire intensity. This makes differences in autoconversion
between the polluted-scenario run and the clean-scenario run enhance as fire intensity
weakens. The enhancing difference in autoconversion between the polluted-scenario run
and the clean-scenario run causes greater differences in freezing-related latent heat as fire
intensity weakens. Through feedback between freezing, deposition, updrafts, and
condensation, differences in freezing-related latent heat induce differences in updrafts
between the polluted-scenario run and the clean-scenario run. Those greater differences in
freezing-related latent heat also lead to greater differences in updrafts, producing the
greater differences in the UTLS water vapor and cirrus clouds between the runs with
weaker fire intensity. This means that the role of fire-produced aerosols in water-vapor
transport to the UTLS and the production of cirrus cloud in the pyroCb becomes more
significant as fire intensity weakens. This role which is more significant with weaker fire
intensity is robust to the magnitude of the given fire-induced aerosol perturbation which



was assumed not to vary with varying fire intensity. This more significant role with weaker
fire intensity is also robust to the variation of the fire-induced aerosol perturbation with the
varying fire intensity unless the variation is very high.

It is true that the level of the understanding of a mechanism that controls the role

played by fire-produced aerosols in the development of pyroCbs and their impacts on water
vapor and cirrus clouds in the UTLS has been low. This study shows that fire-produced
aerosols can invigorate convection and updrafts and thus cause enhanced transportation of
water vapor to the UTLS and enhanced formation of cirrus clouds. This study finds that
the mechanism that controls the invigoration of convection by aerosols in the pyroCb is
consistent with the traditional invigoration mechanism which was proposed and detailed in
Rosenfeld et al. (2008). However, this study shows that for pyroCbs produced by strong
fires, the aerosol-induced invigoration and its effects on water vapor and cirrus clouds in
the UTLS are insignificant. Note that traditional understanding generally focuses on effects
of fire-produced heat and water vapor and their associated fluxes around the surface on the
pyroCb and does not consider effects of fire-produced aerosols on the pyroCb, and this
understanding adequately explains the mechanics for pyroCbs in association with strong
fires. However, this study suggests that the role of fire-produced aerosols in pyroCb
development and its effects on the UTLS water vapor and cirrus clouds should be
considered for cases where pyroCbs form over weak-intensity fires, should one be observed
in nature.

It is of interest to note that when fire-induced aerosol perturbations are strongly

reduced for cases of weaker-intensity fires compared with strong-intensity fires, the
significance of the role played by fire-produced aerosol perturbation does not increase any
longer and starts to reduce with weakening fire. This suggests that there is a critical level
of aerosol perturbation below which the increase in the significance with weakening fire
intensity ceases.









**Author contributions**
SSL came up with the research goals and aims, preformed the simulations, and wrote the
manuscript. GK and ZL selected the case, analyzed observations, and provided data to set
up the simulations while reviewing and providing comments on the manuscript.

**Acknowledgements**
This study is supported by the National Aeronautics and Space Administration (NASA)
supported through grant NNX16AN61G.

























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



**FIGURE CAPTIONS**

Figure 1. VIIRS visible image of fire, smoke and cirrus cloud which are associated with
the selected pyroCb. The bright white represents cirrus cloud or anvil cloud at the top of
the pyroCb, while the red circle marks the fire spot. The dark white represents smoke
produced by fire. Adapted from Kablick et al. (2018).

Figure 2. The simulated fire spot and the field of cloud-ice mass density at the top of the
simulated pyroCb when the pyroCb is about to advance to its mature stage. The red circle
marks the simulated fire spot, while the field represents the simulated cirrus cloud.

Figure 3. Initial aerosol size distribution in the PBL over the fire spot. N represents aerosol
number concentration per unit volume of air and D aerosol diameter.

Figure 4. The vertical distribution of the radar reflectivity which is averaged over the
Cloudsat path.

Figure 5. Vertical distributions of the averaged updraft mass fluxes over cloudy areas,
where the sum of liquid-water content (LWC) and ice-water content (IWC) is non-zero,
and over the simulation period between 17:00 GMT on August 5$^{th}$ and 12:00 GMT on
August 6$^{th}$ in the control run and the low-aerosol run.

Figure 6. Vertical distributions of the averaged water-vapor mass density in the control run
and the low-aerosol run over altitudes above 13 km and over the simulation period between
17:00 GMT on August 5$^{th}$ and 12:00 GMT on August 6$^{th}$. Colored lines represent the
averaged values over cloudy grid columns where there is the non-zero sum of liquid-water
path (LWP) and ice-water path (IWP) in the control run and the low-aerosol run, while the
black line represents those values over non-cloudy columns where there is the zero sum of
LWP and IWP in the control run.

Figure 7.  Vertical distributions of the averaged cloud-ice mass density over cloudy areas,
where the sum of liquid-water content (LWC) and ice-water content (IWC) is non-zero,



and over the simulation period between 17:00 GMT on August 5$^{th}$ and 12:00 GMT on August 6$^{th}$ in the control run and the low-aerosol run.

Figure 8. Vertical distributions of the averaged deposition rate over cloudy areas and over the simulation period between 17:00 GMT on August 5$^{th}$ and 12:00 GMT on August 6$^{th}$ in the control run and the low-aerosol run.

Figure 9. Same as Figure 5 but for all three types of fire intensity.

Figure 10. Same as Figure 6 but for all three types of fire intensity.

Figure 11. Same as Figure 7 but for all three types of fire intensity.

Figure 12. The averaged CDNC, $R_v$, and LWC over cloudy areas at all altitudes and over period between 17:00 and 19:00 GMT on August 5th.

Figure 13. Graphic depiction of y values as a function of " $x^{\frac{1}{3}}$ ". $x_{L1}$ and $x_{L2}$ represent large x values, while $x_{S1}$ and $x_{S2}$ represent small x values. y values corresponding to $x_{L1}$, $x_{L2}$, $x_{S1}$ and $x_{S2}$ are $y_{L1}$, $y_{L2}$, $y_{S1}$ and $y_{S2}$, respectively. The variation of x value from $x_{L1}$ to $x_{L2}$, is greater than that from $x_{S1}$ to $x_{S2}$.

Figure 14. Diagrammatic depiction of the varying minimum size of aerosol activation with varying fire intensity in the unimodal aerosol size distribution which is assumed in this study. The details of the varying minimum size are described in Section 4.1.2. $D_{cs}$ and $D_{cw}$ represent the minimum size in the low-aerosol run for strong fire intensity and in the weak-low run for weak fire intensity, respectively, while $D_{ps}$ and $D_{pw}$ represent the minimum size in the control run for strong fire intensity and in the weak run for weak fire intensity, respectively. Here, the variation of the minimum size from $D_{cs}$ to $D_{cw}$ is identical to that from $D_{ps}$ to $D_{pw}$.



Figure 15. The averaged rates of condensation, deposition and cloud-liquid freezing over
cloudy areas at all altitudes and over (a) periods 2, (b) period 3 and (c) period 4. In panel
(a), the averaged autoconversion rates over cloudy areas at all altitudes and over periods 2
is additionally shown.

Figure 16.  Time series of differences in the averaged values of variables, which are related
to aerosol-induced invigoration of convection, over cloudy areas at all altitudes (a) between
the control run and the low-aerosol run for strong fire intensity, (b) between the medium
run and the medium-low run for medium fire intensity and (c) between the weak run and
the weak-low run for weak fire intensity.























| Simulations | Surface sensible heat fluxes in the fire spot (W m$^{-2}$) | Surface latent heat fluxes in the fire spot (W m$^{-2}$) | Aerosol concentration in the PBL over the fire spot (cm$^{-3}$) |
|---|---|---|---|
| Control run | 15000 | 1800 | 15000 |
| Low-aerosol run | 15000 | 1800 | 150 |
| Control-30000 | 15000 | 1800 | 30000 |
| Control-7500 | 15000 | 1800 | 7500 |
| Medium run | 7500 | 900 | 15000 |
| Medium-low run | 7500 | 900 | 150 |
| Medium-30000 | 7500 | 900 | 30000 |
| Medium-7500 | 7500 | 900 | 7500 |
| Medium-2000 | 7500 | 900 | 2000 |
| Weak run | 3750 | 450 | 15000 |
| Weak-low run | 3750 | 450 | 150 |
| Weak-30000 | 3750 | 450 | 30000 |
| Weak-7500 | 3750 | 450 | 7500 |
| Weak-1000 | 3750 | 450 | 1000 |


Table 1. Summary of simulations
























| | Backg-round | Control | Low-aerosol | Differe-nce (%) | Medium | Meidum-low | Differe-nce (%) | Weak | Weak-low | Differe-nce (%) |
|---|---|---|---|---|---|---|---|---|---|---|
| Updraft mass fluxes (kg m$^{-2}$ s$^{-1}$) | | 1.23 | 1.19 | 3 | 0.89 | 0.70 | 27 | 0.42 | 0.21 | 100 |
| Water-vapor mass density between 13 and 16 km (10$^{-3}$ g m$^{-3}$) | 0.46 | 2.31 | 2.26 | 2 | 1.61 | 1.32 | 22 | 0.93 | 0.58 | 60 |
| Cirrus-cloud mass density between 9 and 13 km (g m$^{-3}$) | | 0.024 | 0.023 | 4 | 0.017 | 0.012 | 42 | 0.008 | 0.004 | 100 |


Table 2. The averaged updraft mass fluxes over cloudy areas at all altitudes, the averaged
water-vapor mass density over altitudes between 13 and 16 km and over cloudy columns
except for the averaged background water-vapor mass density which is also over altitudes
between 13 and 16 km but over non-cloudy columns, and the averaged cirrus-cloud mass
density over cloudy areas between 9 and 13 km. These averaged values are obtained over
the simulation period between 17:00 GMT on August 5[th] and 12:00 GMT on August 6[th].
"Difference" is the percentage difference between the polluted-scenario run and the clean-
scenario run for each fire intensity. Note that the control run for strong fire intensity, the
medium run for medium fire intensity and the weak run for weak fire intensity constitute
the polluted-scenario runs with  higher aerosol concentrations over the fire spot, and the
low-aerosol run for strong fire intensity, the medium-low run for medium fire intensity and
the weak-low run for weak fire intensity constitute the clean-scenario runs with lower
aerosol concentrations over the fire spot. The percentage difference is
$\frac{The\ polluted-scenario\ run\ minus\ the\ clean-scenario\ run}{The\ clean-scenario\ run} \times 100\ (\%)$.







| | Control-30000 | Control-7500 | Medium-30000 | Medium-7500 | Medium-2000 | Weak-30000 | Weak-7500 | Weak-1000 |
|---|---|---|---|---|---|---|---|---|
| Water vapor mass density between 13 and 16 km ($10^{-3}$ g m$^{-3}$) | 2.38 (5%) | 2.28 (0.9%) | 1.87 (42%) | 1.50 (14%) | 1.36 (3%) | 1.31 (125%) | 0.75 (29%) | 0.60 (3%) |
| Cirrus cloud mass density between 9 and 13 km (g m$^{-3}$) | 0.025 (9%) | 0.023 (0.2%) | 0.023 (92%) | 0.014 (17%) | 0.012 (3%) | 0.013 (225%) | 0.006 (50%) | 0.004 (8%) |


Table 3. The averaged water-vapor mass density over cloudy columns between 13 and 16
km and, the averaged cirrus-cloud mass density over cloudy areas between 9 and 13 km.
These averaged values are obtained over the simulation period between 17:00 GMT on
August 5$^{th}$ and 12:00 GMT on August 6$^{th}$. The number in parenthesis is the percentage
difference which is $\frac{The\ control-30000\ (or\ the\ control-7500)\ run\ minus\ the\ low-aerosol\ run}{The\ low-aerosol\ run} \times$
100 (%) for strong fire intensity,
$\frac{The\ medium-30000\ (or\ the\ medium-7500\ or\ the\ medium-2000)\ run\ minus\ the\ medium-low\ run}{The\ medium-low\ run} \times$
100 (%) for medium fire intensity, and
$\frac{The\ weak-30000\ (or\ the\ weak-7500\ or\ the\ weak-1000)\ run\ minus\ the\ weak-low\ run}{The\ weak-low\ run} \times 100$ (%) for
weak fire intensity.






Figure 1





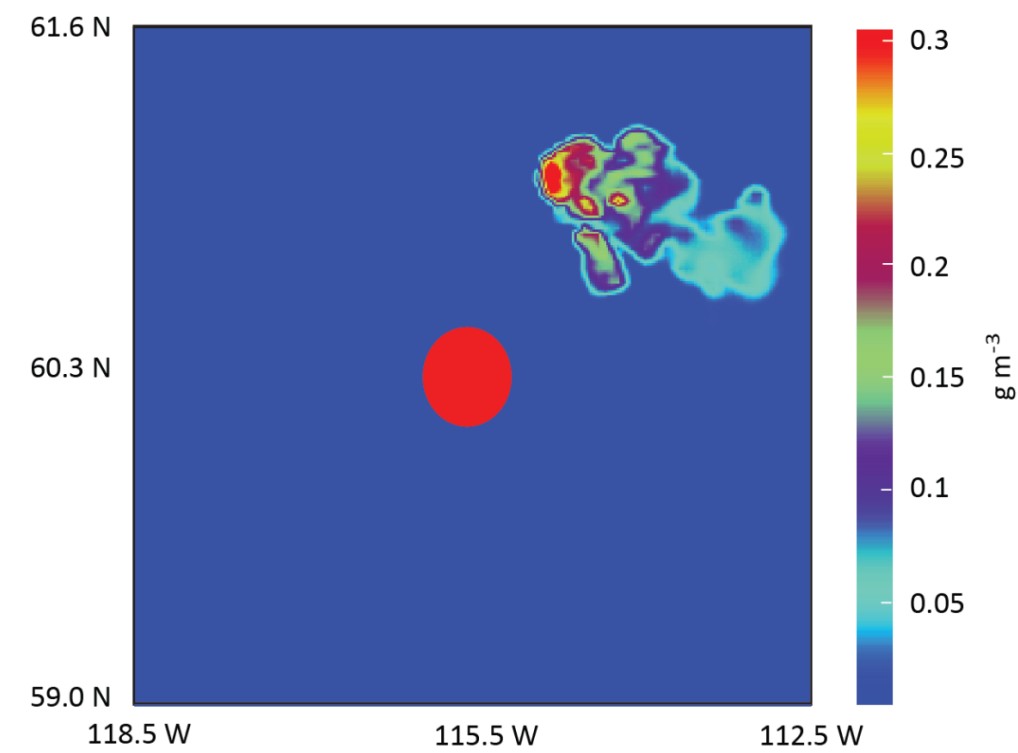



**Figure 2**











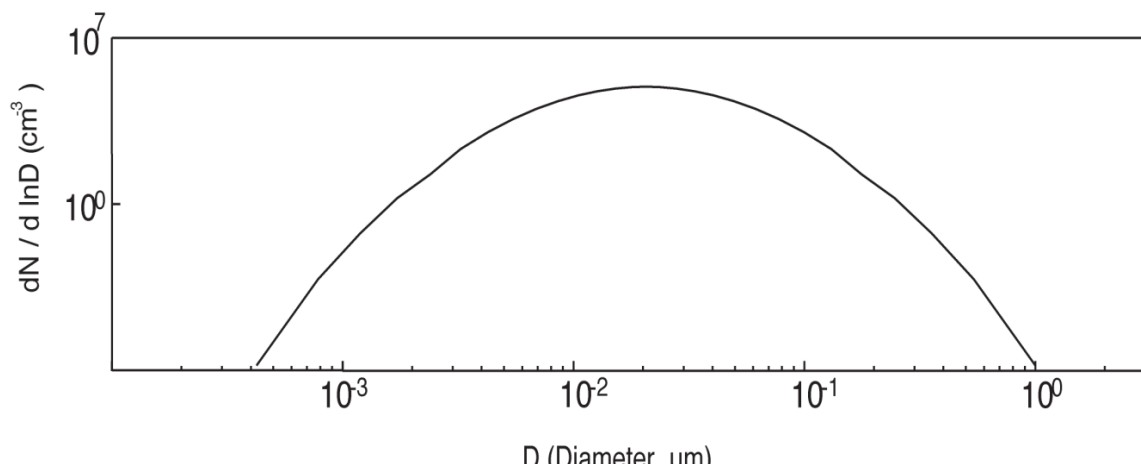


**Figure 3**
















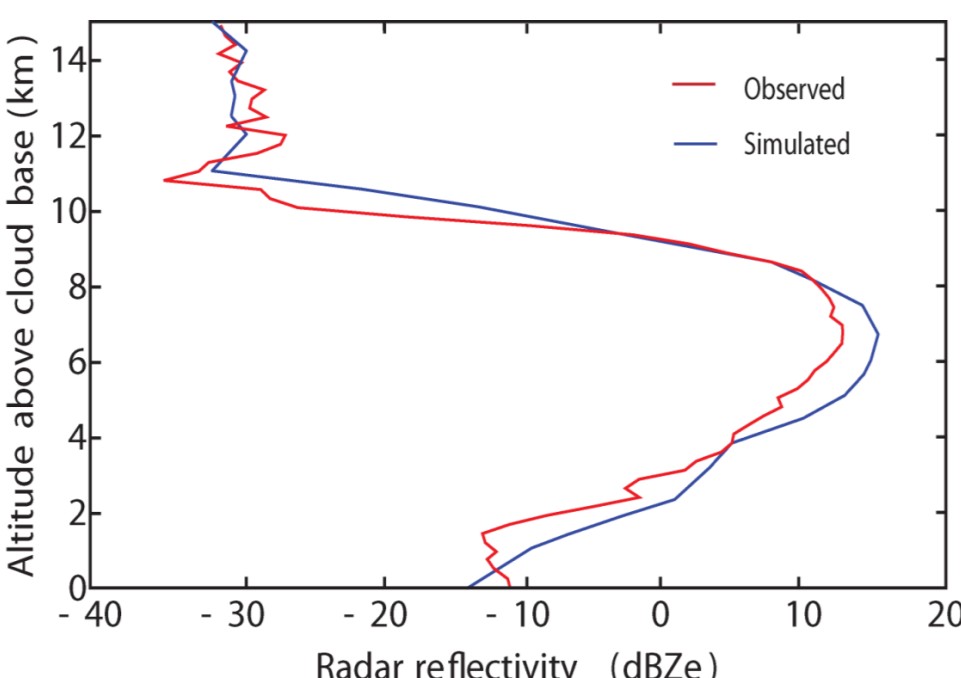


**Figure 4**











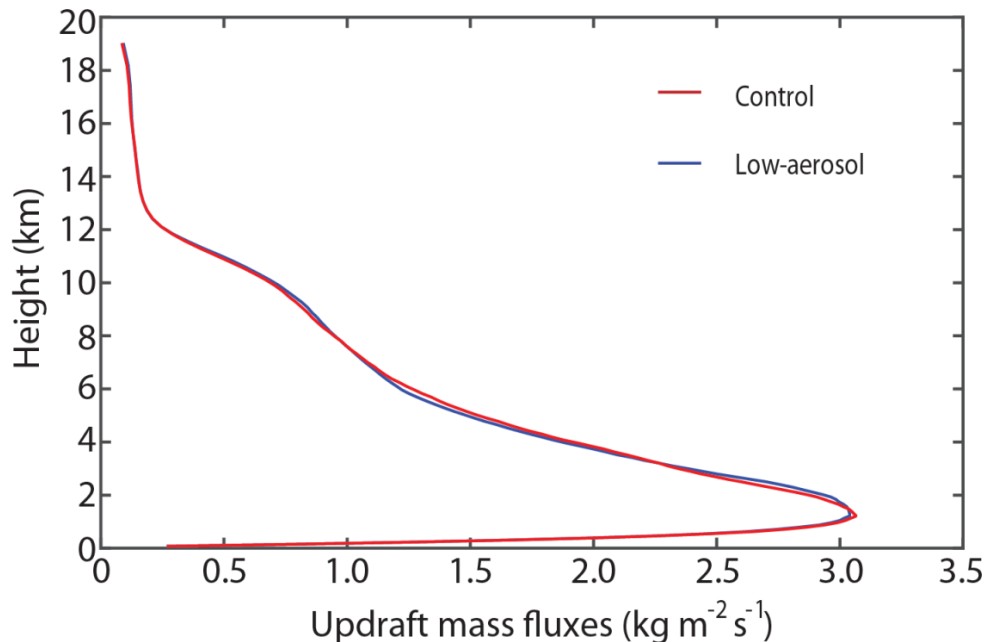


**Figure 5**












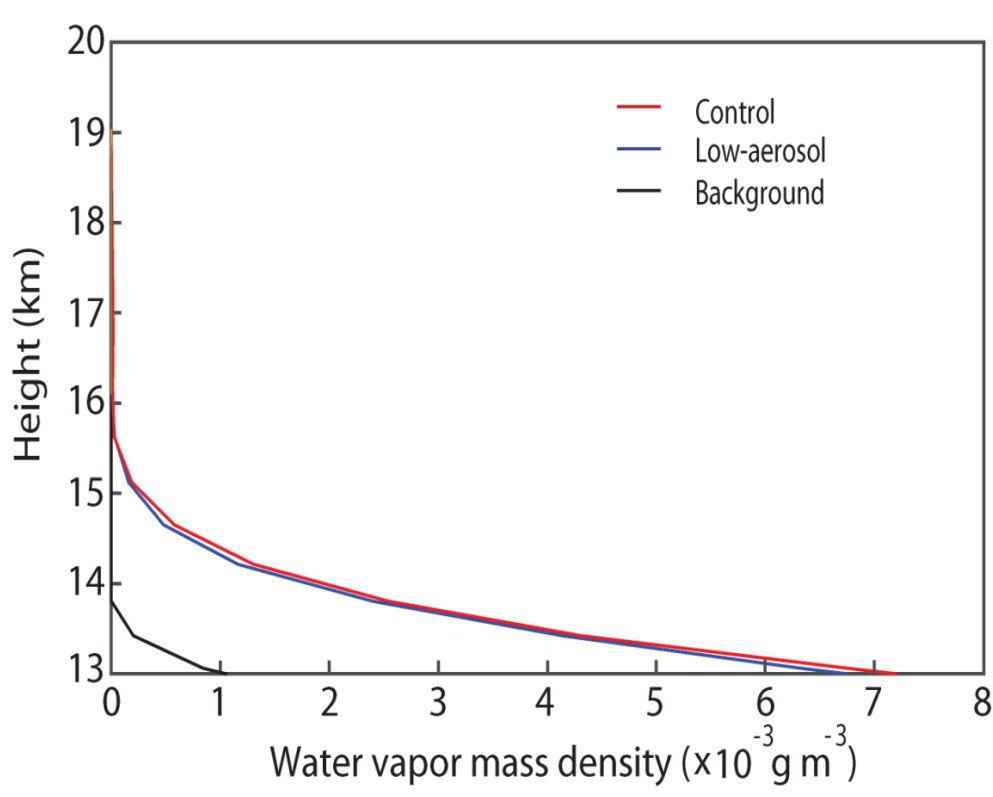


**Figure 6**










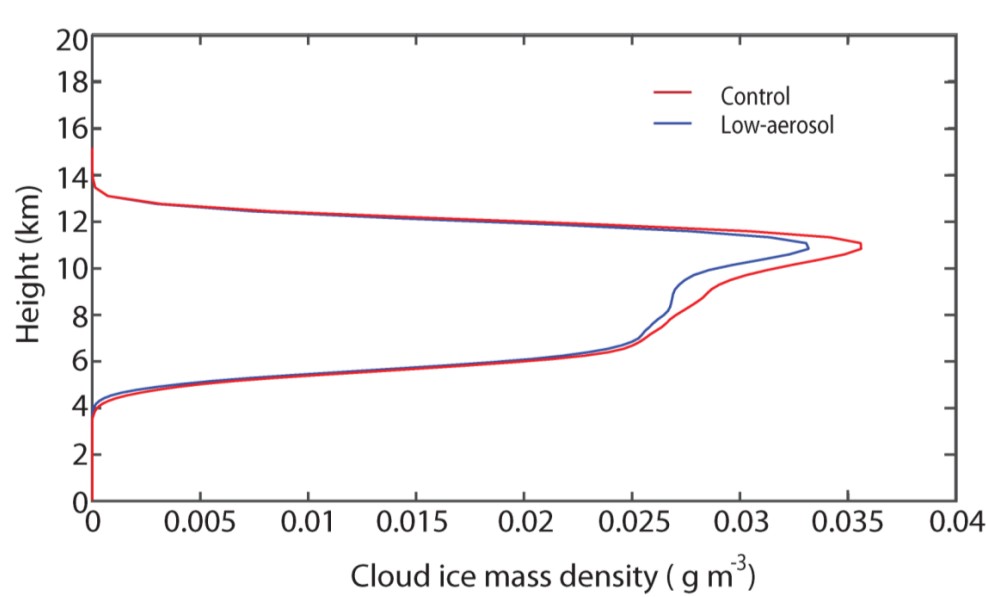


**Figure 7**














**Figure 8**







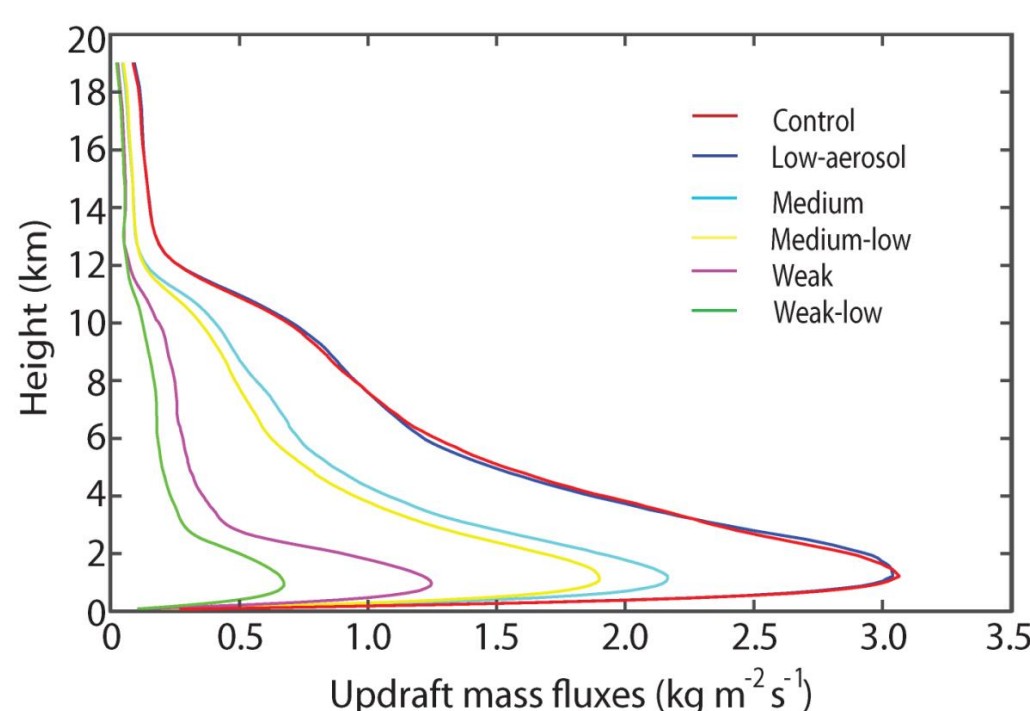


**Figure 9**














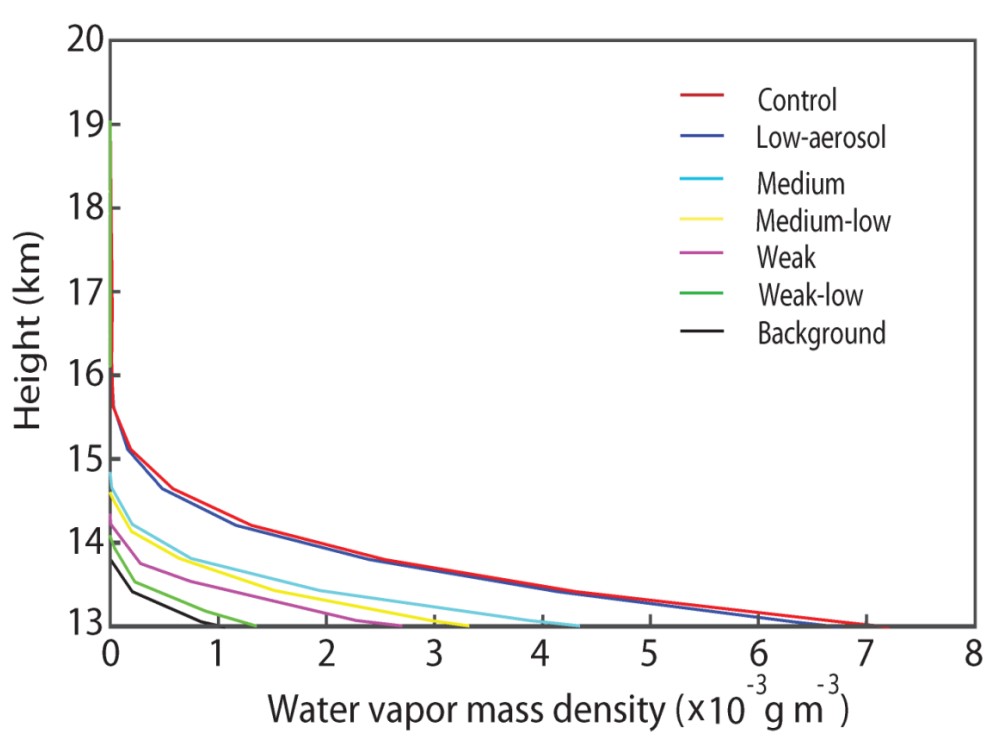


**Figure 10**














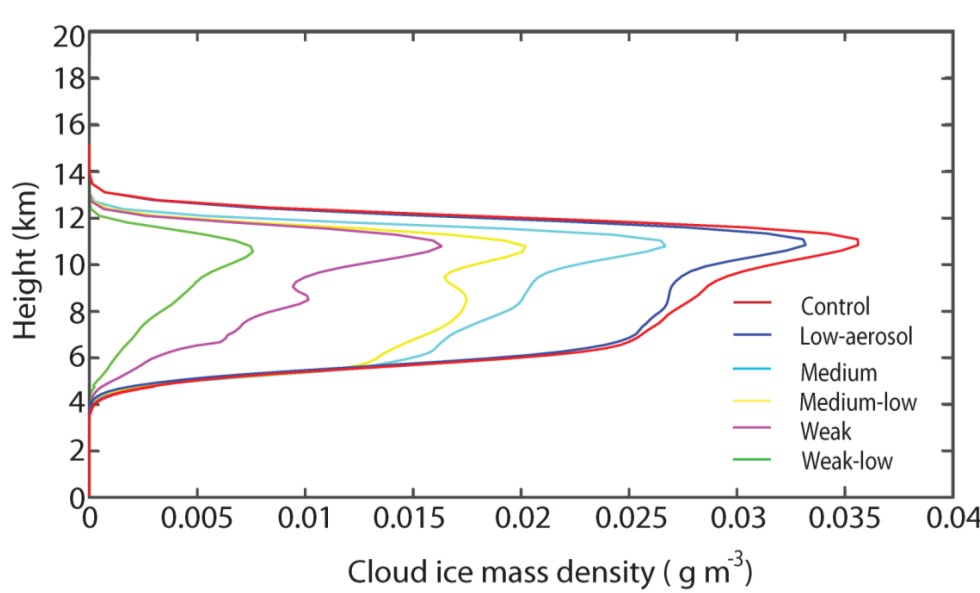


**Figure 11**















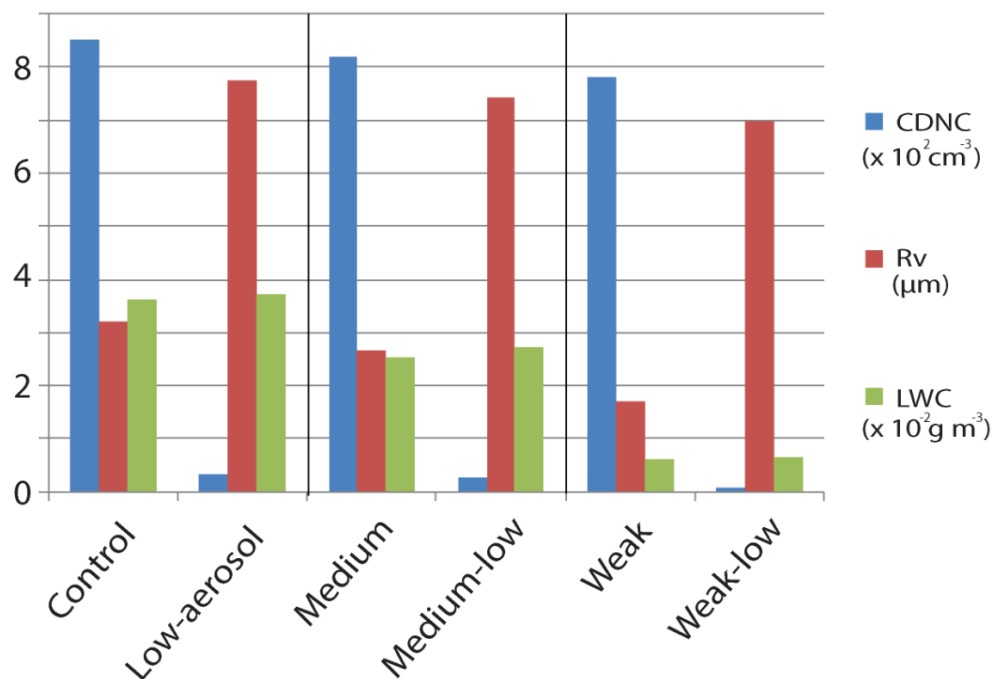


1378                    **Figure 12**












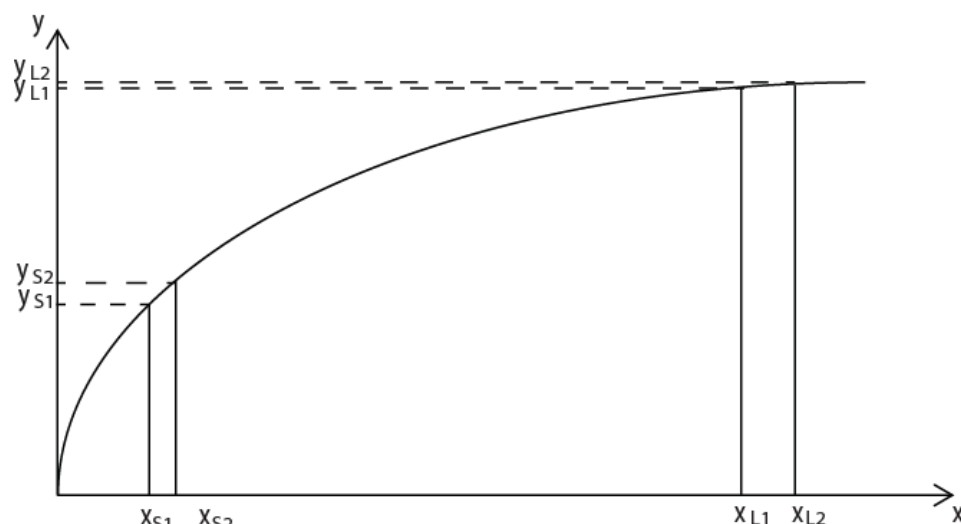


**Figure 13**












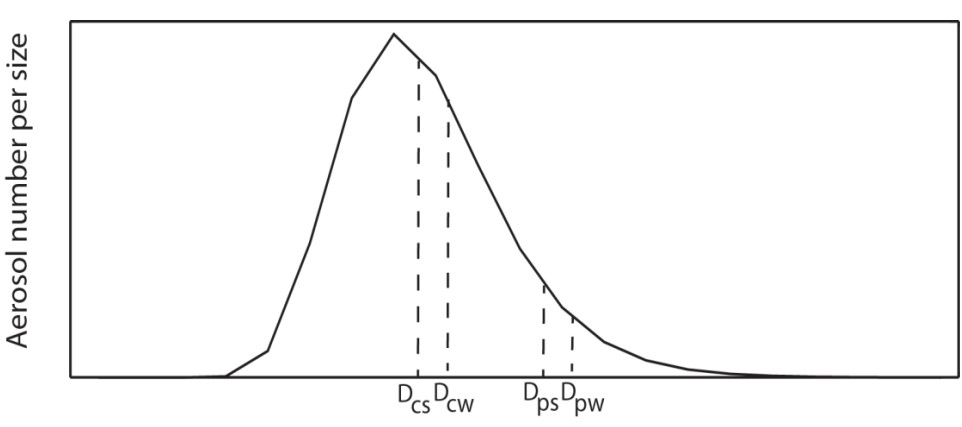


**Figure 14**
















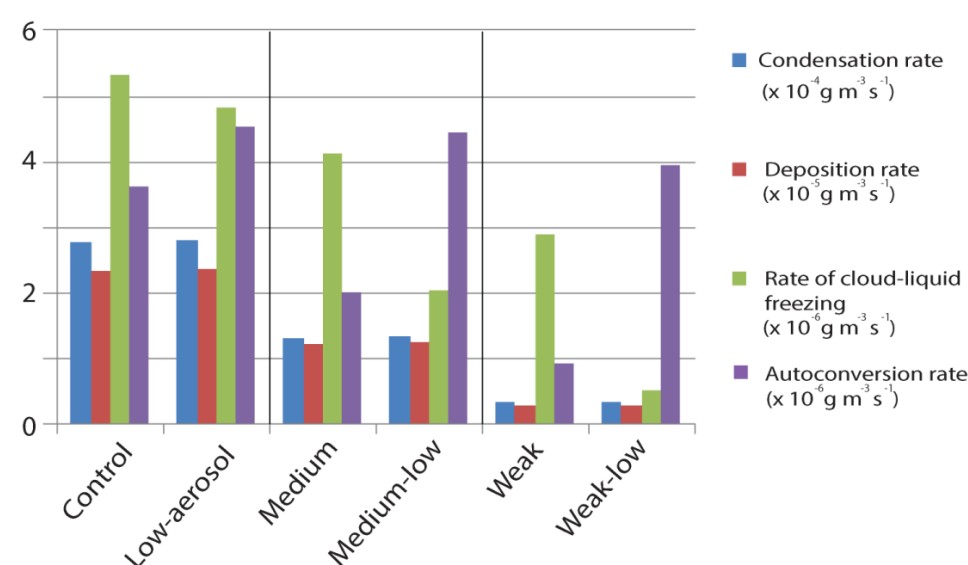

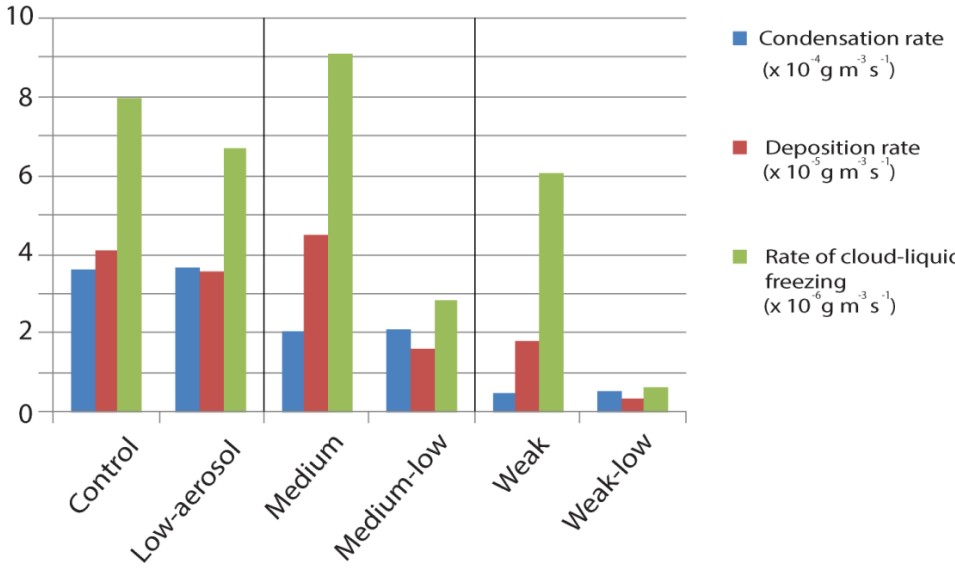


**Figure 15**





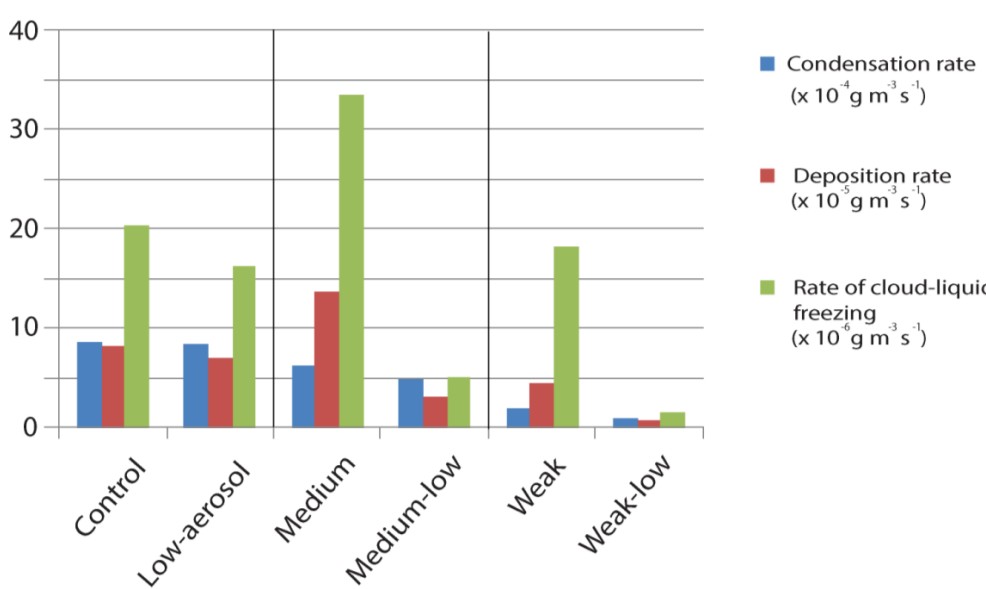


**Figure 15**









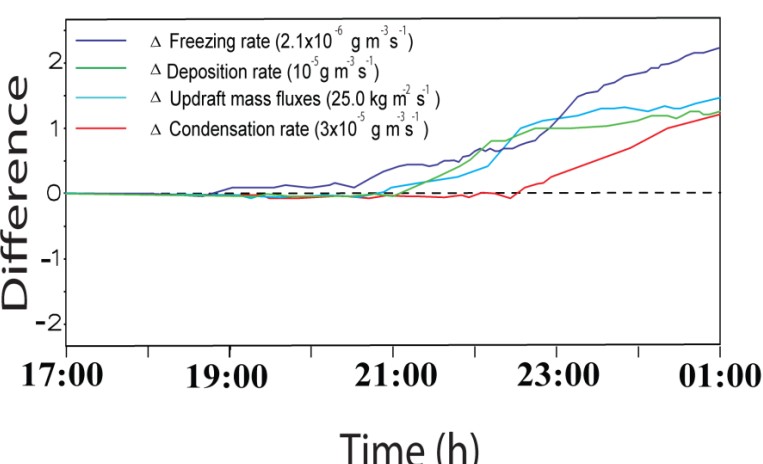

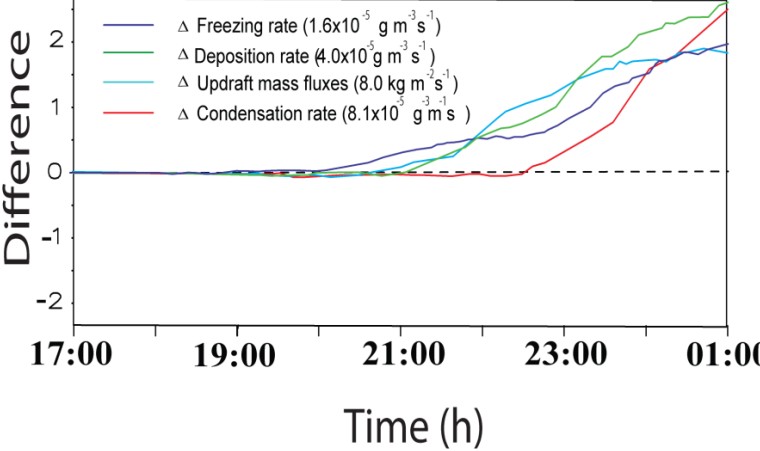


**Figure 16**







**Figure 16**






