# Peer review of "Examination of effects of aerosols on a pyroCb and their dependence on fire intensity and aerosol perturbation using a cloud-system resolving model"

_Atmospheric Chemistry and Physics, 2019_

## Referee Comment (RC1) · Anonymous Referee #1 · 25 Jul 2019

**Referee Report**

Lee et al., Examination of effects of aerosol on a pyroCb and their dependence on fire intensity and aerosol perturbation using a cloud-system resolving model, submitted to Atmos. Chem. Phys., 2019

**General comments**

      The paper addresses the important question of aerosol effects on pyroCb impacts on UTLS water vapor and cirrus cloud area, both of which have substantial, but uncertain, radiative impacts. The authors conclude that microphysical processes (e.g. autoconversion) in pyroCbs associated with weak fires / surface heat fluxes are sensitive to smoke aerosol enhancements, and that such aerosols intensify convection, affecting UTLS water vapor and cirrus. On the other hand, pyroCbs from more intense fires are not very sensitive to smoke effects. Such understanding of UTLS water vapor and cirrus response to smoke is critical to better predicting future atmospheric response to fires, whose frequency and intensity continue to increase.

      The contribution of the paper is that it extends a previous modeling study (Kablick et al., 2018) by testing the effects of a range of aerosol concentrations and surface heat fluxes in a cloud-system resolving model and thus is able to show the variation in sensitivity to aerosols of pyroCb UTLS impacts. The authors find that UTLS and cirrus clouds are increased by the presence of smoke for weak fires but not strong fires, and explain the causal mechanisms. In the future, these sensitivities / processes could be parameterized in climate models.

      The simulation methodology and analysis are rigorous, and the introduction and conclusions are well-organized and logical. However, the results section is very wordy/lengthy and often unclear, so substantial efforts should be made to improve clarity/efficiency of the text, and some figures should be removed or moved to the SI. I recommend publication in ACP after major revisions.

**Specific comments**

*Main Text*

I'm not familiar with the terminology "basic updrafts." Perhaps it should be "maximum" or "typical" "updraft speed"?

Line 80 "makes individual droplet smaller." Please include references.

Line 84 "the efficiency is proportional to the sizes." Please include references.

Line 87 "invigorates updrafts and associated convection." Please include references.

Line 114 Replace "confident information" with "more confident information than from a microphysics-parameterizing model." There are no in-situ observations used in the study for model validation and the conclusions are drawn purely from model representations of microphysical processes. Additionally, the pyroCb case is idealized (no temporal variation in heat flux or interactions with the atmosphere). Therefore, the sensitivity tests provide useful but limited information.

Line 119 After "used by Kablick et al. (2018)," please add a brief explanation of the benefit of using a more sophisticated microphysical scheme.

Line 139 I suggest changing "CSRM" to "Modeling Framework." Remember that ACP has a broader audience than the cloud modeling community.

Lines 197-199 Why were these CCN concentrations chosen? Please include references that they are typical values, e.g. for smoke plumes and background or urban areas.

Line 228 To shorten the paper, I suggest eliminating sentences that describe the contents of the figures, which is already contained in the figure legends, and replacing "in Figure X" etc. with "(Fig. X)" after the result is described. Change "In Figure 1, the observed cirrus cloud at the top of the pyroCb" to "The observed cirrus cloud at the top of the pyroCb (Fig. 1)"

Line 238 Remove sentences on lines 238-240, and add "(Fig. 4)" at the end of line 241 after "cloud reflectivity fields."

Lines 266-271 Condense to "Hence, the variation of fire intensity can be represented by variation of fire-induced surface latent and sensible heat fluxes. As a first step, the control run is repeated by reducing fire-induced surface latent and sensible heat fluxes by factors of 2 and 4. The first repeated run represents a case with medium fire intensity, while the second represents a case with weak fire intensity."

Line 289 Redundant. Alter to: "can also be dependent on the magnitude of fire-induced increases in aerosol concentrations."

Lines 294-302 What real-world situations do aerosol concentrations of 30000, 7500, 2000, and 1000 cm-3 represent? Please provide some context in the text.

Results section: Lengthy and hard to follow. Please make an effort to condense the text, and summarize key results at the end of each section.

Line 307: Change to "Results from the control and low-aerosol runs"

Lines 313-316 Change to "In this study, we expand upon the results of Kablick et al. (2018) by focusing on aerosol effects on pyroCb development and subsequent impacts on UTLS water vapor and cirrus clouds."

Move sentence on lines 327-328 to the beginning of the paragraph starting on line 317.

Lines 330-332 Remove description of Table 2

Lines 332-333 Remove "In Figure 5 and Table 2", and add "(Fig. 5 and Table 2)" at end of sentence

Lines 336-340 Delete first three sentences of this paragraph.

Lines 345-350 Delete last two sentences of this paragraph.

Line 351 In what regions does 16 km correspond to the UTLS? Please provide some context in the text.

Lines 367-369 What is the significance of increasing the vertical extent of water vapor from 14 to 16 km? Please explain in the text.

Lines 370-372 How is UTLS defined here? Please explain in the text.

Line 375 How is tropopause defined here? Please explain in the text.

Lines 379-381 Alter to: "In addition to water vapor in the UTLS, ice crystals comprising cirrus clouds around the tropopause play an important role in the global radiation budget. To identify the impact of the pyroCb on cirrus clouds…"

Line 409 Alter to: "The sensitivity of updrafts, water vapor, and cirrus clouds to aerosol loading in the pyroCb may be affected by fire intensity."

Line 414 Change "In other words" to "Thus".

Line 422 Change "just based on" to "to evaluate the".

Line 426 Change "Updrafts and the UTLS water vapor and cirrus cloud" to "Effects of updrafts on UTLS water vapor and cirrus cloud".

Lines 438-443 Condense/clarify to "Of interest is that the greatest percentage increase in updraft mass flux is in the case of weak fire (weak-low to weak runs), smallest in the case of strong fire (low-aerosol to control runs), and intermediate in the case of medium fire (medium-low to medium runs) (Figure 9 and Table 2)."

Lines 452-453 Alter to: "The percentage difference for medium (weak) fire intensity is obtained by replacing the control run with the medium (weak) run and the low-aerosol run with the medium-low (weak-low) run in Equation (1)."

Lines 454-456 Delete "The percentage difference… Equation (1)."

Lines 456-459 Alter to: "Associated with the greatest (smallest) increases in updraft mass fluxes, the percentage increases in water vapor and cloud-ice mass in the UTLS (Equation 1), are the greatest (smallest) in the case of weak (strong) fire (Figures 10 and 11 and Table 2)."

Lines 459-464 Delete "Figures 10 and 11 … intensity."

Lines 473-476 Alter to: "The simulation period is divided into four sub-periods for this next analysis: period 1 (initial formation of the pyroCb) between 17:00 and 19:00 GMT on August 5th, period 2 between 19:00 and 21:00 GMT on August 5th and period 3 between 21:00 GMT and 23:00 GMT on August 5th (initial stages of cloud development), and period 4 between 23:00 GMT on August 5th and 12:00 GMT on August 6th (mature and decaying stages)."

Lines 476-479 Delete "The initial formation" to "the decaying stages."

Lines 481-485 Combine/condense these two sentences

Line 505 Alter "weakens" to "decreases"

Lines 525-529 Change "reduces" to "decreases"

Lines 545-558 Move this discussion and accompanying figure to SI. Summarize the section.

Lines 598-601 Alter "aerosol chemical composition and aerosol" to "aerosol chemical composition. Aerosol …"

Line 602 Alter to "aerosol composition (Rogers and Yau (1991))"

Line 607-655 This is a very long paragraph. Please divide it into multiple ones.

Lines 663-666 Confusing. Delete "The concentration of … Figures 3 and 14;"

Line 668 Delete "Stated differently"

Lines 676-681 Alter to "This contributes to a situation where the increment (the average minimum size with weak fire intensity minus that with strong fire intensity, 0.03 $\mu$m), among the low-aerosol, medium-low, and weak-low runs is similar to that among the control, medium, and weak runs during period 1 (Figure 14)."

Lines 683-685 Accordingly, the increase in the average minimum size with decreasing fire intensity reduces the number of aerosol particles that can be activated to droplets (Figure 14)."

Lines 698 and 700 Alter "varies" to "decreases"

Lines 741-742 Redundant. Remove "Autoconversion … proportional."

Lines 770-779 Remove "Remember … (Figure 15a)."

Line 848 Alter to "establishing a positive feedback"

Line 876-877 Clarify to state what the effects are

930 Alter to "conclusions"

Lines 942-973 Divide this paragraph in two, one for each conclusion. (1) "This means that the role of fire-produced aerosols in water-vapor transport to the UTLS and the production of cirrus cloud in the pyroCb becomes more significant as fire intensity weakens." (2) "This more significant role with weaker fire intensity is also robust to the variation of the fire-induced aerosol perturbation with the varying fire intensity unless the variation is very high."

*Figures and Tables*
Which figures are essential to the main text and which could be moved to the SI? I suggest moving Figs. 3 and 13 to the SI. I also suggest eliminating Figs. 5-7, since the same information is contained in Figs. 9-11.

Figures w/ multiple lines (4-11): Use multiple line styles (e.g. dashed, dotted, solid). Current lines are difficult to distinguish when printed in grayscale.

Change "over cloudy areas" to "in cloudy areas" and state altitude ranges first, e.g. "at all altitudes in cloudy areas"

Table 2: Remove sentence lines 1255-1260 "Note…"
Put equation in parentheses after "for each fire intensity ((polluted – clean)/clean x 100 (%))"

Table 3: "cirrus-cloud mass density between 9 and 13 km in cloudy areas, over the simulation period between 17:00 GMT on August 5th and 12:00 GMT on August 6th
The numbers in parentheses are the percentage differences: $(control-30000$ $(or$ $control-7500)$ - $low-aerosol)/$ $low-aerosol)$" (etc. for other equations)

Fig. 1: Bright white represents cirrus (anvil) at the top of the pyroCb, while the red circle marks the fire spot. Dark white represents smoke produced by the fire.

Fig. 2: The simulated fire spot (red circle) …
Eliminate "The red circle… "

Fig. 5: "Vertical distributions of the averaged updraft mass fluxes over cloudy areas (where the sum of liquid-water content (LWC) and ice-water content (IWC) is non-zero) over the simulation period between 17:00 GMT on August 5th and 12:00 GMT on August 6th in the control and low-aerosol runs."

Fig. 6: Specify line colors

"Vertical distributions of average water-vapor mass density in the control and low-aerosol runs at altitudes above 13 km and over the simulation period between 17:00 GMT on August 5th and 12:00 GMT on August 6th. Colored lines represent the average values in cloudy grid columns (non-zero sum of liquid-water path (LWP) and ice-water path (IWP)) in the control and low-aerosol runs, while the black line represents those values in non-cloudy columns (zero sum of LWP and IWP) in the control run."

Fig. 7: "Vertical distributions of averaged cloud-ice mass density in cloudy areas (non-zero sum of liquid-water content (LWC) and ice-water content (IWC)) over the simulation period between 17:00 GMT on August 5th and 12:00 GMT on August 6th in the control and low-aerosol runs."

Fig. 8: "Same as Fig. 7, for averaged deposition rate."

Fig. 12: "the averaged CDNC, Rv, and LWC at all altitudes in cloudy areas, over the period between 17:00 and 19:00 GMT on August 5th."

Fig. 14: "Diagrammatic depiction of the varying minimum size of aerosol activation (see Section 4.1.2 for details) with varying fire intensity in the unimodal aerosol size distribution which is assumed in this study."

Eliminate: "The details of the varying minimum size are described in Section 4.1.2"

Eliminate or move to main text: "Here, the variation of the minimum size from Dcs to Dcw is identical to that from Dps to Dpw."

Fig. 15: "The average rates of condensation, deposition and cloud-liquid freezing at all altitudes in cloudy areas and over periods (a) 2, (b) 3 and (c) 4. In panel (a), the average autoconversion rates are additionally shown."

Fig. 16: "Time series of differences in the average values of variables related to aerosol-induced invigoration of convection, at all altitudes in cloudy areas between the (a) control and low-aerosol runs for strong fire intensity, (b) medium and medium-low runs for medium fire intensity and (c) weak and weak-low runs for weak fire intensity."

---

## Referee Comment (RC2) · Anonymous Referee #2 · 17 Aug 2019

Review of: "Examination of effects of aerosol on a pyroCb and their dependence on fire intensity and aerosol perturbation using a cloud-system resolving model"

Authors: Seoung Soo Lee, George Kablick III Zhanqing Li

Recommend major revisions.

General comment:

This manuscript examines the impacts of fire intensity and aerosol concentration on the strength of convection, microphysics processes, and upper level moisture through

simulations using spectral-bin microphysics. I find that much of the paper is not particularly novel since it's fairly well known as this point that aerosol effects tend to be muted with increasing strength of convection. However, details regarding the impacts on microphysics processes are insightful. I found the most novel portion of the paper to be the final result regarding the fact that when a weak fire produces weaker aerosol emissions, the results tend to be muted. I think the paper needs to focus more heavily on the more novel aspects of the work.

In general, the paper needs to be greatly shortened. It is far longer than a typical journal article and needs to be made more concise, particular since it's a follow-on study. There are many places in the paper where the language is too "wordy". Many sentences and statements are written in a way that is difficult to read and can get in the way of representing the scientific results. Examples are given below in the specific comments section of this review.

Specific comments:

Title: The title is too long. You could remove "using a cloud-system resolving model".

1.Lines 63-64: The changes in cirrus clouds altering the radiation budget has become a true but very common motivating factor for cloud microphysics related research as it applies to climate change. Your main motivation here is that pyroCbs with high aerosol loading can change climate. However, pycoCbs are a subcategory of deep convection that comprises a very small percentage of actual deep convective storms and cirrus anvils. As such, I think you need to improve your motivating statements for this work.

2.Line 114: Here you state that using a CSRM allows you to have "confident information" on aerosol effects. I think this assumption is a bit premature given that you have not yet discussed the model you are using or the microphysics parameterization and its capabilities. Some microphysics schemes do not necessarily provide "confident information". Perhaps you should first offer some assessment of your choice of model schemes being used.

3.Introduction: The introduction appears quite short on references. Other work has
been done on pyroCbs and several additional relevant papers should be referenced.

4.Line 160: What is your size distribution of IN? You discuss this here but there's no
other mention of heterogeneous ice nucleation in the paper that I recall seeing.

5.Line 199: At 150/cm3 this seems overly clean for representing a continental type
case. Can you justify your choice here?

6.Section 2: Please provide more description of how aerosols are treated in the model.
Are they transported around the domain after initialization? Is there nucleation scav-
enging and precipitation scavenging? Does the fire continue to act as an aerosol
source after initialization or do the initial aerosols just get depleted? Each of these ef-
fects can impact the interpretation of the results. A fire continually producing aerosols
over time.

7.Section 2: What data was used to initialize and nudge the simulations?

8.Line 235: Your assessment of "good agreement" between the model and satellite
image is based on very little comparison. Can you provide more convincing evidence
that these simulations are well representing this pyroCb event?

9.Lines 341-342: Experience has shown that what you choose as your lower threshold
for averaging cloud water or LWP impacts the interpretation of the results. What do
you consider the lower threshold given that models can provide very small numbers of
LWP that are non-zero? Including tiny values of LWP in an average can impact trends
in results.

10.Lines 386-389: It seems to me that these two sentences about cloud-ice mass
density are just stating that there is cloud-ice in the pyroCb and no cloud-ice outside of
it. It seems unnecessary to state this since non-cloudy areas imply a lack of cloud/ice.
Perhaps you can be clearer on what the intent is for these sentences.

11.Lines 390-391: What about considering homogeneous freezing? This process

should be a major contributor to anvil ice mass and number concentration. I would expect ice numbers to be huge if a large portion of your aerosols are nucleated and transported aloft.

12.Section 4: This section should probably contain some discussion of changes to cirrus cloud ice crystal number concentration. There could be quite an enhancement in cirrus crystal sizes and number which can strongly impact cloud top albedo. Given your motivation factor regarding radiation and climate change, this would seem relevant.

13.Line 439: You state "percentage increase in updrafts"? Are you referring to the number of updrafts or the updraft speed?

14.Lines 462-467: This couple of sentences is an example where the main point could be made more concise.

15.Section 4.1.2: This entire section needs to be re-written or removed. The discussion of (LWC/CNDC) is overtly long and could be greatly condensed. To this same point, figures 13 and 14 are not necessary. The discussion here refers to basic algebra that goes into unnecessary description for the anticipated audience. Further, the section on equilibrium supersaturation is also unnecessary; a very short refresher regarding supersaturation could be useful, but most of the potential readers do not need a full review of this. Referring to Rogers and Yao (1991) is adequate.

16.Lines 743-746: This is an example that shows up numerous times where the lengthy wording interferes with reading the paper in a concise manner. Here you state, "are higher in the low-aerosol run than in the control run for strong fire intensity, in the medium-low run than in the medium run for medium fire intensity, and in the weak-low run than in the weak run for weak fire intensity…" This is very cumbersome to read and needs to be written in a concise way. When you see this type of monotonic behavior, you can simply state. This type of writing shows up many times in the paper and this represents just one example. Please examine the full paper for areas that can be written more concisely.

17.Lines 743-760: It seems here that you're stating that larger Rv = more autoconversion in lower aerosol runs, and then in the next sentence it seems to state reduced Rv = less autoconversion in higher aerosol runs. You don't need to state both of these. One of them implies that the other must be true.

18.Line 960: The use of "enhancing difference" seems awkward. Directly state what the difference is (increase? decrease?)

---

## Author Comment (AC1) · 20 Sep 2019

First of all, we appreciate the reviewer's comments and suggestions. In response to the reviewer's comments, we have made relevant revisions to the manuscript. Listed below are our answers and the changes made to the manuscript according to the questions and suggestions given by the reviewer. Each comment of the reviewer (in black) is listed and followed by our responses (in blue).

**Referee Report**

Lee et al., Examination of effects of aerosol on a pyroCb and their dependence on fire intensity and aerosol perturbation using a cloud-system resolving model, submitted to Atmos. Chem. Phys., 2019

**General comments**

The paper addresses the important question of aerosol effects on pyroCb impacts on UTLS water vapor and cirrus cloud area, both of which have substantial, but uncertain, radiative impacts. The authors conclude that microphysical processes (e.g. autoconversion) in pyroCbs associated with weak fires / surface heat fluxes are sensitive to smoke aerosol enhancements, and that such aerosols intensify convection, affecting UTLS water vapor and cirrus. On the other hand, pyroCbs from more intense fires are not very sensitive to smoke effects. Such understanding of UTLS water vapor and cirrus response to smoke is critical to better predicting future atmospheric response to fires, whose frequency and intensity continue to increase.

The contribution of the paper is that it extends a previous modeling study (Kablick et al., 2018) by testing the effects of a range of aerosol concentrations and surface heat fluxes in a cloud-system resolving model and thus is able to show the variation in sensitivity to aerosols of pyroCb UTLS impacts. The authors find that UTLS and cirrus clouds are increased by the presence of smoke for weak fires but not strong fires, and explain the causal mechanisms. In the future, these sensitivities / processes could be parameterized in climate models.

The simulation methodology and analysis are rigorous, and the introduction and conclusions are well-organized and logical. However, the results section is very wordy/lengthy and often unclear, so substantial efforts should be made to improve clarity/efficiency of the text, and some figures should be removed or moved to the SI. I recommend publication in ACP after major revisions.

Based on the reviewer's comments below and through authors' efforts, the result section is condensed by removing unnecessary text as a way of improving the readability of text. Some figures are removed, following the reviewer's and the other reviewer's comments. For details, see authors' responses below.

**Specific comments**
*Main Text*

I'm not familiar with the terminology "basic updrafts." Perhaps it should be "maximum" or "typical" "updraft speed"?

"Basic" is replaced with "typical". Also, the following is added:

(LL110-111 on p4)

For the simplicity of the term, in this study, "typical updraft speeds" are referred to as "typical updrafts".

Line 80 "makes individual droplet smaller." Please include references.

References are added.

Line 84 "the efficiency is proportional to the sizes." Please include references.

References are added.

Line 87 "invigorates updrafts and associated convection." Please include references.

A reference is added.

Line 114 Replace "confident information" with "more confident information than from a microphysics-parameterizing model." There are no in-situ observations used in the study for model validation and the conclusions are drawn purely from model representations of microphysical processes. Additionally, the pyroCb case is idealized (no temporal variation in heat flux or interactions with the atmosphere). Therefore, the sensitivity tests provide useful but limited information.

The corresponding text is revised as follows:

(LL120-128 on p5)

These simulations are for a case of a pyroCb which is identical to that in Kablick et al. (2018), and performed by using a cloud-system resolving model (CSRM) which is able to resolve cloud-scale dynamic and thermodynamic processes. By resolving these processes that play a critical role in the development of clouds and their interactions with aerosols, we are able to obtain information on aerosol effects on the pyroCb development and its impacts the UTLS water vapor and cirrus clouds, and on associated dynamic and thermodynamic mechanisms. This information is likely to be more confident than that from a model that does not resolve but parameterize those cloud-scale processes.

Line 119 After "used by Kablick et al. (2018)," please add a brief explanation of the benefit of using a more sophisticated microphysical scheme.

The following is added:

(LL131-140 on p5)

Through extensive comparisons between various types of bin schemes and bulk schemes, Fan et al. (2012) and Khain et al. (2015) have concluded that the use of bin schemes is desirable for reasonable simulations of clouds, precipitation, and their interactions with aerosols. This is because the bin scheme explicitly predicts cloud-particle size distributions, while the bulk scheme prescribes those size distributions. The bin scheme also uses collection efficiencies and terminal velocities varying with varying cloud-particle sizes to emulate this variation in reality, while the bulk scheme in general uses fixed efficiencies and terminal velocities, which are not able to consider the variation of collection efficiencies and terminal velocities in reality. This makes the bin scheme more sophisticated than the bulk scheme.

Line 139 I suggest changing "CSRM" to "Modeling Framework." Remember that ACP has a broader audience than the cloud modeling community.

Done.

Lines 197-199 Why were these CCN concentrations chosen? Please include references that they are typical values, e.g. for smoke plumes and background or urban areas.

The following is added:

(LL229-232 on p8)

These prescribed concentrations of aerosols are typically observed in fire spots and their background (Pruppacher and Klett, 1997; Seinfeld and Pandis, 1998; Reid et al., 1999; Andreae et al., 2004; Reid et al., 2005; Luderer et al., 2009).

Line 228 To shorten the paper, I suggest eliminating sentences that describe the contents of the figures, which is already contained in the figure legends, and replacing "in Figure X" etc. with "(Fig. X)" after the result is described. Change "In Figure 1, the observed cirrus cloud at the top of the pyroCb" to "The observed cirrus cloud at the top of the pyroCb (Fig. 1)"

We followed this suggestion except for situations where some description of figures and "in Figure x" are needed for the flow of text.

Line 238 Remove sentences on lines 238-240, and add "(Fig. 4)" at the end of line 241 after "cloud reflectivity fields."

Done.

Lines 266-271 Condense to "Hence, the variation of fire intensity can be represented by variation of fire-induced surface latent and sensible heat fluxes. As a first step, the control run is repeated by reducing fire-induced surface latent and sensible heat fluxes by factors of 2 and 4. The first repeated run represents a case with medium fire intensity, while the second represents a case with weak fire intensity."

Done.

Line 289 Redundant. Alter to: "can also be dependent on the magnitude of fire-induced increases in aerosol concentrations."

Done.

Lines 294-302 What real-world situations do aerosol concentrations of 30000, 7500, 2000, and 1000 cm-3 represent? Please provide some context in the text.

The following is added:

(LL361-366 on p12-13)

The aerosol concentration of 30000 cm$^{-3}$ over the fire spot corresponds to a situation when fire produces a larger concentration of aerosols than a typically observed range between 10000 and 20000 cm$^{-3}$, while the aerosol concentrations of 7500, 2000 and 1000 cm$^{-3}$ over the fire spot corresponds to a situation when fire produces a lower concentration of aerosols than the typically observed range (Reid et al, 1999; Andreae et al, 2004; Reid et al, 2005; Luderer et al., 2009).

Results section: Lengthy and hard to follow. Please make an effort to condense the text, and summarize key results at the end of each section.

Done.

Line 307: Change to "Results from the control and low-aerosol runs"

Section "Results" itself includes sections for all the simulations. So, we created a new subsection 4.1 which is named "the control run and the low-aerosol run"

Lines 313-316 Change to "In this study, we expand upon the results of Kablick et al. (2018) by focusing on aerosol effects on pyroCb development and subsequent impacts on UTLS water vapor and cirrus clouds."

Done.

Move sentence on lines 327-328 to the beginning of the paragraph starting on line 317.

Done.

Lines 330-332 Remove description of Table 2

Done.

Lines 332-333 Remove "In Figure 5 and Table 2", and add "(Fig. 5 and Table 2)" at end of sentence

Done

Lines 336-340 Delete first three sentences of this paragraph.

Done.

Lines 345-350 Delete last two sentences of this paragraph.

Done.

Line 351 In what regions does 16 km correspond to the

UTLS? Please provide some context in the text.

We defined the UTLS as follows:

(LL384-393 on p13)

Regarding the UTLS, in this study, the upper troposphere is defined to be between ~ 9 km in altitude and the tropopause that is ~ 13 km in altitude; the equilibrium level where the buoyancy of a rising air parcel becomes zero above the level of free convection is considered to be the tropopause (Emanuel, 1994). Hence, the defined upper troposphere occupies around a quarter of the total vertical extent of the troposphere. The lower stratosphere is defined to be between the tropopause and an altitude which is 10 km above the tropopause. Hence, the UTLS is between ~9 km and ~23 km in this study. Considering that the stratosphere is between the tropopause and its top that is generally ~ 50 km in altitude, the defined lower stratosphere occupies around a quarter of the total vertical extent of the stratosphere.

Lines 367-369 What is the significance of increasing the vertical extent of water vapor from 14 to 16 km? Please explain in the text.

The following is added:

(LL415-424 on p14)

This means that air parcels that include water vapor and rise from below the tropopause overshoot the tropopause by ~ 3 km in the pyroCb, while those parcels in the background do so by ~ 1 km.  This in turn implies that air parcels and associated updrafts in the pyroCb are stronger to reach higher altitudes before their demise in the stratosphere than those in the background. Those stronger air parcels enable water-vapor layers to be deepened in the lower stratosphere, which in turn enable the interception of longwave radiation by water vapor to occur over longer paths in the lower stratosphere. These longer paths and greater water-vapor mass over the paths both contribute to more interception of longwave radiation by water vapor in the UTLS over the pyroCb than in the background.

Lines 370-372 How is UTLS defined here? Please explain in the text.

The following is added:

(LL384-393 on p13)

Regarding the UTLS, in this study, the upper troposphere is defined to be between ~ 9 km in altitude and the tropopause that is ~ 13 km in altitude; the equilibrium level where the buoyancy of a rising air parcel becomes zero above the level of free convection is considered to be the tropopause (Emanuel, 1994). Hence, the defined upper troposphere occupies around a quarter of the total vertical extent of the troposphere. The lower stratosphere is defined to be between the tropopause and an altitude which is 10 km above the tropopause. Hence, the UTLS is between ~9 km and ~23 km in this study. Considering that the stratosphere is between the tropopause and its top that is generally ~ 50 km in altitude, the defined lower stratosphere occupies around a quarter of the total vertical extent of the stratosphere.

(LL402-403 on p14)

Henceforth, the UTLS water vapor means water vapor in a part of the UTLS at and above the tropopause.

(LL443-444 on p15)

Henceforth, the UTLS cirrus clouds mean those clouds in a part of the UTLS below the tropopause.

Line 375 How is tropopause defined here? Please explain in the text.

The following is added:

(LL385-387 on p13)

the equilibrium level where the buoyancy of a rising air parcel becomes zero above the level of free convection is considered to be the tropopause (Emanuel, 1994).

Lines 379-381 Alter to: "In addition to water vapor in the UTLS, ice crystals comprising cirrus clouds around the tropopause play an important role in the global radiation budget. To identify the impact of the pyroCb on cirrus clouds…"

This sentence is considered redundant and is removed, since this important role by cirrus clouds is already mentioned in introduction.

Line 409 Alter to: "The sensitivity of updrafts, water vapor, and cirrus clouds to aerosol loading in the pyroCb may be affected by fire intensity."

Done.

Line 414 Change "In other words" to "Thus".

Done.

Line 422 Change "just based on" to "to evaluate the".

Done.

Line 426 Change "Updrafts and the UTLS water vapor and cirrus cloud" to "Effects of updrafts on UTLS water vapor and cirrus cloud".

Done.

Lines 438-443 Condense/clarify to "Of interest is that the greatest percentage increase in updraft mass flux is in the case of weak fire (weak-low to weak runs), smallest in the case of strong fire (low-aerosol to control runs), and intermediate in the case of medium fire (medium-low to medium runs) (Figure 9 and Table 2)."

Done.

Lines 452-453 Alter to: "The percentage difference for medium (weak) fire intensity is obtained by replacing the control run with the medium (weak) run and the low-aerosol run with the medium-low (weak-low) run in Equation (1)."

Done.

Lines 454-456 Delete "The percentage difference… Equation (1)."

Done.

Lines 456-459 Alter to: "Associated with the greatest (smallest) increases in updraft mass fluxes, the percentage increases in water vapor and cloud-ice mass in the UTLS (Equation 1), are the greatest (smallest) in the case of weak (strong) fire (Figures 10 and 11 and Table 2)."

We changed text pointed out here as follows:

(LL512-514 on p17)

Associated with the greater increases in updraft mass fluxes, the percentage increases in the UTLS water vapor and cloud-ice mass (Equation 1) are greater in the case of weaker fire (Figures 5 and 6 and Table 2).

Lines 459-464 Delete "Figures 10 and 11 … intensity."

Done.

Lines 473-476 Alter to: "The simulation period is divided into four sub-periods for this next analysis: period 1 (initial formation of the pyroCb) between 17:00 and 19:00 GMT on August 5th, period 2 between 19:00 and 21:00 GMT on August 5$^{th}$ and period 3 between 21:00 GMT and 23:00 GMT on August 5$^{th}$ (initial stages of cloud development), and period 4 between 23:00 GMT on August 5th and 12:00 GMT on August 6$^{th}$ (mature and decaying stages)."

Done.

Lines 476-479 Delete "The initial formation" to "the decaying stages."

Done.

Lines 481-485 Combine/condense these two sentences

Done.

Line 505 Alter "weakens" to "decreases"

A phrase including "weakens" is considered redundant and is removed.

Lines 525-529 Change "reduces" to "decreases"

Done.

Lines 545-558 Move this discussion and accompanying figure to SI. Summarize the section.

The discussion part pointed out here is removed following the other reviewer's comment. This section is summarized at its end.

Lines 598-601 Alter "aerosol chemical composition and aerosol" to "aerosol chemical composition. Aerosol …"

Following the other reviewer's comment, the section including text pointed out here is shortened a lot. Associated with this, text pointed out here is removed.

Line 602 Alter to "aerosol composition (Rogers and Yau (1991))"

Text including this part pointed out here is removed during the process of shortening corresponding section following the other reviewer's comment.

Line 607-655 This is a very long paragraph. Please divide it into multiple ones.

The corresponding paragraph is shortened a lot following this comment here and the other reviewer's comment.

Lines 663-666 Confusing. Delete "The concentration of … Figures 3 and 14;"

Done.

Line 668 Delete "Stated differently"

Done. Also, the following sentence is considered redundant and is removed.

Lines 676-681 Alter to "This contributes to a situation where the increment (the average minimum size with weak fire intensity minus that with strong fire intensity, 0.03    m), among the low-aerosol, medium-low, and weak-low runs is similar to that among the control, medium, and weak runs during period 1 (Figure 14)."

Text pointed out here is removed during the process of shortening the corresponding section, following the other reviewer's comment.

Lines 683-685 Accordingly, the increase in the average minimum size with decreasing fire intensity reduces the number of aerosol particles that can be activated to droplets (Figure 14)."

Text pointed out here is removed during the process of shortening the corresponding section, following the other reviewer's comment.

Lines 698 and 700 Alter "varies" to "decreases"

Done.

Lines 741-742 Redundant. Remove "Autoconversion … proportional."

Done.

Lines 770-779 Remove "Remember … (Figure 15a)."

We removed text between line 770 and 775. However, we did not remove text between line 776 and 779, since information in this text was not given before.

Line 848 Alter to "establishing a positive feedback"

Done.

Line 876-877 Clarify to state what the effects are

We clarified that these effects are about effect of fire-produced aerosols as follows:

(LL717-718 on p24)

This leads to the greater overall effects of fire-produced aerosols on the UTLS water vapor and ice with weaker fire intensity.

930 Alter to "conclusions"

Done.

Lines 942-973 Divide this paragraph in two, one for each conclusion. (1) "This means that the role of fire-produced aerosols in water-vapor transport to the UTLS and the production of cirrus cloud in the pyroCb becomes more significant as fire intensity weakens." (2) "This more significant role with weaker fire intensity is also robust to the variation of the fire-induced aerosol perturbation with the varying fire intensity unless the variation is very high."

Done.

*Figures and Tables*
Which figures are essential to the main text and which could be moved to the SI? I suggest moving Figs. 3 and 13 to the SI. I also suggest eliminating Figs. 5-7, since the same information is contained in Figs. 9-11.

Done. Following the comment by the other reviewer, Figures 3 and 13 are also removed.

Figures w/ multiple lines (4-11): Use multiple line styles (e.g. dashed, dotted, solid). Current lines are difficult to distinguish when printed in grayscale.

There is no additional cost for color printing in ACP. Hence, all the figures will be published in colored form.

Change "over cloudy areas" to "in cloudy areas" and state altitude ranges first, e.g. "at all altitudes in cloudy areas"

Done.

Table 2: Remove sentence lines 1255-1260 "Note…"
Put equation in parentheses after "for each fire intensity ((polluted – clean)/clean x 100 (%))"

Done.

Table 3: "cirrus-cloud mass density between 9 and 13 km in cloudy areas, over the simulation period between 17:00 GMT on August 5th and 12:00 GMT on August 6th
The numbers in parentheses are the percentage differences: ($control$−30000 ($or$ $control$−7500) - $low-aerosol$)/ $low-aerosol$)" (etc. for other equations)

Done.

Fig. 1: Bright white represents cirrus (anvil) at the top of the pyroCb, while the red circle marks the fire spot. Dark white represents smoke produced by the fire.

Done.

Fig. 2: The simulated fire spot (red circle) …
Eliminate "The red circle… "

Done.

Fig. 5: "Vertical distributions of the averaged updraft mass fluxes over cloudy areas (where the sum of liquid-water content (LWC) and ice-water content (IWC) is non-zero) over the simulation period between 17:00 GMT on August 5th and 12:00 GMT on August 6th in the control and low-aerosol runs."

Done.

Fig. 6: Specify line colors

"Vertical distributions of average water-vapor mass density in the control and low-aerosol runs at altitudes above 13 km and over the simulation period between 17:00 GMT on August 5th and 12:00 GMT on August 6th. Colored lines represent the average values in cloudy grid columns (non-zero sum of liquid-water path (LWP) and ice-water path (IWP)) in the control and low-aerosol runs, while the black line represents those values in non-cloudy columns (zero sum of LWP and IWP) in the control run."

Done.

Fig. 7: "Vertical distributions of averaged cloud-ice mass density in cloudy areas (non- zero sum of liquid-water content (LWC) and ice-water content (IWC)) over the simulation period between 17:00 GMT on August 5th and 12:00 GMT on August 6th in the control and low-aerosol runs."

Done.

Fig. 8: "Same as Fig. 7, for averaged deposition rate."

Done.

Fig. 12: "the averaged CDNC, Rv, and LWC at all altitudes in cloudy areas, over the period between 17:00 and 19:00 GMT on August 5th."

Done.

Fig. 14: "Diagrammatic depiction of the varying minimum size of aerosol activation (see Section 4.1.2 for details) with varying fire intensity in the unimodal aerosol size distribution which is assumed in this study."

Figure 14 in the old manuscript is removed by following the comment by the other reviewer.

Eliminate: "The details of the varying minimum size are described in Section 4.1.2"

Figure 14 in the old manuscript is removed by following the comment by the other reviewer.

Eliminate or move to main text: "Here, the variation of the minimum size from Dcs to Dcw is identical to that from Dps to Dpw."

Figure 14 in the old manuscript is removed by following the comment by the other reviewer.

Fig. 15: "The average rates of condensation, deposition and cloud-liquid freezing at all altitudes in cloudy areas and over periods (a) 2, (b) 3 and (c) 4. In panel (a), the average autoconversion rates are additionally shown."

Done.

Fig. 16: "Time series of differences in the average values of variables related to aerosol-induced invigoration of convection, at all altitudes in cloudy areas between the (a) control and low-aerosol runs for strong fire intensity, (b) medium and medium-low runs for medium fire intensity and (c) weak and weak-low runs for weak fire intensity."

Done.

---

## Author Comment (AC2) · 20 Sep 2019

First of all, we appreciate the reviewer's comments and suggestions. In response to the reviewer's comments, we have made relevant revisions to the manuscript. Listed below are our answers and the changes made to the manuscript according to the questions and suggestions given by the reviewer. Each comment of the reviewer (in black) is listed and followed by our responses (in blue).

Review of: "Examination of effects of aerosol on a pyroCb and their dependence on fire intensity and aerosol perturbation using a cloud-system resolving model"

Authors: Seoung Soo Lee, George Kablick III Zhanqing Li

Recommend major revisions.

General comment:

This manuscript examines the impacts of fire intensity and aerosol concentration on the strength of convection, microphysics processes, and upper level moisture through simulations using spectral-bin microphysics. I find that much of the paper is not particularly novel since it's fairly well known as this point that aerosol effects tend to be muted with increasing strength of convection. However, details regarding the impacts on microphysics processes are insightful. I found the most novel portion of the paper to be the final result regarding the fact that when a weak fire produces weaker aerosol emissions, the results tend to be muted. I think the paper needs to focus more heavily on the more novel aspects of the work.

In general, the paper needs to be greatly shortened. It is far longer than a typical journal article and needs to be made more concise, particular since it's a follow-on study. There are many places in the paper where the language is too "wordy". Many sentences and statements are written in a way that is difficult to read and can get in the way of representing the scientific results. Examples are given below in the specific comments section of this review.

We, authors, revised the manuscript based on this comment particularly by focusing on shortening the manuscript. To shorten the manuscript, we removed unnecessary text and figures not only by the following comments below but also by our own decision.

Specific comments:

Title: The title is too long. You could remove "using a cloud-system resolving model".

Done.

1. Lines 63-64: The changes in cirrus clouds altering the radiation budget has become a true but very common motivating factor for cloud microphysics related research as it applies to climate change. Your main motivation here is that pyroCbs with high aerosol loading can change climate. However, pycoCbs are a subcategory of deep convection that comprises a very small percentage of actual deep convective storms and cirrus anvils. As such, I think you need to improve your motivating statements for this work.

The following is added:

(LL65-74 on p3)

The level of our understanding of impacts of pyroCbs on water vapor and cirrus clouds in the UTLS over the global scale is very low and studies to improve this understanding has been going on (Fromm et al., 2010). However, this paper does not focus on these pyroCb impacts at the global scale. Instead, this paper aims to

gain a process-level understanding of mechanisms that control impacts of individual pyroCbs on water vapor and cirrus clouds in the UTLS. The examination of these mechanisms can provide useful information to parameterize interactions among pyroCbs, water vapor and cirrus clouds in climate models. Hence, this examination can contribute to studies that try to improve our understanding of the global-scale impacts of pyroCbs on water vapor and cirrus clouds by using climate models.

2. Line 114: Here you state that using a CSRM allows you to have "confident information" on aerosol effects. I think this assumption is a bit premature given that you have not yet discussed the model you are using or the microphysics parameterization and its capabilities. Some microphysics schemes do not necessarily provide "confident information". Perhaps you should first offer some assessment of your choice of model schemes being used.

Text pointed out is revised as follows:

(LL120-140 on p5)

These simulations are for a case of a pyroCb which is identical to that in Kablick et al. (2018), and performed by using a cloud-system resolving model (CSRM) which is able to resolve cloud-scale dynamic and thermodynamic processes. By resolving these processes that play a critical role in the development of clouds and their interactions with aerosols, we are able to obtain information on aerosol effects on the pyroCb development and its impacts the UTLS water vapor and cirrus clouds, and on associated dynamic and thermodynamic mechanisms. This information is likely to be more confident than that from a model that does not resolve but parameterize those cloud-scale processes. The basic modeling methodology in this study is similar to that used by Kablick et al. (2018). However, this study uses a more sophisticated microphysical scheme, i.e., a bin scheme, rather than the two-moment bulk scheme used by Kablick et al. (2018). Through extensive comparisons between various types of bin schemes and bulk schemes, Fan et al. (2012) and Khain et al. (2015) have concluded that the use of bin schemes is desirable for reasonable simulations of clouds, precipitation, and their interactions with aerosols. This is because the bin scheme explicitly predicts cloud-particle size distributions, while the bulk scheme prescribes those size distributions. The bin scheme also uses collection efficiencies and terminal velocities varying with varying cloud-particle sizes to emulate this variation in reality, while the bulk scheme in general uses fixed efficiencies and terminal velocities, which are not able to consider the variation of collection efficiencies and terminal velocities in reality. This makes the bin scheme more sophisticated than the bulk scheme.

3. Introduction: The introduction appears quite short on references. Other work has been done on pyroCbs and several additional relevant papers should be referenced.

The following references are added:

Fromm, M., D. T. Lindsey, R. Servranckx, G. Yue, T. Trickl, R. Sica, P. Doucet, S. Godin-Beekmann, et al. (2010), The untold story of pyrocumulonimbus, B. Am. Meteorol. Soc., 91 (9), 1193, doi:10.1175/2010BAMS3004.1.

Peterson, D., M. Fromm, J. Solbrig, E. Hyer, M. Surratt, and J. Campbell (2017), Detection and inventory of intense pyroconvection in western North America using GOES-15 daytime infrared data, J. Appl. Meteorol. Clim., 56 (2), 471-493.

Pumphrey, H., M. Santee, N. Livesey, M. Schwartz, and W. Read (2011), Microwave Limb Sounder observations of biomass-burning products from the Australian bush fires of February 2009, Atmos. Chem. Phys. , 11 (13), 6285-6296.

4. Line 160: What is your size distribution of IN? You discuss this here but there's no other mention of heterogeneous ice nucleation in the paper that I recall seeing.

The following is added:

(LL251-258 on p9)

Those studies have indicated that in general, median aerosol diameter and standard deviation of the distribution range from ~0.01 to ~0.03 μm and from ~2.0 to ~2.2, respectively, for aerosols that act as CCN. By taking the approximate median value of each of these ranges, median aerosol diameter and standard deviation of the adopted unimodal distribution of aerosols as CCN are assumed to be 0.02 μm and 2.1, respectively, for the control run. Following Seinfeld and Pandis (1998) and Phillips et al. (2007), for aerosols that act as IN, median aerosol diameter and standard deviation of the unimodal distribution are assumed to be 0.1 μm and 1.6 that are typical values in the continent.

The heterogeneous ice nucleation is calculated by using temperature and supersaturation in addition to IN size distribution. This is reflected as follows:

(LL180-182 on p7)

In these parameterizations, contact, immersion, condensation-freezing, and deposition nucleation paths are all considered by taking into account the size distribution of IN, temperature and supersaturation.

5. Line 199: At 150/cm3 this seems overly clean for representing a continental type case. Can you justify your choice here?

Yes, it is rather low concentration for the continental type case. However, this concentration is frequently observed in remote continental sites including the forest site adopted in this study, according to Pruppacher and Klett (1997), Seinfeld and Pandis (1998), Reid et al. (1999), Andreae et al. (2004), Reid et al. (2005), Luderer et al. (2009). These references are included in text

6. Section 2: Please provide more description of how aerosols are treated in the model. Are they transported around the domain after initialization? Is there nucleation scavenging and precipitation scavenging? Does the fire continue to act as an aerosol source after initialization or do the initial aerosols just get depleted? Each of these effects can impact the interpretation of the results. A fire continually producing aerosols

over time.

Aerosols are diffused and advected by air flow in clouds. After activation or captured by precipitating hydrometeors, aerosols are transported within hydrometeors and removed from the atmosphere once hydrometeors that contain aerosols reach the surface. It is assumed that in non-cloudy areas, aerosol size and spatial distributions are set to follow the background counterparts which are set at the first time step. In other words, once clouds disappear completely at any grid points, aerosol size distribution and number concentration at those points recover to the background counterparts. This assumption has been used by numerous CSRM studies and proven to simulate overall aerosol properties and their impacts on clouds and precipitation reasonably well (Morrison and Grabowski, 2011; Lebo and Morrison, 2014; Lee et al., 2016). This assumption means that a situation where fire continuously produces aerosols to maintain the initial background aerosol concentrations is adopted by this study.

7. Section 2: What data was used to initialize and nudge the simulations?

Balloon soundings of winds, temperature and dew-point temperature were obtained every 6 hours from Ft. Smith observation station, which is located near the forested site, as described in Kablick et al. (2018). The sounding data at 12:00 GMT on August $5_{th}$ are used to prescribe the initial atmospheric condition. Using the sequential soundings, at each altitude, temperature and humidity tendencies are obtained. These tendencies represent the impacts of synoptic- or large-scale motion on temperature and humidity with the assumption that sounding data represent the synoptic conditions, following Grabowski et al. (1996), Krueger et al. (1999) and Lee et al. (2018). These tendencies are horizontally homogeneous and applied to the control run every time step by interpolation.

8. Line 235: Your assessment of "good agreement" between the model and satellite image is based on very little comparison. Can you provide more convincing evidence that these simulations are well representing this pyroCb event?

We made an additional comparison of cloud macro-physical properties, as represented by LWP, IWP, cloud-top and cloud-base heights, between the observation and the control run.

The averaged liquid-water path (LWP) over areas with non-zero LWP in the control run is 960 g $m_{-2}$, while the averaged ice-water path (IWP) over areas with non-zero IWP in the control run is 202 g $m_{-2}$. These simulated LWP and IWP are ~10 % different from the satellite-retrieved counterparts. In this study, for the calculation of LWP (IWP), we only considered droplets (ice crystals); drops with radii smaller (greater) than 20 μm are classified as droplets (raindrops). Stated differently, droplet mass but not rain mass is used to obtain liquid-water content (LWC) and LWP, and the mass of ice crystals but not the mass of snow aggregates, graupel and hail is used to obtain ice-water content (IWC) and IWP. The averaged cloud-top height and cloud-base height over the period between when the pyroCb forms and when the pyroCb disappears is 10.3 km and 3.6 km in the control run, respectively, and these simulated top and base heights are ~7% different from the satellite-retrieved counterparts. This indicates the overall cloud macro-physical structures, as represented by LWP, IWP, cloud-top and cloud-base heights, are simulated reasonably well as compared to the observation.

9. Lines 341-342: Experience has shown that what you choose as your lower threshold for averaging cloud water or LWP impacts the interpretation of the results. What do you consider the lower threshold given that models can provide very small numbers of LWP that are non-zero? Including tiny values of LWP in an average can impact trends in results.

In this study, all the LWP values which are not zero are considered for calculation related to LWP. Stated differently, as long as LWP is greater than 0.00 g $m_{-2}$, LWP is considered for the averaging process. For the calculation of LWP, we need to obtain LWC and it is true that results can be dependent on how to set up the threshold for LWC. Note that when LWC is greater than the threshold value, the LWC is used to calculate LWP and the averaging process. To respond to this comment, we vary the LWC threshold from 0.0 g $m_{-3}$ to 0.01 g $m_{-3}$ through 0.005 g $m_{-3}$; note that

The threshold value of 0.00 g m-3 is adopted by this study, and in addition to this threshold value of 0.00 g m-3, the threshold values of 0.005 g m-3 and 0.01 g m-3 are also frequently adopted by previous studies for LWC and LWP calculations. We find that this variation of the LWC threshold does not change the qualitative nature of results in this study.

10. Lines 386-389: It seems to me that these two sentences about cloud-ice mass density are just stating that there is cloud-ice in the pyroCb and no cloud-ice outside of it. It seems unnecessary to state this since non-cloudy areas imply a lack of cloud/ice. Perhaps you can be clearer on what the intent is for these sentences.

The corresponding sentence is removed and related text is revised as follows:

(LL434-444 on p15)

The altitude of homogeneous freezing is at 9 km , so cirrus clouds which are composed of ice crystals (or cloud ice) only are between 9 km and 13 km. Between 9 km and 13 km, there are the presence of cloud ice and thus cirrus clouds in the control run, meaning that the pyroCb, which is simulated in the control run, produces cirrus clouds (Figure 6). The amount of cirrus clouds in the control run, as represented by the averaged cloud-ice mass density, ranges from 0.028 to 0.037 g m-3 between 9 km and 13 km (Figure 6). The averaged cloud-ice number concentration and cloud-ice size, as represented by its volume mean radius, between 9 km and 13 km ranges from 6 to 20 cm-3, and from 10 to 20 micron, respectively. The altitudes between 9 km and 13 km correspond to a part of the UTLS below the troposphere. Henceforth, the UTLS cirrus clouds mean those clouds in a part of the UTLS below the tropopause.

11. Lines 390-391: What about considering homogeneous freezing? This process should be a major contributor to anvil ice mass and number concentration. I would expect ice numbers to be huge if a large portion of your aerosols are nucleated and transported aloft.

Ice numbers are shown as in our response to comment 12. To indicate the role of homogeneous freezing in ice number, the following is added:

(LL450-454 on p15)

However, mainly due to the larger aerosol concentrations, and associated greater homogeneous aerosol and droplet freezing, there is a large ~20-fold increase in cloud-ice number concentration and associated with this, there is  a large ~2-fold decrease in cloud-ice size in the control run between 9 km and 13 km as compared to that in the low-aerosol run.

12. Section 4: This section should probably contain some discussion of changes to cirrus cloud ice crystal number concentration. There could be quite an enhancement in cirrus crystal sizes and number which can strongly impact cloud top albedo. Given your motivation factor regarding radiation and climate change, this would seem relevant.

The following is added:

(LL440-442 on p15)

The averaged cloud-ice number concentration and cloud-ice size, as represented by its volume mean radius, between 9 km and 13 km ranges from 6 to 20 cm-3, and from 10 to 20 micron, respectively.

(LL450-454 on p15)

However, mainly due to the larger aerosol concentrations, and associated greater homogeneous aerosol and droplet freezing, there is a large ~20-fold increase in cloud-ice number concentration and associated with this, there is  a large ~2-fold decrease in cloud-ice size in the control run between 9 km and 13 km as compared to that in the low-aerosol run.

Line 439: You state "percentage increase in updrafts"? Are you referring to the number of updrafts or the updraft speed?

Here, percentage increase in updrafts means percentage increase in updraft mass fluxes. Since

the updrafts mass flux is "updraft speed" times "air density", and air density at each altitude does vary negligibly among simulations, differences in updraft mass fluxes are mostly explained by those in updraft speed. Hence, percentage increase in updraft mass fluxes means percentage increase in updraft speed with good confidence.

The corresponding text is revised as follows based on the comment here and the other reviewer's comment:

(LL495-498 on p17)

Of interest is that the greatest percentage increase in updraft mass flux is in the case of weak fire (weak-low to weak runs), smallest in the case of strong fire (low-aerosol to control runs), and intermediate in the case of medium fire (medium-low to medium runs) (Figure 4 and Table 2).

The following is added:

(LL498-503 on p17)

Since the updrafts mass flux is updraft speed that is multiplied by air density, and air density at each altitude does vary negligibly among simulations, differences in updraft mass fluxes are mostly explained by those in updraft speed. Hence, it can be said that percentage differences in updraft mass fluxes mean percentage differences in updraft speed with good confidence.

13. Lines 462-467: This couple of sentences is an example where the main point could be made more concise.

Text pointed out here is revised as follows:

(LL512-514 on p17)

Associated with the greater increases in updraft mass fluxes, the percentage increases in the UTLS water vapor and cloud-ice mass (Equation 1) are greater in the case of weaker fire (Figures 5 and 6 and Table 2).

14. Section 4.1.2: This entire section needs to be re-written or removed. The discussion of (LWC/CNDC) is overtly long and could be greatly condensed. To this same point, figures 13 and 14 are not necessary. The discussion here refers to basic algebra that goes into unnecessary description for the anticipated audience.

This section is simplified as seen in the revised manuscript. However, following a comment by the other reviewer, summary is added at the end of this section about LWC, CDNC and $R_v$.

Further, the section on equilibrium supersaturation is also unnecessary; a very short refresher regarding supersaturation could be useful, but most of the potential readers do not need a full review of this. Referring to Rogers and Yao (1991) is adequate.

Following the comment here, the corresponding section is shortened.

15. Lines 743-746: This is an example that shows up numerous times where the lengthy wording interferes with reading the paper in a concise manner. Here you state, "are higher in the low-aerosol run than in the control run for strong fire intensity, in the medium-low run than in the medium run for medium fire intensity, and in the weak-low run than in the weak run for weak fire intensity. . ." This is very cumbersome to read and needs to be written in a concise way. When you see this type of monotonic behavior, you can simply state. This type of writing shows up many times in the paper and this represents just one example. Please examine the full paper for areas that can be written more concisely.

Based on the comment here, we revised the manuscript in a concise way, particularly focusing on expressions similar to those pointed out here.

16. Lines 743-760: It seems here that you're stating that larger Rv = more autoconversion in lower aerosol runs, and then in the next sentence it seems to state reduced Rv = less autoconversion in higher aerosol runs. You don't need to state both of these. One of them implies that the other must be true.

We just want to clarify that in the next sentence which is pointed out by the reviewer here, we do not talk about "reduced Rv= less autoconversion in higher aerosol runs" as phrased by the reviewer here. In the sentence, we talk about the variation of Rv and autoconversion with varying fire intensity. Hence, the two sentences pointed out here deliver two different types of information.

17. Line 960: The use of "enhancing difference" seems awkward. Directly state what the difference is (increase? decrease?)

The enhancing is replaced with "increasing"

---

## Referee Report (RR1)

**Referee Report**

Lee et al., Examination of effects of aerosol on a pyroCb and their dependence on fire intensity and aerosol perturbation using a cloud-system resolving model, submitted to Atmos. Chem. Phys., 2019

**General comments**

In response to comments and requested from both reviewers, the authors shortened the paper and improved the structure. After making the following minor language revisions, many of which will further decrease the length of the text, I recommend the manuscript for publication in ACP. Even after implementing these changes, I strongly recommend that the manuscript is proofread for correct English before publication. Thanks!

**Specific comments**

Please provide examples in the text of "good agreement"

29-30 "specifically focusing on how fire-produced aerosols affect this role via a modeling framework" -> "specifically focusing via a modeling framework on how fire-produced aerosols affect this role"

"studies to improve this understanding has been going on" -> "studies have been conducted to improve this understanding"

Insert "of pyroCB" after "microphysical properties"

"make" -> "decrease"

remove "smaller"

86-87 remove "or increasing aerosol"

88-89 replace "this more competition makes" with "decreasing the size of"

remove "smaller"

change "sizes of droplets" to "droplet sizes"

"autoconversion that is" -> "autoconversion,"

"for them to" -> "in which they", "be" -> "become"

94-95 "More cloud liquid is thus available for transport to places above the freezing level by updrafts." -> "More cloud liquid is thus available for transport by updrafts to altitudes above the freezing level."

"by" -> "of"

"cloud typical properties" -> "cloud properties"

"For the simplicity of the term" -> "For simplicity"

"have shown" -> "showed", "updrafts throughout the manuscript, ex. 115: "the pyroCb development and its impacts on the

UTLS" -> "pyroCb development and its impacts on the UTLS"

"impacts" -> "impacts on"

"parameterize" -> "parameterizes"

remove "have"

"cloud-particle" -> "cloud-particle and aerosol"

"varying with varying" -> "varying with"

137-138 "the bulk schemes in general uses" -> "bulk schemes in general use"

138-139 remove "which are not able to consider the variation of collection efficiencies
and terminal velocities in reality"

139-140 "This makes the bin scheme more sophisticated than the bulk scheme." ->
"Thus, the bin scheme is more sophisticated than the bulk scheme."

"are strongly dependent" -> "is strongly dependent"

"are referred to as fire-driven updrafts, henceforth" -> "are henceforth referred to as
fire-driven updrafts"

153-156 "Aerosol effects on clouds are initiated by an increase in aerosol concentration,
which can be caused by an increase in aerosol emission at and near the surface, and
dependent on how much aerosol concentration increases, or on the magnitude of an
increase in aerosol concentration, i.e., aerosol perturbation" -> "Aerosol effects on
clouds are initiated by an increase in aerosol concentration, which can be caused by an
increase in aerosol emission at and near the surface, and dependent on how much
aerosol concentration increases (aerosol perturbation)"

"has not been" -> "was not"

166-167 "Shortwave and longwave radiation parameterizations have been included in
all simulations by adopting" -> "Shortwave and longwave radiation is parameterized by"

"To represent the microphysical processes" -> "To represent microphysical
processes"

"The cloud-droplet" -> "A cloud-droplet", "parameterization, which is based" -> "parameterization based"

"Arbitrary aerosol mixing states and arbitrary" -> "Arbitrary aerosol mixing states and"

"case is performed" -> "case was performed"

"the site and the pyroCb" -> "the site and pyroCb"

remove "between them"

"from Ft. Smith" -> "from the Ft. Smith"

204-205 "These tendencies are horizontally homogeneous and applied to the control run every time step by interpolation" -> "These tendencies are applied to the control run every time step by interpolation, in a horizontally homogeneous manner"

"lengths" -> "extents"

"For the simulation, the" -> "The simulation"

211-212 "simulation, at the center of the simulation domain, a fire spot with a diameter of 40 km is placed" -> "simulation, a fire spot with a diameter of 40 km is placed at the center of the simulation domain"

"the previous studies which are Trentmann et al. (2006) and Luderer et al. (2006) and adopt boreal forest emissions" -> "previous studies which adopt boreal forest emissions (Trentmann et al. (2006) and Luderer et al. (2006))"

"idealized and this enables" -> "idealized, enabling"

remove "aerosol properties that can be represented by"

235-236 "~50-70% of organic-carbon (OC) compounds, ~5-10% of black-carbon (BC) material, and ~20-45% of inorganic species" -> "~50-70% organic-carbon (OC) compounds, ~5-10% black-carbon (BC) material, and ~20-45% inorganic species"

Insert new paragraph after "in the control run."

247-249 "According to Reid et al. (2005), Knobelspiessel et al. (2011), and Lee et al. (2014), it is reasonable to assume that the initial aerosol size distribution follows the unimodal lognormal distribution in fire sites. Hence, the control run adopts the unimodal lognormal distribution as an initial aerosol size distribution." -> "The control run adopts the unimodal lognormal distribution as an initial aerosol size distribution, a reasonable assumption in fire sites (Reid et al. (2005), Knobelspiessel et al. (2011), Lee et al. (2014))"

Throughout, e.g. 251-253 "median aerosol diameter and standard deviation of the distribution" -> "median and standard deviation aerosol diameter"

Insert new paragraph after "(Pruppacher and Klett, 1978)."

"captured" -> "capture"

"points" -> "point"

267, 270 "counterparts" -> "values"

279-280 "Figure 1. This field in Figure 2 represents" -> "Figure 1, representing"

Throughout, e.g. 284, including figures, "averaged" <cloud property> -> "average" <cloud property>

"well as" -> "well"

Throughout, including figures, anytime multiple rounds are referred to together in a row, e.g. 33,3 "the medium run and the weak run" -> "the medium and weak runs"

remove "or aerosol perturbation"

353, 356 "intensity" -> "intensities"

"corresponds" -> "correspond"

remove "themselves"

"(2018) by" -> "(2018),"

379-380 "The updraft mass flux is one of the most representative variables that are indicative of
the cloud dynamic intensity and the magnitude of convective invigoration." -> ""The updraft mass flux is one of the most indicative variables of cloud dynamic intensity and magnitude of convective invigoration."

"is" -> "was"

382-383 "17:00 GMT on August 5th and 12:00 GMT on August 6th, and 17:00 GMT on August 5th is a time around which the pyroCb starts to from" -> "17:00 GMT on August 5th, approximately when the pyroCb starts to form, and 12:00 GMT on August 6th, and 17:00 GMT on August 5th is"

"that" -> "which"

390-393 "Considering that the stratosphere is between the tropopause and its top that is generally ~ 50 km in altitude, the defined lower stratosphere occupies around a quarter of the total vertical extent of the stratosphere." -> "The defined lower stratosphere occupies around a quarter of the total vertical extent of the stratosphere, the top of which is generally ~ 50 km in altitude."

"the cloudy columns and that over" -> "cloudy and"

"in a part" -> "in the part"

410, 411 remove "those"

remove "or the pyroCb"

remove "both"

"as compared to that in" -> "versus"

430-433 "These averaged fluxes are over cloudy columns for the simulation period between 17:00 GMT on August 5th and 12:00 GMT on August 6th. The averaged water-vapor fluxes vary from 8.30x10-6 kg m-2 s-1 in the control run to 8.21x10-6 kg m-2 s-1 in the low-aerosol run." -> "The averaged water-vapor fluxes over cloudy columns for the simulation period between 17:00 GMT on August 5th and 12:00 GMT on August 6th vary from 8.30x10-6 kg m-2 s-1 in the control run to 8.21x10-6 kg m-2 s-1 in the low-aerosol run."

"only are" -> "are only"

"there are the presence of" -> "there is the presence of"

"cloud-ice number concentration and cloud-ice size" -> "cloud-ice number concentration and size"

"micron" -> $\mu$m

443-444 "The altitudes between 9 km and 13 km correspond to a part of the UTLS below the troposphere. Henceforth, the UTLS cirrus clouds mean those clouds in a part of the UTLS below the tropopause." -> "The altitudes between 9 km and 13 km correspond to a part of the UTLS below the tropopause, and henceforth, "UTLS cirrus clouds" refer to clouds in the troposphere."

445-446 "Updrafts in the pyroCb produce supersaturation, which leads to the generation of cloud ice mass and associated cirrus clouds via deposition, the primary source of cloud-ice mass" -> "Updrafts in the pyroCb produce supersaturation, which leads to the primary source of cloud-ice mass and associated cirrus clouds via deposition"

"circus" -> "cirrus"

452, 453 remove "large"

456-457 "simulations (Figure 7), and thus a negligible variation of the mass of the UTLS cirrus clouds" -> "simulations (Figure 7), and thus a negligible variation of UTLS cirrus cloud mass"

"The role, which is played by" -> "The role of"

464-465 "the UTLS at and above the tropopause, and in the production of the mass of the UTLS cirrus clouds is not significant for strong fire intensity" -> "the UTLS at and above the tropopause, and in the production UTLS cirrus cloud mass, is not significant for strong fire intensity"

"the mass of the UTLS cirrus clouds" -> "UTLS cirrus cloud mass"

"updrafts, produced" -> "updrafts produced"

480-481 "when the fire-generated surface heat fluxes and the fire intensity" -> "when the fire intensity and fire-generated surface heat fluxes"

485-486 "relative magnitude of aerosol-induced perturbation of latent heat to" -> "relative magnitudes of aerosol-induced perturbation of latent heat and"

"does vary" -> "varies"

501-503 "Hence, it can be said that percentage differences in updraft mass fluxes mean percentage differences in updraft speed with good confidence." -> "Hence, it can be said with good confidence that percentage differences in updraft mass fluxes mean percentage differences in updraft speed."

"run than the clean-scenario run" -> "run and clean-scenario runs"

and similar equations: remove "the"

"gets larger" -> "increases"

"than the" -> "than in the"

"the averaged LWC and the averaged CDNC" -> "the average LWC and CDNC"

550, 551 "varies" -> "decreases"

"reduce" -> "decreases"

"to increase" -> "increasing", "lowers" -> "decreases"

"averaged associated" -> "associated average"

"reduces" -> "decreases"

remove "size in"

"reduces" -> "decreases"

"less close to" -> "further from"

588-590 "distribution; most of aerosol activation occurs for aerosol sizes on the right-hand side of the distribution peak, here we are only concerned with the size ranges on the right-hand side." -> "distribution. Here we are only concerned with the size ranges on the right-hand side of the distribution peak, where most of aerosol activation occurs." Throughout the manuscript: <cloud property>", which is averaged" -> <cloud property>"averaged" e.g. 593 "CNDC, which is averaged" -> "CNDC averaged"

"to greater" -> "to a greater"

"thus the" -> "thus a"

"rates, which are averaged" -> "rates averaged"

624-629 "The averaged autoconversion rates over period 2 reduce from 3.61x10-6 g m-3 s-1 in the control run with strong fire intensity to 0.93x10-6 g m-3 s-1 in the weak run with weak fire intensity through 2.01x10-6 g m-3 s-1 in the medium run with medium fire intensity by 74%. Those averaged autoconversion rates reduce from 4.52x10-6 g m-3 s-1 in the low-aerosol run with strong fire intensity to 3.94x10-6 g m-3 s-1 in the weak-low run with weak fire intensity through 4.43x10-6 g m-3 s-1 in the medium-low run with medium fire intensity by 14%." -> "The averaged autoconversion rates over period 2 decrease by 74% from 3.61x10-6 g m-3 s-1 in the control run with strong fire intensity to 0.93x10-6 g m-3 s-1 in the weak run with weak fire intensity through 2.01x10-6 g m-3 s-1 in the medium run with medium fire intensity. Those averaged autoconversion rates decrease by 14%from 4.52x10-6 g m-3 s-1 in the low-aerosol run with strong fire intensity to 3.94x10-6 g m-3 s-1 in the weak-low run with weak fire intensity through 4.43x10-6 g m-3 s-1 in the medium-low run with medium fire intensity."

"get greater" -> "increase"

"smaller" -> "lower"

"freezing, which is averaged" -> "freezing averaged"

640, 642 "get greater" -> "increase"

644-648 "When fire intensity is strong, the difference in freezing-related latent heat, which is averaged in cloudy areas and period 2, between the polluted-scenario run, which is the control run, and the clean-scenario run, which is the low-aerosol run," -> "When fire intensity is strong, the difference in freezing-related latent heat averaged in cloudy areas and period 2 between the polluted-scenario run (control run) and the clean-scenario run (low-aerosol run)"

649-650 "while with weak fire intensity, that difference between the polluted-scenario run, which is the weak run, and the clean-scenario run, which is the weak-low run" -> "while with weak fire intensity, that difference between the polluted-scenario run (weak run) and the clean-scenario run (weak-low run)"

651-652 "differences, which are calculated by Equation (1)," -> "differences (calculated by Equation (1))"

653-654 "and the clean-scenario run from 9% with strong fire intensity to 83% with weak fire intensity through 51% with medium fire intensity over the period 2." -> "and the clean-scenario run over the period 2 from 9% with strong fire intensity to 83% with weak fire intensity to 51% with medium fire intensity."

657, 674 "deposition, which is averaged" -> "deposition averaged"

"period 2," -> "period 2"

"transportation" -> "transport"

remove "as compared to those before 20:30 GMT"

"from" -> "starting at"

"in polluted-scenario" -> "in the polluted-scenario"

"The more mass fluxes and the more convergence" -> "The higher mass fluxes and convergence"

"be greater" -> "increase"

695-696 "thus, enhancing freezing, deposition, condensation, and updrafts further." -> "thus further enhancing freezing, deposition, condensation, and updrafts."

708, 710, 713, 727,741 "get greater" -> "increase"

714-715 "Then, the increases in condensation, in turn, further enhance the increases in updrafts in the polluted-scenario run for each fire intensity." -> "The increases in condensation further enhance the increases in updrafts in the polluted-scenario run for each fire intensity."

"induce" -> "induces"

752-753 "This possibility is not that unrealistic, since stronger fire likely involves more material burnt and more aerosols from it." -> "This possibility is not that unrealistic, since stronger fires likely involve more material burned and higher aerosol emissions."

757, 763, 778 "reduces" -> "decreases"

"less" -> "lower"

759-760 "For strong fire, the perturbation-related aerosol concentration is 30000

cm-3, for medium fire, it is 15000 cm-3, and for weak fire, it is 7500 cm-3." -> "The perturbation-related aerosol concentration is 30000 cm-3 for strong fire, 15000 cm-3 for medium fire, and 7500 cm-3 for weak fire."

"when fire-induced" -> "when the fire-induced"

"has the" -> "has an"

"run; when" -> "run, and when"

"with the weakening" -> "with weakening fire"

"whether those aerosol perturbations vary with varying fire intensity or not" -> "whether or not those aerosol perturbations vary with varying fire intensity"

"transport water vapor to the tropopause and above efficiently" -> "efficiently transport water vapor to the tropopause and above"

"gets more" -> "becomes"

"reduce" -> "decrease"

"makes" -> "results in"

"increase" -> "increasing"

"It is true that the" -> "The"

"transportation" -> "transport"

"reduce" -> "decrease"

"came up with" -> "generated", "preformed" -> "performed"

"revised manuscript based on the reviewers' comments and perform" -> "revised the manuscript based on the reviewers' comments and performed"

Table 2 "Meidum-low" -> "Medium-low"

1172, 1190 "The averaged" -> "Averaged"

Figure 10 "Differences in the averaged values" -> "Differences in average values"

---

## Editor Decision (ED1)

**Examination of effects of aerosols on a pyroCb and their dependence on fire intensity and aerosol perturbation**

Seoung Soo Lee[1], George Kablick III[2,3], Zhanqing Li[2], Chang-Hoon Jung[4], Yong-Sang Choi[5]

[1]Research Foundation, San Jose State University, San Jose, California, USA

[2]Earth System Science Interdisciplinary Center, University of Maryland, College Park, Maryland, USA

[3]US Naval Research Laboratory, Washington, DC, USA

[4]Department of Health Management, Kyungin Women's University, Incheon, South Korea

[5]Department of Environmental Science and Engineering, Ewha Womans University, Seoul, South Korea

**Abstract**

This study investigates how a pyrocumulonimbus (pyroCb) event influences water vapor concentrations and cirrus cloud properties near the tropopause, specifically focusing on how fire-produced aerosols affect this role via a modeling framework. Results from a case study show that when observed fire intensity is high, there is an insignificant impact of fire-produced aerosols on the convective development of the pyroCb and associated changes in water vapor and the amount of cirrus cloud near the tropopause. However, as fire intensity weakens, effects of aerosols on microphysical variables and processes such as droplet size and autoconversion increase. Modeling results shown herein indicate that aerosol-induced invigoration of convection is significant for pyroCb with weak-intensity fires and associated weak surface heat fluxes. Thus, there is a greater aerosol effect on the transportation of water vapor to the upper troposphere and the production of cirrus cloud with weak-intensity fires, whereas these effects are muted with strong-intensity fires.

line 29: this makes it sound like the modeling framework is a physical mechanism. Please reword.

[revised manuscript text omitted]

**Differences in the averaged values**

[Figure]

[Figure]

**Figure 10**

**Figure 10**

---

## Author Response (AR2)

**Editor comment:**

Authors appreciate the editor's comments below that improved the quality of the manuscript a lot. In response to each of the comments, a related line number at which each comment is made in the previous manuscript is first shown in black and then our responses are shown in blue. Line numbers in our responses are based on the revised manuscript.

In addition to revision based on the comments here, we also identified words, sentences and paragraphs that we believed required revision, and revised them to make the manuscript concise.

**29:** Revised also following a reviewer's comment as follows:

(LL27-29 on p2)

Through a modeling framework, this study investigates how a pyrocumulonimbus (pyroCb) event influences water vapor concentrations and cirrus cloud properties near the tropopause, specifically focusing on how fire-produced aerosols affect this role.

**67:** Revised as follows also following a reviewer's comment:

(LL67 on p3)

studies have been conducted to improve this understanding

**88:** Corrected also following a reviewer's comment as follows:

(LL84-86 on p3)

More droplets mean more competition among them for available water vapor needed for their condensational growth, decreasing the size of individual droplets (Twomey, 1977; Albrecht, 1989).

**101:** Revised as follows:

(LL94-96 on p4)

The role of fire-generated aerosols in the development of pyroCbs and their effects on water vapor and cirrus clouds in the UTLS lacks a firm scientific understanding, and hence this paper focuses on that role of those aerosols.

**107:** Corrected.

**127:** The corresponding sentence is considered redundant and removed.

**137:** The last sentence is removed. The corresponding paragraph is shortened.

**147:** Corrected.

**157:** The corresponding sentence is revised as follows:

(LL131-132 on p5)

This study examines this dependence, not studied by Kablick et al. (2018).

**194:** Corresponding sentence is removed.

**215:** Corrected also following a reviewer's comment as follows:

(LL182-184 on p7)

These surface heat-flux values follow previous studies which adopt boreal forest emissions (Trentmann et al., 2006; Luderer et al., 2006).

**242:** Corrected.

**244:** Corrected.

**249:** Sentences are contracted as follows:

(LL211-213 on p8)

The control run adopts the unimodal lognormal distribution as an initial aerosol size distribution, a reasonable assumption in fire sites (Reid et al., 2005; Knobelspiessel et al., 2011; Lee et al., 2014).

**255:** The unit of standard deviation is μm and this is indicated in text.

**274:** The corresponding paragraph is made more concise.

**293:** The corresponding sentence is revised as follows:

(LL246-249 on p9)

The average cloud-top and cloud-base heights over the life span of the pyroCb are 10.3 km and 3.6 km in the control run, respectively, and these simulated heights are ~7% different from the satellite-retrieved values.

**309:** The repetitive sentence is removed.

**378:** Sentences of corresponding text are removed, since they contain the same information as in Introduction.

**384:** Done.

**396:** Done.

**418:** Sentence is revised as follows:

(LL349-350 on p12)

This implies that air parcels and associated updrafts in the pyroCb are stronger, reaching higher altitudes.

**436:** The corresponding sentence is considered redundant and removed.

**444:** Corrected.

**465:** Wang's work is mentioned as follows:

(LL377-383 on p13)

In summary, the pyroCb and associated updrafts cause a substantial enhancement of the transport of water vapor to the UTLS at and above the tropopause. Wang (2019) also reported this enhancement. Using modeling work and satellite observation, Wang (2019) indicated that the upward transport of water vapor in deep convective storms was possibly a major pathway through which water substance entered the stratosphere. Wang (2019) showed that the upward transport of water vapor was aided by gravity wave and its breaking in overshooting convective parcels.

**469:** We removed an unnecessary sentence and clarified text.

**493:** The sentence pointed out is considered redundant and thus removed.

**520:** CDNC is replaced with "$N_d$"

**532:** The corresponding sentence is revised as follows:

(LL443-445 on p15)

Increasing $N_d$ enhances competition among droplets for a given amount of water vapor. Enhanced competition eventually curbs the condensational growth and reduces droplet size ($R_v$).

**537:** The repetitive text is removed.

**540:** $R_v$ is defined in the title of Section 4.2.2 (LL431 on p15), $N_d$ is defined in the title of sub-section "a" in Section 4.2.2 (LL433 on p15), and LWC is defined at line number 244 on p9.

We thought that the editor may want us to define or explain the proportionality itself between $R_v$ and $(LWC/N_d)^{(1/3)}$. To explain the proportionality, Equation (2) and associated text are added.

**550:** $(LWC/N_d)^{(1/3)}$, but not $R_v$, corresponds to numbers shown in text after $(LWC/N_d)^{(1/3)}$. Since $R_v$ is linearly proportional to $(LWC/N_d)^{(1/3)}$ according to Equation (2), the larger variation of $(LWC/N_d)^{(1/3)}$ leads to the larger variation of $R_v$.

**558:** The already defined expression is removed.

**559:** Corrected.

**565, 566:** The corresponding sentence is revised as follows:

(LL473-478 on p16)

During period 1, as fire intensity weakens and updraft speed decreases, parcel equilibrium supersaturation decreases and thus, the minimum size of activated aerosol particles increases not only among the clean-scenario runs but also among the polluted-scenario runs. When the production of parcel supersaturation by updrafts and the consumption of supersaturation by droplets balance out, parcel supersaturation reaches parcel equilibrium supersaturation (Rogers and Yau, 1991).

**579:** Khain's HUCM model calculates equilibrium supersaturation at every model time step by considering dynamic effects on supersaturation (e.g., advection and updrafts) and microphysical effects on supersaturation (e.g., condensation and evaporation). For the consideration of dynamic effects and microphysical effects, dynamic and microphysical sub-time steps are considered for each model time step. After a model time step is completed, both of dynamic and microphysical impacts on supersaturation are completed.  So, we obtain equilibrium supersaturation when each model time step is completed, and average it over those time steps and areas with positive updraft speed.

**583:** Corresponding sentences are contracted by removing unnecessary words.

**596:** $(LWC/CDNC)^{(1/3)}$ is removed.

**606:** The corresponding paragraph is revised substantially to make it more concise as follows:

**(LL502-505 on p17)**

In association with larger aerosol concentration and the assumed aerosol size distribution, a smaller percentage variation of the number of activated aerosols and Nd with fire intensity is simulated in the polluted-scenario runs than in the clean-scenario runs. This smaller variation of Nd aids the greater reduction in Rv among the polluted-scenario runs.

**616:** The corresponding repetitive sentence is removed.

**636:** Throughout the manuscript, when what I am comparing is established in a sentence, the following sentences are simplified by shortening parts related to the comparison in each paragraph. For example, look at following sentences in Section 4.2.3:

To satisfy mass conservation, the freezing- and deposition-enhanced updrafts above the freezing level induce more updraft mass fluxes below the freezing level in the polluted-scenario run than in the clean-scenario run for each fire intensity. This leads to more convergence around and below cloud base in the polluted-scenario run than in the clean-scenario run for each fire intensity.

In the first sentence, the comparison between the polluted-scenario run and the clean-scenario run for each fire intensity is established, so, in the second sentence, "than in the clean-scenario run for each fire intensity" is removed.

**640:** Following a reviewer's comment, "get greater" is replaced with "increase" and the corresponding sentence is revised as follows:

**(LL519-521 on p18)**

The increasing differences in autoconversion rates between the polluted-scenario and clean-scenario runs increase those differences in the amount of cloud liquid available for freezing with weakening fire intensity (Figure 9a).

**642:** Following a reviewer's comment, "gets greater" is replaced with "increase" and the corresponding sentence is revised as follows:

**(LL521-523 on p18)**

Thus, differences in the average rate of cloud-liquid freezing and freezing-related latent heat over the period 2 between the runs increase with weakening fire intensity (Figure 9a).

**644:** Corresponding sentence is removed, following a reviewer's comment. However, following the comment here, in other parts of text, we removed the redundant statements of period.

**648:** Following the editor's and a reviewer's comments here, the repetitive text is removed.

**650:** Following the editor's and a reviewer's comments here, the repetitive text is removed.

**654:** We removed unnecessary numbers. We believe that even without a table for those numbers, what we intend to deliver can be delivered well. So, we do not add the table.

**674:** Revised as follows:
The corresponding sentences are substantially revised to make it more concise as follows:

(LL533-536 on p18)

The greater freezing and thus freezing-related latent heat eventually cause updrafts to be stronger in the polluted-scenario run starting at ~21:00 GMT (Figure 10). Then, the stronger updrafts induce deposition to be greater in the polluted-scenario run around 21:10 GMT (Figure 10).

**682:** Redundant expressions related to comparisons are removed throughout the manuscript.

**686:** Removed.

**688:** Also following a reviewer's comment, the text is revised as follows:

(LL544-545 on p19)

The higher mass fluxes and convergence below the freezing level

**693:** Redundant expressions related to comparisons are removed throughout the manuscript.

**703:** Revised as follows:

(LL554-557 on p19)

those differences in the average freezing-affected updrafts and subsequently in deposition-related latent heat over period 3 increase with weakening fire intensity (Figures 9a, 9b and 10).

**769:** Removed.

**781:** Corrected.

**801:** Redundant expressions (i.e., words in parentheses) are removed.

**806:** Revised as follows:

(LL642-645 on p22)

Much greater $N_d$ is in the polluted-scenario run than in the clean-scenario run for each fire intensity. $N_d$ decreases are smaller among the polluted-scenario runs than among the clean-scenario runs with weakening fire intensity. This situation during the initial stage induces $R_v$ to decrease much more among the polluted-scenario runs with weakening fire intensity.

**834:** Revised.

**836:** Done.

**837:** We find Koren et al. (2008) at Ilan Koren's homopage (https://www.weizmann.ac.il/EPS/Koren/publications) as a paper for invigoration. The citation is as follows:

Koren I., Martins J. V., Remer L. A. & Afargan H. (2008). Smoke invigoration versus inhibition of clouds over the Amazon. Science. 321:(5891)946-949.

We add Koren et al. (2008) in the manuscript.

**850:** Revised.

**861:** Corrected.

**1100:** "which is" is removed.

**1124:** Revised as follows:

(LL957-959 on p32)

Figure 9. Average rates of condensation, deposition and cloud-liquid freezing at all altitudes in cloudy areas and over periods (a) 2, (b) 3 and (c) 4. In panel (a), average autoconversion rates are also shown.

**Referee Report 1**

Lee et al., Examination of effects of aerosol on a pyroCb and their dependence on fire intensity and aerosol perturbation using a cloud-system resolving model, submitted to Atmos. Chem. Phys., 2019

**General comments**

In response to comments and requested from both reviewers, the authors shortened the paper and improved the structure. After making the following minor language revisions, many of which will further decrease the length of the text, I recommend the manuscript for publication in ACP. Even after implementing these changes, I strongly recommend that the manuscript is proofread for correct English before publication. Thanks!

Authors appreciate the reviewer's comments below that improved the quality of the manuscript a lot. We revised the manuscript based on the comments and the manuscript was proofread. In addition, we identified words, sentences and paragraphs that we believed required revision, and revised them to make the manuscript concise with the better structure.

In the following, each comment by the reviewer (black) is followed by our response (blue).

**Specific comments**

Please provide examples in the text of "good agreement"

In the corresponding Section 3.1, we compared the simulation to the observation in terms of cirrus cloud, cloud macro-physical (e.g., LWP, IWP, cloud-top and cloud-base heights), and reflectively fields. We believe that these fields are examples of good agreement between the simulation and the observation. These examples are described in Section 3.1 and the last sentence in Section 3.1 is as follows:

(LL258-260 on p9)

These favorable comparisons between the observed and simulated cirrus clouds, cloud macro-physical and reflectivity fields demonstrate that the pyroCb-case simulation reasonably reproduces the event.

We believe that "examples" mentioned by the reviewer here may also mean the description of quantified differences and similarity in the reflectivity field between the simulation and the observation. Hence, the corresponding text is revised as follows:

(LL252-258 on p9)

Compared are the control-run and observed reflectivity fields. Kablick et al. (2018) provide details about the reflectivity field observed by CloudSat. Observations and the control run both show increasing reflectivity up to ~7 km, decreasing reflectivity between ~7 km and ~11 km and insignificant changes in reflectivity above ~11 km in altitude (Figure 3). The average reflectivity over altitudes along the Cloudsat path in the control run is -8.1 dBZe, ~15% different from the observed value. Hence, the reflectivity field is simulated fairly well compared to the observation.

29-30 "specifically focusing on how fire-produced aerosols affect this role via a modeling framework" -> "specifically focusing via a modeling framework on how fire-produced  aerosols affect this role"

This corresponding sentence is revised as follows:

(LL27-29 on p2)

Through a modeling framework, this study investigates how a pyrocumulonimbus (pyroCb) event influences water vapor concentrations and cirrus cloud properties near the tropopause, specifically focusing on how fire-produced aerosols affect this role.

"studies to improve this understanding has been going on" -> "studies have been  conducted to improve this understanding"

Done.

Insert "of pyroCB" after "microphysical properties"

Done.

"make" -> "decrease"

Done.

remove "smaller"

Done.

86-87 remove "or increasing aerosol"

Done.

88-89 replace "this more competition makes" with "decreasing the size of"

Done.

remove "smaller"

Done.

change "sizes of droplets" to "droplet sizes"

Done.

"autoconversion that is" -> "autoconversion,"

Done.

"for them to" -> "in which they", "be" -> "become"

Done.

94-95 "More cloud liquid is thus available for transport to places above the freezing level  by updrafts." -> "More cloud liquid is thus available for transport by updrafts to altitudes  above the freezing level."

Done.

"by" -> "of"

Done.

"cloud typical properties" -> "cloud properties"

Done.

"For the simplicity of the term" -> "For simplicity"

It is replaced with "For simplicity herein" (LL102 on p4)

"have shown" -> "showed", "updrafts
throughout the manuscript, ex. 115: "the pyroCb development and its impacts on the  UTLS" ->
"pyroCb development and its impacts on the UTLS"

"have shown" is replaced with "demonstrated"

 "the pyroCb development and its impacts on the UTLS" is replaced with "pyroCb development
 and its impacts on the UTLS" throughout the manuscript.

Regarding updrafts the following is added:

(LL102-104 on p4)

For simplicity herein, an "updraft" refers to the general upward motion of convective air, or to the
actual updraft speed representing the updraft or convective intensity, depending on the context.

Based on the added sentence above, we generally use "updrafts" in places where the meaning of "updrafts", which can be the upward motion or the updraft speed or both, is rather obvious to readers based on the context. However, in a few sentences where conventional rules indicate that the use of "updraft speeds" is better than that of "updrafts", we write "updraft speeds" explicitly.

An example of the sentences is as follows:

(LL478-482 on p16-17)

Mostly due to greater aerosol concentrations, associated average equilibrium supersaturation and minimum size of activated aerosol particles over areas with positive updraft speeds and period 1 are lower and larger, respectively, in the polluted-scenario run than in the clean-scenario run for each fire intensity (Rogers and Yau, 1991).

In this example sentence, simply because the word "positive" is generally coupled with "updraft speed" not but with "updrafts", we just use this customary expression "positive updraft speed"

"impacts" -> "impacts on"

The corresponding sentence is considered redundant and removed.

"parameterize" -> "parameterizes"

The corresponding sentence is considered redundant and removed.

remove "have"

[Figure]

The corresponding sentence is considered redundant and removed, following editor's comment.

"cloud-particle" -> "cloud-particle and aerosol"

The corresponding sentence is considered redundant and removed, following editor's comment.

"varying with varying" -> "varying with"

The corresponding sentence is considered redundant and removed, following editor's comment.

137-138 "the bulk schemes in general uses" -> "bulk schemes in general use"

The corresponding sentence is considered redundant and removed, following editor's comment.

138-139 remove "which are not able to consider the variation of collection efficiencies  and terminal velocities in reality"

The corresponding sentence is considered redundant and removed, following editor's comment.

139-140 "This makes the bin scheme more sophisticated than the bulk scheme." ->  "Thus, the bin scheme is more sophisticated than the bulk scheme."

The corresponding sentence is considered redundant and removed, following editor's comment.

"are strongly dependent" -> "is strongly dependent"

Done.

"are referred to as fire-driven updrafts, henceforth" -> "are henceforth referred to as fire-driven updrafts"

Done.

153-156 "Aerosol effects on clouds are initiated by an increase in aerosol concentration, which can be caused by an increase in aerosol emission at and near the surface, and dependent on how much aerosol concentration increases, or on the magnitude of an increase in aerosol concentration, i.e., aerosol perturbation" -> "Aerosol effects on clouds are initiated by an increase in aerosol concentration, which can be caused by an increase in aerosol emission at and near the surface, and dependent on how much aerosol concentration increases (aerosol perturbation)"

The corresponding text is simplified as follows to make it more concise:

(LL129-131 on p4)

Effects of fire-induced increases in aerosol concentration on pyroCbs are likely to be dependent on how much aerosol concentration increases (aerosol perturbation) (e.g., Rosenfeld et al., 2008; Koren et al., 2012).

"has not been" -> "was not"

The corresponding sentence is revised as follows:

This study examines this dependence, not studied by Kablick et al. (2018).

166-167 "Shortwave and longwave radiation parameterizations have been included in  all simulations by adopting" -> "Shortwave and longwave radiation is parameterized by"

Done.

"To represent the microphysical processes" -> "To represent microphysical  processes"

Done.

"The cloud-droplet" -> "A cloud-droplet", "parameterization, which is based" -> "parameterization based"

Done.

"Arbitrary aerosol mixing states and arbitrary" -> "Arbitrary aerosol mixing states  and"

Done.

"case is performed" -> "case was performed"

The corresponding sentence is revised as follows:

(LL161-162 on p6)

The control run for an observed pyroCb case involved a forested site in the Canadian Northwest Territories (60.03 ° N, 115.45° W).

"the site and the pyroCb" -> "the site and pyroCb"

Done.

remove "between them"

Done.

"from Ft. Smith" -> "from the Ft. Smith"

Done.

204-205 "These tendencies are horizontally homogeneous and applied to the control  run every time step by interpolation" -> "These tendencies are applied to the control run  every time step by interpolation, in a horizontally homogeneous manner"

The corresponding sentence is revised as follows:

(LL169-171 on p6)

Temperature and humidity tendencies at each altitude from sequential soundings are obtained and applied to the control run every time step by interpolation in a horizontally homogeneous manner.

"lengths" -> "extents"

Done.

"For the simulation, the" -> "The simulation"

Done.

211-212 "simulation, at the center of the simulation domain, a fire spot with a diameter  of 40 km is placed" -> "simulation, a fire spot with a diameter of 40 km is placed at the  center of the simulation domain"

Done.

"the previous studies which are Trentmann et al. (2006) and Luderer et al. (2006)  and adopt boreal forest emissions" -> "previous studies which adopt boreal forest  emissions (Trentmann et al. (2006) and Luderer et al. (2006))"

Done.

"idealized and this enables" -> "idealized, enabling"

Done.

remove "aerosol properties that can be represented by"

Done.

235-236 "~50-70% of organic-carbon (OC) compounds, ~5-10% of black-carbon (BC) material, and ~20-45% of inorganic species" -> "~50-70% organic-carbon (OC)  compounds, ~5-10% black-carbon (BC) material, and ~20-45% inorganic species"

Done.

Insert new paragraph after "in the control run."

Done.

247-249 "According to Reid et al. (2005), Knobelspiessel et al. (2011), and Lee et al.  (2014), it is reasonable to assume that the initial aerosol size distribution follows the  unimodal lognormal distribution in fire sites. Hence, the control run adopts the unimodal  lognormal distribution as an initial aerosol size distribution." -> "The control run adopts  the unimodal lognormal distribution as an initial aerosol size distribution, a reasonable assumption in fire sites (Reid et al. (2005), Knobelspiessel et al. (2011), Lee et al.  (2014))"

Done.

Throughout, e.g. 251-253 "median aerosol diameter and standard deviation of the distribution" -> "median and standard deviation aerosol diameter"

Done.

Insert new paragraph after "(Pruppacher and Klett, 1978)."

Done.

"captured" -> "capture"

Done.

"points" -> "point"

Done.

267, 270 "counterparts" -> "values"

Done.

279-280 **"Figure 1. This field in Figure 2 represents" -> "Figure 1, representing"

Done.

Throughout, e.g. 284, including figures, "averaged" <cloud property> -> "average" <cloud property>

Done.

"well as" -> "well"

Done.

Throughout, including figures, anytime multiple rounds are referred to together in a row, e.g. 33,3 "the medium run and the weak run" -> "the medium and weak runs"

Done.

remove "or aerosol perturbation"

Done.

353, 356 "intensity" -> "intensities"

Done.

"corresponds" -> "correspond"

Done.

remove "themselves"

The corresponding paragraph is considered redundant and removed.

"(2018) by" -> "(2018),"

The corresponding paragraph is considered redundant and removed.

379-380 "The updraft mass flux is one of the most representative variables that are  indicative of the cloud dynamic intensity and the magnitude of convective invigoration." -> ""The updraft mass flux is one of the most indicative variables of cloud dynamic intensity and magnitude of convective invigoration."

The corresponding sentence is revised as follows:

(LL318-319 on p11)

The updraft mass flux is one of the most indicative variables of the upward air motion and magnitude of convective invigoration.

"is" -> "was"

Done.

382-383 "17:00 GMT on August 5th and 12:00 GMT on August 6th, and 17:00 GMT on August 5th is a time around which the pyroCb starts to from" -> "17:00 GMT on August 5th, approximately when the pyroCb starts to form, and 12:00 GMT on August 6th, and 17:00 GMT on August 5th is"

Done.

"that" -> "which"

Done.

390-393 "Considering that the stratosphere is between the tropopause and its top that is  generally ~ 50 km in altitude, the defined lower stratosphere occupies around a quarter  of the total vertical extent of the stratosphere." -> "The defined lower stratosphere  occupies around a quarter of the total vertical extent of the stratosphere, the top of   which is generally ~ 50 km in altitude."

The corresponding sentences are simplified as follows:

(LL329-330 on p11)

The defined upper troposphere and lower stratosphere occupy around a quarter of the total vertical extent of the troposphere and stratosphere, respectively.

"the cloudy columns and that over" -> "cloudy and"

Done.

"in a part" -> "in the part"

Done.

410, 411 remove "those"

Done.

remove "or the pyroCb"

Done.

remove "both"

Done.

"as compared to that in" -> "versus"

Done.

430-433 "These averaged fluxes are over cloudy columns for the simulation period  between 17:00 GMT on August 5th and 12:00 GMT on August 6th. The averaged water-  vapor fluxes vary from 8.30x10-6 kg m-2 s-1 in the control run to 8.21x10-6 kg m-2 s-1  in the low-aerosol run." -> "The averaged water-vapor fluxes over cloudy columns for   the simulation period between 17:00 GMT on August 5th and 12:00 GMT on August 6th  vary from 8.30x10-6 kg m-2 s-1 in the control run to 8.21x10-6 kg m-2 s-1 in the low-  aerosol run."

The corresponding sentences are revised as follows:

(LL356-360 on p12)

The small variation in updraft mass fluxes between the runs results in a small variation in the average water-vapor fluxes at the tropopause from 8.30×10-6 kg m-2 s-1 in the control run to 8.21×10-6 kg m-2 s-1 in the low-aerosol run over cloudy columns for the simulation period between 17:00 GMT on August 5th and 12:00 GMT on August 6th.

"only are" -> "are only"

We thought it was better to put "are", since "only" is used to emphasize cirrus clouds are composed of ice crystals but not other types of hydrometeors. Stated differently, "only" is not used to emphasize cirrus clouds are between 9 and 13 km. In case we use "are only" as recommended by the reviewer here, the corresponding sentence emphasizes cirrus clouds are between 9 and 13 km. This is not what we intend to emphasize and thus we just put "are" here.

"there are the presence of" -> "there is the presence of"

The corresponding sentence is considered redundant and removed.

"cloud-ice number concentration and cloud-ice size" -> "cloud-ice number  concentration and size"

Done.

"micron" -> μm

Done.

443-444 "The altitudes between 9 km and 13 km correspond to a part of the UTLS  below the troposphere. Henceforth, the UTLS cirrus clouds mean those clouds in a part  of the UTLS below the tropopause." -> "The altitudes between 9 km and 13 km  correspond to a part of the UTLS below the tropopause, and henceforth, "UTLS cirrus  clouds" refer to clouds in the troposphere."

The corresponding sentences are simplified as follows:

(LL366-367 on p13)

Henceforth, "the UTLS cirrus clouds" refer to clouds in the upper troposphere.

445-446 "Updrafts in the pyroCb produce supersaturation, which leads to the generation  of cloud ice mass and associated cirrus clouds via deposition, the primary source of cloud-ice mass" -> "Updrafts in the pyroCb produce supersaturation, which leads to the primary source of cloud-ice mass and associated cirrus clouds via deposition"

Done.

"circus" -> "cirrus"

Done.

452, 453 remove "large"

As a process of making the corresponding paragraph succinct, the last two sentences in the paragraph are removed. In the removed sentences, "significant variations of cloud-ice number concentration and size" are indicated. Although those sentences are removed, we believe that it is important to deliver this indication in the rest of the sentences not removed. For this deliverance, "significant" replaces "large" in the sentence pointed out here by the reviewer.

456-457 "simulations (Figure 7), and thus a negligible variation of the mass of the UTLS cirrus clouds" -> "simulations (Figure 7), and thus a negligible variation of UTLS cirrus cloud mass"

Done.

"The role, which is played by" -> "The role of"

The corresponding paragraph is revised substantially following the editor's comment and the corresponding sentence is revised as follows:

(LL384-386 on p13)

The effects of fire-generated aerosols on the pyroCb updrafts, cirrus cloud mass and the enhancement of the water vapor transport are insignificant when fire intensity is strong.

464-465 "the UTLS at and above the tropopause, and in the production of the mass of the UTLS cirrus clouds is not significant for strong fire intensity" -> "the UTLS at and above the tropopause, and in the production UTLS cirrus cloud mass, is not significant for strong fire intensity"

The corresponding paragraph is revised substantially following the editor's comment and the corresponding sentence is revised as follows:

(LL384-386 on p13)

The effects of fire-generated aerosols on the pyroCb updrafts, cirrus cloud mass and the enhancement of the water vapor transport are insignificant when fire intensity is strong.

"the mass of the UTLS cirrus clouds" -> "UTLS cirrus cloud mass"

The corresponding sentence is considered redundant and removed.

"updrafts, produced" -> "updrafts produced"

Done.

480-481 "when the fire-generated surface heat fluxes and the fire intensity" -> "when the fire intensity and fire-generated surface heat fluxes"

Done.

485-486 "relative magnitude of aerosol-induced perturbation of latent heat to" -> "relative magnitudes of aerosol-induced perturbation of latent heat and"

Done.

"does vary" -> "varies"

Done.

501-503 "Hence, it can be said that percentage differences in updraft mass fluxes mean percentage differences in updraft speed with good confidence." -> "Hence, it can be  said with good confidence that percentage differences in updraft mass fluxes mean  percentage differences in updraft speed."

The corresponding sentence is revised as follows:

(LL319-323 on p11)

Since the updraft mass flux is updraft speed that is multiplied by air density, and air density at each altitude varies negligibly, differences in updraft mass fluxes are mostly explained by those in updraft speeds among the simulations. Hence, with good confidence, differences in updraft mass fluxes mean those in updraft speeds.

"run than the clean-scenario run" -> "run and clean-scenario runs"

Done.

and similar equations: remove "the"

Done.

"gets larger" -> "increases"

Done.

"than the" -> "than in the"

Done.

"the averaged LWC and the averaged CDNC" -> "the average LWC and CDNC"

Done.

550, 551 "varies" -> "decreases"

Done.

"reduce" -> "decreases"

Done.

"to increase" -> "increasing", "lowers" -> "decreases"

The corresponding sentence is revised as follows:

(LL473-478 on p16)

During period 1, as fire intensity weakens and updraft speed decreases, parcel equilibrium supersaturation decreases and thus, the minimum size of activated aerosol particles increases not only among the clean-scenario runs but also among the polluted-scenario runs. When the production of parcel supersaturation by updrafts and the consumption of supersaturation by droplets balance out, parcel supersaturation reaches parcel equilibrium supersaturation (Rogers and Yau, 1991).

"averaged associated" -> "associated average"

The corresponding sentence is revised as follows:

(LL478-482 on p16-17)

Mostly due to greater aerosol concentrations, associated average equilibrium supersaturation and minimum size of activated aerosol particles over areas with positive updraft speeds and period 1 are lower and larger, respectively, in the polluted-scenario run than in the clean-scenario run for each fire intensity (Rogers and Yau, 1991).

"reduces" -> "decreases"

Done.

remove "size in"

Done.

"reduces" -> "decreases"

Done.

"less close to" -> "further from"

The corresponding sentence is revised as follows:

(LL491-494 on p17)

A smaller portion of the total aerosol concentration is in the size range closer to the right tail of the distribution as long as the range is on the right-hand side of the distribution peak where most of aerosol activation occurs.

588-590 "distribution; most of aerosol activation occurs for aerosol sizes on the right- hand side of the distribution peak, here we are only concerned with the size ranges on the right-hand side." -> "distribution. Here we are only concerned with the size ranges on the right-hand side of the distribution peak, where most of aerosol activation occurs."

The corresponding sentence is revised as follows:

(LL491-494 on p17)

A smaller portion of the total aerosol concentration is in the size range closer to the right tail of the distribution as long as the range is on the right-hand side of the distribution peak where most of aerosol activation occurs.

Throughout the manuscript: <cloud property>", which is averaged" -> <cloud property>"averaged" e.g. 593 "CNDC, which is averaged" -> "CNDC averaged"

Done.

"to greater" -> "to a greater"

Done.

"thus the" -> "thus a"

Done.

"rates, which are averaged" -> "rates averaged"

Done.

624-629 "The averaged autoconversion rates over period 2 reduce from 3.61x10-6 g m- 3 s-1 in the control run with strong fire intensity to 0.93x10-6 g m-3 s-1 in the weak run with weak fire intensity through 2.01x10-6 g m-3 s-1 in the medium run with medium fire intensity by 74%. Those averaged autoconversion rates reduce from 4.52x10-6 g m-3 s- 1 in the low-aerosol run with strong fire intensity to 3.94x10-6 g m-3 s-1 in the weak-low run with weak fire intensity through 4.43x10-6 g m-3 s-1 in the medium-low run with medium fire intensity by 14%." -> "The averaged autoconversion rates over period 2 decrease by 74% from 3.61x10-6 g m-3 s-1 in the control run with strong fire intensity to 0.93x10-6 g m-3 s-1 in the weak run with weak fire intensity through 2.01x10-6 g m-3 s- 1 in the medium run with medium fire intensity. Those averaged autoconversion rates decrease by 14%from 4.52x10-6 g m-3 s-1 in the low-aerosol run with strong fire intensity to 3.94x10-6 g m-3 s-1 in the weak-low run with weak fire intensity through 4.43x10-6 g m-3 s-1 in the medium-low run with medium fire intensity."

The corresponding sentence is shortened substantially as follows to make it more concise and succinct:

(LL513-516 on p18)

Due to the larger absolute and percentage reduction in Rv among the polluted-scenario runs than among the clean-scenario runs with weakening fire intensity during period 2, the average autoconversion rates decrease by 74% (14%) from the control (low-aerosol) to weak (weak-low) runs (Figure 9a).

"get greater" -> "increase"

Done.

"smaller" -> "lower"

The corresponding sentence is considered redundant and removed.

"freezing, which is averaged" -> "freezing averaged"

The corresponding sentence is considered redundant and removed.

640, 642 "get greater" -> "increase"

Done.

644-648 "When fire intensity is strong, the difference in freezing-related latent heat, which is averaged in cloudy areas and period 2, between the polluted-scenario run, which is the control run, and the clean-scenario run, which is the low-aerosol run," -> "When fire intensity is strong, the difference in freezing-related latent heat averaged in cloudy areas and period 2 between the polluted-scenario run (control run) and the clean-scenario run (low-aerosol run)"

Following the other reviewer's comment, the corresponding sentences are simplified as follows:

(LL521-523 on p18)

Thus, differences in the average rate of cloud-liquid freezing and freezing-related latent heat over the period 2 between the runs increase with weakening fire intensity (Figure 9a).

649-650 "while with weak fire intensity, that difference between the polluted-scenario run, which is the weak run, and the clean-scenario run, which is the weak-low run" -> "while with weak fire intensity, that difference between the polluted-scenario run (weak run) and the clean-scenario run (weak-low run)"

Following the other reviewer's comment, the corresponding sentences are simplified as follows:

(LL521-523 on p18)

Thus, differences in the average rate of cloud-liquid freezing and freezing-related latent heat over the period 2 between the runs increase with weakening fire intensity (Figure 9a).

651-652 "differences, which are calculated by Equation (1)," -> "differences (calculated by Equation (1))"

The corresponding sentence is removed as a process of making the corresponding paragraph more concise and succinct, following the other reviewer's comment.

653-654 "and the clean-scenario run from 9% with strong fire intensity to 83% with weak fire intensity through 51% with medium fire intensity over the period 2." -> "and the clean-scenario run over the period 2 from 9% with strong fire intensity to 83% with weak fire intensity to 51% with medium fire intensity."

The corresponding sentence is removed as a process of making the corresponding paragraph more concise and succinct, following the other reviewer's comment.

657, 674 "deposition, which is averaged" -> "deposition averaged"

It is replaced with "average deposition"

"period 2," -> "period 2"

Done.

"transportation" -> "transport"

Done.

remove "as compared to those before 20:30 GMT"

Done.

"from" -> "starting at"

Done.

"in polluted-scenario" -> "in the polluted-scenario"
Done.

"The more mass fluxes and the more convergence" -> "The higher mass fluxes and convergence"

Done.

"be greater" -> "increase"

Done.

695-696 "thus, enhancing freezing, deposition, condensation, and updrafts further." ->  "thus further enhancing freezing, deposition, condensation, and updrafts."

Done.

708, 710, 713, 727,741 "get greater" -> "increase"

Done.

714-715 "Then, the increases in condensation, in turn, further enhance the increases in  updrafts in the polluted-scenario run for each fire intensity." -> "The increases in condensation further enhance the increases in updrafts in the polluted-scenario run for each fire intensity."

The corresponding sentences are revised as follows:

(LL564-566 on p19)

The greater increases in condensation cause the greater further enhancement of the increases in updrafts in the polluted-scenario run with weaker fire intensity.

"induce" -> "induces"

Done.

752-753 "This possibility is not that unrealistic, since stronger fire likely involves more material burnt and more aerosols from it." -> "This possibility is not that unrealistic, since stronger fires likely involve more material burned and higher aerosol emissions."

Done.

757, 763, 778 "reduces" -> "decreases"

Done.

"less" -> "lower"

Done.

759-760 "For strong fire, the perturbation-related aerosol concentration is 30000

cm-3, for medium fire, it is 15000 cm-3, and for weak fire, it is 7500 cm-3." -> "The perturbation-related aerosol concentration is 30000 cm-3 for strong fire, 15000 cm-3 for medium fire, and 7500 cm-3 for weak fire."

Done.

"when fire-induced" -> "when the fire-induced"

Done.

"has the" -> "has an"

Done.

"run; when" -> "run, and when"

The corresponding sentence is revised as follows:

(LL613-615 on p21)

Based on this, the medium run is repeated with a fire-induced aerosol perturbation of 2000 cm$^{-3}$ down from 15000 cm$^{-3}$ (the medium-2000 run). The weak run is repeated with a fire-induced aerosol perturbation of 1000 cm$^{-3}$ down from 7500 cm$^{-3}$ (the weak-1000 run).

"with the weakening" -> "with weakening fire"

Done.

"whether those aerosol perturbations vary with varying fire intensity or not" -> "whether or not those aerosol perturbations vary with varying fire intensity"

Done.

"transport water vapor to the tropopause and above efficiently" -> "efficiently transport water vapor to the tropopause and above"

Done.

"gets more" -> "becomes"

Done.

"reduce" -> "decrease"

Done.

"makes" -> "results in"

The corresponding sentence is revised as follows:

(LL646-648 on p22)

This reduces autoconversion more among the polluted-scenario runs and increases differences in autoconversion between the polluted-scenario and clean-scenario runs as fire intensity weakens.

"increase" -> "increasing"

The corresponding sentence is revised as follows:

(LL646-648 on p22)

This reduces autoconversion more among the polluted-scenario runs and increases differences in autoconversion between the polluted-scenario and clean-scenario runs as fire intensity weakens.

"It is true that the" -> "The"

Done.

"transportation" -> "transport"

Done.

"reduce" -> "decrease"

Done.

"came up with" -> "generated", "preformed" -> "performed"

Done.

"revised manuscript based on the reviewers' comments and perform" -> "revised  the manuscript based on the reviewers' comments and performed"

Table 2 "Meidum-low" -> "Medium-low"

Done.

1172, 1190 "The averaged" -> "Averaged"

"The averaged" is replaced with "average"

Figure 10 "Differences in the averaged values" -> "Differences in average values"

Done.

**Referee Report 2**

Review of:
"Examination of effects of aerosol on a pyroCb and their dependence on fire intensity and aerosol perturbation using a cloud-system resolving model"

Authors: Seoung Soo Lee, George Kablick III, Zhanqing Li

Recommend minor revisions.

General comment:

This revised manuscript is much improved. The results are clearer and presented more concisely in most areas. Following a few additional clarifications and rewording, I think this paper will be ready for publication.

Authors appreciate the reviewer's comments below that improved the quality of the manuscript a lot. In the following, each comment by the reviewer (black) is followed by our response (blue).

Specific comments:

1.Lines 126: Use of the phrase "has been going on" seems a little awkward.

(LL67 on p3)

It is corrected to be "have been conducted to improve this understanding"

2.Lines 248-251: The sentence beginning with "Aerosol effects on clouds…" is confusing as stated. Please make this clearer.

Revised as follows by reflecting another reviewers' comment as well:

(LL129-132 on p5)

Effects of fire-induced increases in aerosol concentration on pyroCbs are likely dependent on how much aerosol concentration increases (aerosol perturbation) (e.g., Rosenfeld et al., 2008; Koren et al., 2012). This study examines this dependence, not studied by Kablick et al. (2018).

3.Line 373: Please find more recent specific references related to IN observations.

Fan et al. (2014 and 2017) are added.

4.Line 527: Grammar error. "pyro Cb starts to from…"

Corrected.

5.Lines 1356-1367: This paragraph is still quite wordy, and thus, it's difficult to read and easily discern the main take-away point. Please reword to be more concise. The specific numbers related to differences in latent heating are not particularly meaningful; the % change is more meaningful and is the basis of your main point here. You could include the actual numbers in a table if you wish.

Number are removed *and the corresponding text is revised as follows:*

(LL519-523 on p18)

The increasing differences in autoconversion rates between the polluted-scenario and clean-scenario runs increase those differences in the amount of cloud liquid available for freezing with weakening fire intensity (Figure 9a). Thus, differences in the average rate of cloud-liquid freezing and freezing-related latent heat over the period 2 between the runs increase with weakening fire intensity (Figure 9a).

6.Lines 1611-1612: This sentence is worded in such a way that makes it sound like you are saying that the fire intensity weakens due to reduction in LWC. Please reword to better state your point.

The corresponding text is considered redundant and removed.

[revised manuscript text omitted]

intensity.

[Figure]

**Figure 1**

[Figure]

**Figure 2**

[Figure]

**Figure 3**

[Figure]

**Figure 4**

[Figure]

**Figure 5**

[Figure]

**Figure 6**

[Figure]

**Figure 7**

[Figure]

[Figure]

**Figure 8**

[Figure]

[Figure]

**Figure 9**

c   Period 4 (23 GMT on August 5th - 12 GMT on August 6th
; mature and decaying stages)

[Figure]

**Figure 9**

Differences in the averaged values a    Control run minus Low-aerosol run

[Figure]

b    Medium run minus Medium-low run

[Figure]

**Figure 10**

[Figure]

**Figure 10**

| Page 47: [1] Deleted | Seoung Soo Lee | 11/29/19 7:32:00 AM |
|---|---|---|

| Page 47: [2] Deleted | Seoung Soo Lee | 12/13/19 11:19:00 AM |
|---|---|---|

| Page 47: [3] Deleted | Seoung Soo Lee | 12/2/19 1:40:00 PM |
|---|---|---|

| Page 50: [4] Deleted | Seoung Soo Lee | 11/28/19 8:16:00 AM |
|---|---|---|

| Page 50: [4] Deleted | Seoung Soo Lee | 11/28/19 8:16:00 AM |
|---|---|---|

| Page 50: [4] Deleted | Seoung Soo Lee | 11/28/19 8:16:00 AM |
|---|---|---|

| Page 50: [5] Formatted | Seoung Soo Lee | 11/28/19 8:12:00 AM |
|---|---|---|

Font: (Asian) Batang, Not Expanded by / Condensed by

| Page 50: [5] Formatted | Seoung Soo Lee | 11/28/19 8:12:00 AM |
|---|---|---|

Font: (Asian) Batang, Not Expanded by / Condensed by

| Page 50: [5] Formatted | Seoung Soo Lee | 11/28/19 8:12:00 AM |
|---|---|---|

Font: (Asian) Batang, Not Expanded by / Condensed by

| Page 50: [5] Formatted | Seoung Soo Lee | 11/28/19 8:12:00 AM |
|---|---|---|

Font: (Asian) Batang, Not Expanded by / Condensed by

| Page 50: [5] Formatted | Seoung Soo Lee | 11/28/19 8:12:00 AM |
|---|---|---|

Font: (Asian) Batang, Not Expanded by / Condensed by

| Page 50: [5] Formatted | Seoung Soo Lee | 11/28/19 8:12:00 AM |
|---|---|---|

Font: (Asian) Batang, Not Expanded by / Condensed by

| Page 50: [5] Formatted | Seoung Soo Lee | 11/28/19 8:12:00 AM |
|---|---|---|

Font: (Asian) Batang, Not Expanded by / Condensed by

| Page 50: [5] Formatted | Seoung Soo Lee | 11/28/19 8:12:00 AM |
|---|---|---|

Font: (Asian) Batang, Not Expanded by / Condensed by

| Page 50: [5] Formatted | Seoung Soo Lee | 11/28/19 8:12:00 AM |
|---|---|---|

Font: (Asian) Batang, Not Expanded by / Condensed by

| Page 50: [5] Formatted | Seoung Soo Lee | 11/28/19 8:12:00 AM |
|---|---|---|

Font: (Asian) Batang, Not Expanded by / Condensed by

| Page 50: [5] Formatted | Seoung Soo Lee | 11/28/19 8:12:00 AM |
|---|---|---|

Font: (Asian) Batang, Not Expanded by / Condensed by

| Page 50: [5] Formatted | Seoung Soo Lee | 11/28/19 8:12:00 AM |
|---|---|---|

Font: (Asian) Batang, Not Expanded by / Condensed by

| Page 50: [5] Formatted | Seoung Soo Lee | 11/28/19 8:12:00 AM |
|---|---|---|

Font: (Asian) Batang, Not Expanded by / Condensed by

| Page 50: [5] Formatted | Seoung Soo Lee | 11/28/19 8:12:00 AM |
|---|---|---|

Font: (Asian) Batang, Not Expanded by / Condensed by

| Page 50: [5] Formatted | Seoung Soo Lee | 11/28/19 8:12:00 AM |
|---|---|---|

Font: (Asian) Batang, Not Expanded by / Condensed by

| Page 50: [6] Deleted | Seoung Soo Lee | 11/26/19 9:25:00 AM |
|---|---|---|

| Page 50: [6] Deleted | Seoung Soo Lee | 11/26/19 9:25:00 AM |
|---|---|---|

| Page 50: [6] Deleted | Seoung Soo Lee | 11/26/19 9:25:00 AM |
|---|---|---|

| Page 50: [6] Deleted | Seoung Soo Lee | 11/26/19 9:25:00 AM |
|---|---|---|

| Page 50: [6] Deleted | Seoung Soo Lee | 11/26/19 9:25:00 AM |
|---|---|---|

| Page 50: [6] Deleted | Seoung Soo Lee | 11/26/19 9:25:00 AM |
|---|---|---|

| Page 50: [7] Deleted | Seoung Soo Lee | 12/3/19 8:30:00 AM |
|---|---|---|

| Page 50: [7] Deleted | Seoung Soo Lee | 12/3/19 8:30:00 AM |
|---|---|---|

| Page 50: [8] Deleted | Seoung Soo Lee | 11/28/19 8:18:00 AM |
|---|---|---|

| Page 50: [8] Deleted | Seoung Soo Lee | 11/28/19 8:18:00 AM |
|---|---|---|

| Page 50: [8] Deleted | Seoung Soo Lee | 11/28/19 8:18:00 AM |
|---|---|---|

| Page 50: [8] Deleted | Seoung Soo Lee | 11/28/19 8:18:00 AM |
|---|---|---|

| Page 50: [8] Deleted | Seoung Soo Lee | 11/28/19 8:18:00 AM |

| Page 50: [8] Deleted | Seoung Soo Lee | 11/28/19 8:18:00 AM |

| Page 50: [8] Deleted | Seoung Soo Lee | 11/28/19 8:18:00 AM |

| Page 50: [9] Deleted | Seoung Soo Lee | 11/28/19 2:09:00 PM |

| Page 50: [9] Deleted | Seoung Soo Lee | 11/28/19 2:09:00 PM |

| Page 50: [9] Deleted | Seoung Soo Lee | 11/28/19 2:09:00 PM |

| Page 50: [9] Deleted | Seoung Soo Lee | 11/28/19 2:09:00 PM |

| Page 50: [9] Deleted | Seoung Soo Lee | 11/28/19 2:09:00 PM |

| Page 50: [9] Deleted | Seoung Soo Lee | 11/28/19 2:09:00 PM |

| Page 50: [9] Deleted | Seoung Soo Lee | 11/28/19 2:09:00 PM |

| Page 50: [9] Deleted | Seoung Soo Lee | 11/28/19 2:09:00 PM |

| Page 50: [10] Formatted | Seoung Soo Lee | 11/26/19 9:28:00 AM |

Subscript

| Page 50: [10] Formatted | Seoung Soo Lee | 11/26/19 9:28:00 AM |

Subscript

| Page 50: [10] Formatted | Seoung Soo Lee | 11/26/19 9:28:00 AM |

Subscript

| Page 50: [10] Formatted | Seoung Soo Lee | 11/26/19 9:28:00 AM |

Subscript

| Page 50: [10] Formatted | Seoung Soo Lee | 11/26/19 9:28:00 AM |

Subscript

| Page 50: [10] Formatted | Seoung Soo Lee | 11/26/19 9:28:00 AM |
|---|---|---|

Subscript

| Page 50: [10] Formatted | Seoung Soo Lee | 11/26/19 9:28:00 AM |
|---|---|---|

Subscript

| Page 50: [10] Formatted | Seoung Soo Lee | 11/26/19 9:28:00 AM |
|---|---|---|

Subscript

| Page 50: [10] Formatted | Seoung Soo Lee | 11/26/19 9:28:00 AM |
|---|---|---|

Subscript

| Page 51: [11] Deleted | Seoung Soo Lee | 11/26/19 9:34:00 AM |
|---|---|---|

| Page 51: [12] Deleted | Seoung Soo Lee | 11/28/19 8:23:00 AM |
|---|---|---|

| Page 51: [13] Deleted | Seoung Soo Lee | 11/28/19 2:13:00 PM |
|---|---|---|

| Page 51: [14] Deleted | Seoung Soo Lee | 11/28/19 2:15:00 PM |
|---|---|---|

| Page 51: [15] Deleted | Seoung Soo Lee | 12/13/19 12:28:00 PM |
|---|---|---|

| Page 51: [16] Deleted | Seoung Soo Lee | 12/5/19 9:06:00 AM |
|---|---|---|

| Page 53: [17] Deleted | Seoung Soo Lee | 12/5/19 9:35:00 AM |
|---|---|---|

| Page 53: [18] Deleted | Seoung Soo Lee | 12/13/19 12:58:00 PM |
|---|---|---|

| Page 53: [19] Deleted | Seoung Soo Lee | 11/29/19 10:35:00 AM |
|---|---|---|

| Page 53: [20] Deleted | Seoung Soo Lee | 11/29/19 10:36:00 AM |
|---|---|---|

| Page 53: [21] Deleted | Seoung Soo Lee | 11/29/19 10:44:00 AM |
|---|---|---|

| Page 53: [22] Deleted | Seoung Soo Lee | 11/29/19 10:48:00 AM |
|---|---|---|

| Page 53: [23] Deleted | Seoung Soo Lee | 11/29/19 10:49:00 AM |
| Page 54: [24] Deleted | Seoung Soo Lee | 11/26/19 1:45:00 PM |
| Page 54: [25] Deleted | Seoung Soo Lee | 12/13/19 1:12:00 PM |
| Page 54: [26] Deleted | Seoung Soo Lee | 11/26/19 1:52:00 PM |
| Page 55: [27] Deleted | Seoung Soo Lee | 12/5/19 10:51:00 AM |
| Page 55: [28] Deleted | Seoung Soo Lee | 11/26/19 2:02:00 PM |
| Page 56: [29] Deleted | Seoung Soo Lee | 12/5/19 10:48:00 AM |
| Page 56: [29] Deleted | Seoung Soo Lee | 12/5/19 10:48:00 AM |
| Page 56: [29] Deleted | Seoung Soo Lee | 12/5/19 10:48:00 AM |
| Page 56: [29] Deleted | Seoung Soo Lee | 12/5/19 10:48:00 AM |
| Page 56: [29] Deleted | Seoung Soo Lee | 12/5/19 10:48:00 AM |
| Page 56: [29] Deleted | Seoung Soo Lee | 12/5/19 10:48:00 AM |
| Page 56: [29] Deleted | Seoung Soo Lee | 12/5/19 10:48:00 AM |
| Page 56: [29] Deleted | Seoung Soo Lee | 12/5/19 10:48:00 AM |
| Page 56: [29] Deleted | Seoung Soo Lee | 12/5/19 10:48:00 AM |

| | | |
|---|---|---|
| **Page 56: [29] Deleted** | **Seoung Soo Lee** | **12/5/19 10:48:00 AM** |
| **Page 56: [29] Deleted** | **Seoung Soo Lee** | **12/5/19 10:48:00 AM** |
| **Page 56: [29] Deleted** | **Seoung Soo Lee** | **12/5/19 10:48:00 AM** |
| **Page 56: [29] Deleted** | **Seoung Soo Lee** | **12/5/19 10:48:00 AM** |
| **Page 56: [29] Deleted** | **Seoung Soo Lee** | **12/5/19 10:48:00 AM** |
| **Page 56: [29] Deleted** | **Seoung Soo Lee** | **12/5/19 10:48:00 AM** |
| **Page 56: [29] Deleted** | **Seoung Soo Lee** | **12/5/19 10:48:00 AM** |
| **Page 56: [30] Deleted** | **Seoung Soo Lee** | **11/26/19 2:15:00 PM** |
| **Page 56: [30] Deleted** | **Seoung Soo Lee** | **11/26/19 2:15:00 PM** |
| **Page 56: [30] Deleted** | **Seoung Soo Lee** | **11/26/19 2:15:00 PM** |
| **Page 56: [30] Deleted** | **Seoung Soo Lee** | **11/26/19 2:15:00 PM** |
| **Page 56: [30] Deleted** | **Seoung Soo Lee** | **11/26/19 2:15:00 PM** |
| **Page 56: [30] Deleted** | **Seoung Soo Lee** | **11/26/19 2:15:00 PM** |
| **Page 56: [30] Deleted** | **Seoung Soo Lee** | **11/26/19 2:15:00 PM** |
| **Page 56: [30] Deleted** | **Seoung Soo Lee** | **11/26/19 2:15:00 PM** |

| Page 56: [30] Deleted | Seoung Soo Lee | 11/26/19 2:15:00 PM |
|---|---|---|

| Page 56: [30] Deleted | Seoung Soo Lee | 11/26/19 2:15:00 PM |
|---|---|---|

| Page 56: [30] Deleted | Seoung Soo Lee | 11/26/19 2:15:00 PM |
|---|---|---|

| Page 56: [31] Deleted | Seoung Soo Lee | 12/13/19 1:26:00 PM |
|---|---|---|

| Page 56: [31] Deleted | Seoung Soo Lee | 12/13/19 1:26:00 PM |
|---|---|---|

| Page 56: [31] Deleted | Seoung Soo Lee | 12/13/19 1:26:00 PM |
|---|---|---|

| Page 56: [31] Deleted | Seoung Soo Lee | 12/13/19 1:26:00 PM |
|---|---|---|

| Page 56: [31] Deleted | Seoung Soo Lee | 12/13/19 1:26:00 PM |
|---|---|---|

| Page 56: [31] Deleted | Seoung Soo Lee | 12/13/19 1:26:00 PM |
|---|---|---|

| Page 56: [31] Deleted | Seoung Soo Lee | 12/13/19 1:26:00 PM |
|---|---|---|

| Page 56: [31] Deleted | Seoung Soo Lee | 12/13/19 1:26:00 PM |
|---|---|---|

| Page 56: [31] Deleted | Seoung Soo Lee | 12/13/19 1:26:00 PM |
|---|---|---|

| Page 56: [31] Deleted | Seoung Soo Lee | 12/13/19 1:26:00 PM |
|---|---|---|

| | | |
|---|---|---|
| **Page 56: [31] Deleted** | **Seoung Soo Lee** | **12/13/19 1:26:00 PM** |
| **Page 56: [32] Deleted** | **Seoung Soo Lee** | **11/26/19 2:39:00 PM** |
| **Page 56: [32] Deleted** | **Seoung Soo Lee** | **11/26/19 2:39:00 PM** |
| **Page 56: [32] Deleted** | **Seoung Soo Lee** | **11/26/19 2:39:00 PM** |
| **Page 56: [32] Deleted** | **Seoung Soo Lee** | **11/26/19 2:39:00 PM** |
| **Page 56: [32] Deleted** | **Seoung Soo Lee** | **11/26/19 2:39:00 PM** |
| **Page 56: [32] Deleted** | **Seoung Soo Lee** | **11/26/19 2:39:00 PM** |
| **Page 58: [33] Deleted** | **Seoung Soo Lee** | **12/15/19 8:18:00 AM** |
| **Page 59: [34] Deleted** | **Seoung Soo Lee** | **11/28/19 2:45:00 PM** |
| **Page 59: [34] Deleted** | **Seoung Soo Lee** | **11/28/19 2:45:00 PM** |
| **Page 59: [34] Deleted** | **Seoung Soo Lee** | **11/28/19 2:45:00 PM** |
| **Page 59: [35] Deleted** | **Seoung Soo Lee** | **1/1/20 11:41:00 AM** |
| **Page 59: [35] Deleted** | **Seoung Soo Lee** | **1/1/20 11:41:00 AM** |
| **Page 59: [35] Deleted** | **Seoung Soo Lee** | **1/1/20 11:41:00 AM** |
| **Page 59: [35] Deleted** | **Seoung Soo Lee** | **1/1/20 11:41:00 AM** |

| | | |
|---|---|---|
| **Page 59: [35] Deleted** | **Seoung Soo Lee** | **1/1/20 11:41:00 AM** |
| **Page 59: [35] Deleted** | **Seoung Soo Lee** | **1/1/20 11:41:00 AM** |
| **Page 59: [35] Deleted** | **Seoung Soo Lee** | **1/1/20 11:41:00 AM** |
| **Page 59: [35] Deleted** | **Seoung Soo Lee** | **1/1/20 11:41:00 AM** |
| **Page 59: [36] Deleted** | **Seoung Soo Lee** | **12/15/19 8:23:00 AM** |
| **Page 59: [36] Deleted** | **Seoung Soo Lee** | **12/15/19 8:23:00 AM** |
| **Page 59: [36] Deleted** | **Seoung Soo Lee** | **12/15/19 8:23:00 AM** |
| **Page 59: [37] Deleted** | **Seoung Soo Lee** | **11/27/19 6:49:00 AM** |
| **Page 59: [37] Deleted** | **Seoung Soo Lee** | **11/27/19 6:49:00 AM** |
| **Page 59: [38] Deleted** | **Seoung Soo Lee** | **12/3/19 2:35:00 PM** |
| **Page 59: [38] Deleted** | **Seoung Soo Lee** | **12/3/19 2:35:00 PM** |
| **Page 59: [38] Deleted** | **Seoung Soo Lee** | **12/3/19 2:35:00 PM** |
| **Page 59: [39] Deleted** | **Seoung Soo Lee** | **11/27/19 6:50:00 AM** |
| **Page 59: [39] Deleted** | **Seoung Soo Lee** | **11/27/19 6:50:00 AM** |
| **Page 59: [39] Deleted** | **Seoung Soo Lee** | **11/27/19 6:50:00 AM** |

| Page 59: [39] Deleted | Seoung Soo Lee | 11/27/19 6:50:00 AM |
|---|---|---|

| Page 59: [39] Deleted | Seoung Soo Lee | 11/27/19 6:50:00 AM |
|---|---|---|

| Page 59: [39] Deleted | Seoung Soo Lee | 11/27/19 6:50:00 AM |
|---|---|---|

| Page 59: [39] Deleted | Seoung Soo Lee | 11/27/19 6:50:00 AM |
|---|---|---|

| Page 59: [39] Deleted | Seoung Soo Lee | 11/27/19 6:50:00 AM |
|---|---|---|

| Page 59: [39] Deleted | Seoung Soo Lee | 11/27/19 6:50:00 AM |
|---|---|---|

| Page 59: [39] Deleted | Seoung Soo Lee | 11/27/19 6:50:00 AM |
|---|---|---|

| Page 59: [39] Deleted | Seoung Soo Lee | 11/27/19 6:50:00 AM |
|---|---|---|

| Page 59: [39] Deleted | Seoung Soo Lee | 11/27/19 6:50:00 AM |
|---|---|---|

| Page 59: [39] Deleted | Seoung Soo Lee | 11/27/19 6:50:00 AM |
|---|---|---|

| Page 59: [40] Deleted | Seoung Soo Lee | 1/1/20 11:47:00 AM |
|---|---|---|

| Page 59: [40] Deleted | Seoung Soo Lee | 1/1/20 11:47:00 AM |
|---|---|---|

| Page 59: [41] Deleted | Seoung Soo Lee | 12/3/19 2:36:00 PM |
|---|---|---|

| Page 59: [41] Deleted | Seoung Soo Lee | 12/3/19 2:36:00 PM |
|---|---|---|

| Page 59: [42] Deleted | Seoung Soo Lee | 11/27/19 8:07:00 AM |
| Page 59: [42] Deleted | Seoung Soo Lee | 11/27/19 8:07:00 AM |
| Page 59: [42] Deleted | Seoung Soo Lee | 11/27/19 8:07:00 AM |
| Page 59: [43] Deleted | Seoung Soo Lee | 11/28/19 7:42:00 AM |
| Page 59: [43] Deleted | Seoung Soo Lee | 11/28/19 7:42:00 AM |
| Page 59: [43] Deleted | Seoung Soo Lee | 11/28/19 7:42:00 AM |
| Page 59: [43] Deleted | Seoung Soo Lee | 11/28/19 7:42:00 AM |
| Page 59: [43] Deleted | Seoung Soo Lee | 11/28/19 7:42:00 AM |
| Page 59: [43] Deleted | Seoung Soo Lee | 11/28/19 7:42:00 AM |
| Page 59: [43] Deleted | Seoung Soo Lee | 11/28/19 7:42:00 AM |
| Page 59: [43] Deleted | Seoung Soo Lee | 11/28/19 7:42:00 AM |
| Page 60: [44] Deleted | Seoung Soo Lee | 12/30/19 8:28:00 AM |
| Page 60: [44] Deleted | Seoung Soo Lee | 12/30/19 8:28:00 AM |
| Page 60: [44] Deleted | Seoung Soo Lee | 12/30/19 8:28:00 AM |
| Page 60: [45] Deleted | Seoung Soo Lee | 11/28/19 9:00:00 AM |

| | | |
|---|---|---|
| **Page 60: [45] Deleted** | **Seoung Soo Lee** | **11/28/19 9:00:00 AM** |
| **Page 60: [45] Deleted** | **Seoung Soo Lee** | **11/28/19 9:00:00 AM** |
| **Page 60: [45] Deleted** | **Seoung Soo Lee** | **11/28/19 9:00:00 AM** |
| **Page 60: [45] Deleted** | **Seoung Soo Lee** | **11/28/19 9:00:00 AM** |
| **Page 60: [45] Deleted** | **Seoung Soo Lee** | **11/28/19 9:00:00 AM** |
| **Page 60: [45] Deleted** | **Seoung Soo Lee** | **11/28/19 9:00:00 AM** |
| **Page 60: [45] Deleted** | **Seoung Soo Lee** | **11/28/19 9:00:00 AM** |
| **Page 60: [45] Deleted** | **Seoung Soo Lee** | **11/28/19 9:00:00 AM** |
| **Page 60: [45] Deleted** | **Seoung Soo Lee** | **11/28/19 9:00:00 AM** |
| **Page 60: [45] Deleted** | **Seoung Soo Lee** | **11/28/19 9:00:00 AM** |
| **Page 60: [45] Deleted** | **Seoung Soo Lee** | **11/28/19 9:00:00 AM** |
| **Page 60: [46] Deleted** | **Seoung Soo Lee** | **11/28/19 3:05:00 PM** |
| **Page 60: [46] Deleted** | **Seoung Soo Lee** | **11/28/19 3:05:00 PM** |
| **Page 60: [46] Deleted** | **Seoung Soo Lee** | **11/28/19 3:05:00 PM** |
| **Page 60: [46] Deleted** | **Seoung Soo Lee** | **11/28/19 3:05:00 PM** |

| Page 60: [46] Deleted | Seoung Soo Lee | 11/28/19 3:05:00 PM |

| Page 60: [46] Deleted | Seoung Soo Lee | 11/28/19 3:05:00 PM |

| Page 60: [46] Deleted | Seoung Soo Lee | 11/28/19 3:05:00 PM |

| Page 60: [46] Deleted | Seoung Soo Lee | 11/28/19 3:05:00 PM |

| Page 60: [46] Deleted | Seoung Soo Lee | 11/28/19 3:05:00 PM |

| Page 60: [46] Deleted | Seoung Soo Lee | 11/28/19 3:05:00 PM |

| Page 60: [46] Deleted | Seoung Soo Lee | 11/28/19 3:05:00 PM |

| Page 60: [47] Deleted | Seoung Soo Lee | 12/3/19 2:45:00 PM |

| Page 60: [47] Deleted | Seoung Soo Lee | 12/3/19 2:45:00 PM |

| Page 60: [48] Deleted | Seoung Soo Lee | 11/28/19 9:54:00 AM |

| Page 60: [48] Deleted | Seoung Soo Lee | 11/28/19 9:54:00 AM |

| Page 60: [48] Deleted | Seoung Soo Lee | 11/28/19 9:54:00 AM |

| Page 60: [48] Deleted | Seoung Soo Lee | 11/28/19 9:54:00 AM |

| Page 60: [48] Deleted | Seoung Soo Lee | 11/28/19 9:54:00 AM |

| Page 60: [48] Deleted | Seoung Soo Lee | 11/28/19 9:54:00 AM |

| Page 60: [48] Deleted | Seoung Soo Lee | 11/28/19 9:54:00 AM |
| --- | --- | --- |

| Page 60: [49] Deleted | Seoung Soo Lee | 11/29/19 4:16:00 PM |
| --- | --- | --- |

| Page 60: [49] Deleted | Seoung Soo Lee | 11/29/19 4:16:00 PM |
| --- | --- | --- |

| Page 60: [49] Deleted | Seoung Soo Lee | 11/29/19 4:16:00 PM |
| --- | --- | --- |

| Page 60: [49] Deleted | Seoung Soo Lee | 11/29/19 4:16:00 PM |
| --- | --- | --- |

| Page 60: [50] Deleted | Seoung Soo Lee | 11/29/19 4:16:00 PM |
| --- | --- | --- |

| Page 60: [50] Deleted | Seoung Soo Lee | 11/29/19 4:16:00 PM |
| --- | --- | --- |

| Page 60: [50] Deleted | Seoung Soo Lee | 11/29/19 4:16:00 PM |
| --- | --- | --- |

| Page 60: [50] Deleted | Seoung Soo Lee | 11/29/19 4:16:00 PM |
| --- | --- | --- |

| Page 60: [51] Deleted | Seoung Soo Lee | 11/28/19 3:16:00 PM |
| --- | --- | --- |

| Page 60: [51] Deleted | Seoung Soo Lee | 11/28/19 3:16:00 PM |
| --- | --- | --- |

| Page 60: [51] Deleted | Seoung Soo Lee | 11/28/19 3:16:00 PM |
| --- | --- | --- |

| Page 60: [51] Deleted | Seoung Soo Lee | 11/28/19 3:16:00 PM |
| --- | --- | --- |

| Page 60: [52] Deleted | Seoung Soo Lee | 11/29/19 4:27:00 PM |
| --- | --- | --- |

| | | |
|---|---|---|
| **Page 60: [52] Deleted** | **Seoung Soo Lee** | **11/29/19 4:27:00 PM** |
| **Page 60: [52] Deleted** | **Seoung Soo Lee** | **11/29/19 4:27:00 PM** |
| **Page 60: [52] Deleted** | **Seoung Soo Lee** | **11/29/19 4:27:00 PM** |
| **Page 61: [53] Deleted** | **Seoung Soo Lee** | **11/27/19 7:02:00 AM** |
| **Page 61: [53] Deleted** | **Seoung Soo Lee** | **11/27/19 7:02:00 AM** |
| **Page 61: [53] Deleted** | **Seoung Soo Lee** | **11/27/19 7:02:00 AM** |
| **Page 61: [53] Deleted** | **Seoung Soo Lee** | **11/27/19 7:02:00 AM** |
| **Page 61: [53] Deleted** | **Seoung Soo Lee** | **11/27/19 7:02:00 AM** |
| **Page 61: [53] Deleted** | **Seoung Soo Lee** | **11/27/19 7:02:00 AM** |
| **Page 61: [53] Deleted** | **Seoung Soo Lee** | **11/27/19 7:02:00 AM** |
| **Page 61: [53] Deleted** | **Seoung Soo Lee** | **11/27/19 7:02:00 AM** |
| **Page 61: [53] Deleted** | **Seoung Soo Lee** | **11/27/19 7:02:00 AM** |
| **Page 61: [53] Deleted** | **Seoung Soo Lee** | **11/27/19 7:02:00 AM** |
| **Page 61: [53] Deleted** | **Seoung Soo Lee** | **11/27/19 7:02:00 AM** |

| Page 61: [53] Deleted | Seoung Soo Lee | 11/27/19 7:02:00 AM |
|---|---|---|

| Page 61: [53] Deleted | Seoung Soo Lee | 11/27/19 7:02:00 AM |
|---|---|---|

| Page 61: [53] Deleted | Seoung Soo Lee | 11/27/19 7:02:00 AM |
|---|---|---|

| Page 61: [53] Deleted | Seoung Soo Lee | 11/27/19 7:02:00 AM |
|---|---|---|

| Page 61: [53] Deleted | Seoung Soo Lee | 11/27/19 7:02:00 AM |
|---|---|---|

| Page 61: [53] Deleted | Seoung Soo Lee | 11/27/19 7:02:00 AM |
|---|---|---|

| Page 61: [53] Deleted | Seoung Soo Lee | 11/27/19 7:02:00 AM |
|---|---|---|

| Page 61: [53] Deleted | Seoung Soo Lee | 11/27/19 7:02:00 AM |
|---|---|---|

| Page 61: [53] Deleted | Seoung Soo Lee | 11/27/19 7:02:00 AM |
|---|---|---|

| Page 61: [53] Deleted | Seoung Soo Lee | 11/27/19 7:02:00 AM |
|---|---|---|

| Page 61: [54] Deleted | Seoung Soo Lee | 11/29/19 4:36:00 PM |
|---|---|---|

| Page 61: [54] Deleted | Seoung Soo Lee | 11/29/19 4:36:00 PM |
|---|---|---|

| Page 61: [54] Deleted | Seoung Soo Lee | 11/29/19 4:36:00 PM |
|---|---|---|

| Page 61: [54] Deleted | Seoung Soo Lee | 11/29/19 4:36:00 PM |
|---|---|---|

| Page 61: [54] Deleted | Seoung Soo Lee | 11/29/19 4:36:00 PM |
|---|---|---|

| Page 61: [54] Deleted | Seoung Soo Lee | 11/29/19 4:36:00 PM |
| Page 61: [54] Deleted | Seoung Soo Lee | 11/29/19 4:36:00 PM |
| Page 61: [54] Deleted | Seoung Soo Lee | 11/29/19 4:36:00 PM |
| Page 61: [54] Deleted | Seoung Soo Lee | 11/29/19 4:36:00 PM |
| Page 61: [54] Deleted | Seoung Soo Lee | 11/29/19 4:36:00 PM |
| Page 61: [54] Deleted | Seoung Soo Lee | 11/29/19 4:36:00 PM |
| Page 61: [54] Deleted | Seoung Soo Lee | 11/29/19 4:36:00 PM |
| Page 61: [54] Deleted | Seoung Soo Lee | 11/29/19 4:36:00 PM |
| Page 61: [54] Deleted | Seoung Soo Lee | 11/29/19 4:36:00 PM |
| Page 61: [54] Deleted | Seoung Soo Lee | 11/29/19 4:36:00 PM |
| Page 61: [54] Deleted | Seoung Soo Lee | 11/29/19 4:36:00 PM |
| Page 61: [54] Deleted | Seoung Soo Lee | 11/29/19 4:36:00 PM |
| Page 61: [54] Deleted | Seoung Soo Lee | 11/29/19 4:36:00 PM |
| Page 61: [54] Deleted | Seoung Soo Lee | 11/29/19 4:36:00 PM |
| Page 61: [54] Deleted | Seoung Soo Lee | 11/29/19 4:36:00 PM |

| Page 61: [54] Deleted | Seoung Soo Lee | 11/29/19 4:36:00 PM |
|---|---|---|

| Page 61: [54] Deleted | Seoung Soo Lee | 11/29/19 4:36:00 PM |
|---|---|---|

| Page 61: [54] Deleted | Seoung Soo Lee | 11/29/19 4:36:00 PM |
|---|---|---|

| Page 61: [54] Deleted | Seoung Soo Lee | 11/29/19 4:36:00 PM |
|---|---|---|

| Page 61: [54] Deleted | Seoung Soo Lee | 11/29/19 4:36:00 PM |
|---|---|---|

| Page 61: [54] Deleted | Seoung Soo Lee | 11/29/19 4:36:00 PM |
|---|---|---|

| Page 61: [54] Deleted | Seoung Soo Lee | 11/29/19 4:36:00 PM |
|---|---|---|

| Page 61: [54] Deleted | Seoung Soo Lee | 11/29/19 4:36:00 PM |
|---|---|---|

| Page 61: [54] Deleted | Seoung Soo Lee | 11/29/19 4:36:00 PM |
|---|---|---|

| Page 61: [54] Deleted | Seoung Soo Lee | 11/29/19 4:36:00 PM |
|---|---|---|

| Page 61: [54] Deleted | Seoung Soo Lee | 11/29/19 4:36:00 PM |
|---|---|---|

| Page 61: [54] Deleted | Seoung Soo Lee | 11/29/19 4:36:00 PM |
|---|---|---|

| Page 61: [54] Deleted | Seoung Soo Lee | 11/29/19 4:36:00 PM |
|---|---|---|

| Page 61: [54] Deleted | Seoung Soo Lee | 11/29/19 4:36:00 PM |
|---|---|---|

| Page 61: [54] Deleted | Seoung Soo Lee | 11/29/19 4:36:00 PM |
|---|---|---|

| Page 61: [54] Deleted | Seoung Soo Lee | 11/29/19 4:36:00 PM |
|---|---|---|

| Page 61: [54] Deleted | Seoung Soo Lee | 11/29/19 4:36:00 PM |
|---|---|---|

| Page 61: [54] Deleted | Seoung Soo Lee | 11/29/19 4:36:00 PM |
|---|---|---|

| Page 61: [54] Deleted | Seoung Soo Lee | 11/29/19 4:36:00 PM |
|---|---|---|

| Page 61: [54] Deleted | Seoung Soo Lee | 11/29/19 4:36:00 PM |
|---|---|---|

| Page 61: [54] Deleted | Seoung Soo Lee | 11/29/19 4:36:00 PM |
|---|---|---|

| Page 61: [54] Deleted | Seoung Soo Lee | 11/29/19 4:36:00 PM |
|---|---|---|

| Page 62: [55] Deleted | Seoung Soo Lee | 11/29/19 4:57:00 PM |
|---|---|---|

| Page 62: [56] Deleted | Seoung Soo Lee | 12/4/19 7:42:00 AM |
|---|---|---|

| Page 62: [57] Deleted | Seoung Soo Lee | 11/29/19 5:09:00 PM |
|---|---|---|

| Page 65: [58] Deleted | Seoung Soo Lee | 12/4/19 9:40:00 AM |
|---|---|---|

| Page 65: [58] Deleted | Seoung Soo Lee | 12/4/19 9:40:00 AM |
|---|---|---|

| Page 65: [58] Deleted | Seoung Soo Lee | 12/4/19 9:40:00 AM |
|---|---|---|

| Page 65: [58] Deleted | Seoung Soo Lee | 12/4/19 9:40:00 AM |
|---|---|---|

| | | |
|---|---|---|
| **Page 65: [58] Deleted** | **Seoung Soo Lee** | **12/4/19 9:40:00 AM** |

| | | |
|---|---|---|
| **Page 65: [59] Deleted** | **Seoung Soo Lee** | **12/7/19 8:59:00 AM** |

| | | |
|---|---|---|
| **Page 65: [60] Deleted** | **Seoung Soo Lee** | **11/27/19 8:38:00 AM** |

| Page 65: [60] Deleted | Seoung Soo Lee | 11/27/19 8:38:00 AM |

| Page 65: [60] Deleted | Seoung Soo Lee | 11/27/19 8:38:00 AM |

| Page 65: [60] Deleted | Seoung Soo Lee | 11/27/19 8:38:00 AM |

| Page 65: [60] Deleted | Seoung Soo Lee | 11/27/19 8:38:00 AM |

| Page 65: [60] Deleted | Seoung Soo Lee | 11/27/19 8:38:00 AM |

| Page 65: [60] Deleted | Seoung Soo Lee | 11/27/19 8:38:00 AM |

| Page 65: [60] Deleted | Seoung Soo Lee | 11/27/19 8:38:00 AM |

| Page 65: [60] Deleted | Seoung Soo Lee | 11/27/19 8:38:00 AM |

| Page 65: [60] Deleted | Seoung Soo Lee | 11/27/19 8:38:00 AM |

| Page 65: [60] Deleted | Seoung Soo Lee | 11/27/19 8:38:00 AM |

| Page 65: [60] Deleted | Seoung Soo Lee | 11/27/19 8:38:00 AM |

| Page 65: [60] Deleted | Seoung Soo Lee | 11/27/19 8:38:00 AM |

| Page 65: [60] Deleted | Seoung Soo Lee | 11/27/19 8:38:00 AM |

| Page 65: [60] Deleted | Seoung Soo Lee | 11/27/19 8:38:00 AM |

| Page 65: [60] Deleted | Seoung Soo Lee | 11/27/19 8:38:00 AM |

| Page 65: [60] Deleted | Seoung Soo Lee | 11/27/19 8:38:00 AM |
| --- | --- | --- |

| Page 65: [60] Deleted | Seoung Soo Lee | 11/27/19 8:38:00 AM |
| --- | --- | --- |

| Page 65: [60] Deleted | Seoung Soo Lee | 11/27/19 8:38:00 AM |
| --- | --- | --- |

| Page 65: [61] Deleted | Seoung Soo Lee | 11/27/19 2:47:00 PM |
| --- | --- | --- |

| Page 65: [61] Deleted | Seoung Soo Lee | 11/27/19 2:47:00 PM |
| --- | --- | --- |

| Page 65: [61] Deleted | Seoung Soo Lee | 11/27/19 2:47:00 PM |
| --- | --- | --- |

| Page 65: [61] Deleted | Seoung Soo Lee | 11/27/19 2:47:00 PM |
| --- | --- | --- |

| Page 65: [61] Deleted | Seoung Soo Lee | 11/27/19 2:47:00 PM |
| --- | --- | --- |

| Page 65: [61] Deleted | Seoung Soo Lee | 11/27/19 2:47:00 PM |
| --- | --- | --- |

| Page 65: [61] Deleted | Seoung Soo Lee | 11/27/19 2:47:00 PM |
| --- | --- | --- |

| Page 65: [61] Deleted | Seoung Soo Lee | 11/27/19 2:47:00 PM |
| --- | --- | --- |

| Page 65: [61] Deleted | Seoung Soo Lee | 11/27/19 2:47:00 PM |
| --- | --- | --- |

| Page 65: [61] Deleted | Seoung Soo Lee | 11/27/19 2:47:00 PM |
| --- | --- | --- |

| Page 76: [62] Deleted | Seoung Soo Lee | 1/10/20 9:44:00 AM |
| --- | --- | --- |

---

## Author Response (AR3)

Authors appreciate the editor's comments and we responded to them as follows.

Dear Authors,
After reviewing the changes to your manuscript, I recommend acceptance with 2 minor changes:
1) The idea of invigoration dates to Koren et al., 2005 doi:10.1029/2005GL023187. Please use this as your main reference.

The recommended reference is added as a main one.

2) In the opening sentence, of the abstract, please change the wording to "Using a modeling framework.." (Not "through").

Corrected.

I know that this process has taken longer than hoped but I do believe the papers is much clearer now.
Sincerely,
Graham Feingold